

**Late Miocene-Pliocene climate evolution recorded by the red clay covered**
**on the Xiaoshuizi planation surface, NE Tibetan Plateau**
Xiaomiao Li[1], Tingjiang Peng[1], Zhenhua Ma[1], Meng Li[1], Zhantao Feng[1], Benhong Guo[1], Hao Yu[1], Xiyan
Ye[1], Zhengchuang Hui[1], Chunhui Song[2], Jijun Li[1,3]
1. MOE Key Laboratory of Western China's Environmental Systems, College of Earth and Environmental
Sciences, Lanzhou University, Lanzhou 730000, China
2. School of Earth Sciences, Key Laboratory of Western China's Mineral Resources of Gansu Province,
Lanzhou University, Lanzhou 730000, China
3. College of Geography Science, Nanjing Normal University, Nanjing 210023,China
___________________________________________________
*Corresponding author: Key Laboratory of Western China's Environmental Systems
(Ministry of Education), College of Earth and Environmental Science, Lanzhou University,
Lanzhou 730000, China; *E-mail address*: lijj@lzu.edu.cn(J.J. Li), *Fax*: +86-931-891-2724;
*Tel.:* +86-931-891-2724



**Abstract**

24        As an analogue for predicting the future climate, Pliocene climate and its driving

mechanism attract much attention for a long time. Late Miocene-Pliocene red clay sequence
on the main Chinese Loess Plateau (CLP) has been widely applied to reconstruct the history
of interior aridification and Asian monsoon climate. However, the typical red clay sequences
deposited on the planation surface of Tibetan Plateau are rare. Recently, continuous red clay
has been found on the uplifted Xiaoshuizi peneplain in the Maxian Mountains, northeastern
(NE) Tibetan Plateau (TP). To reconstruct the late Miocene-early Pliocene climate history of
NE Tibetan Plateau and to assess the regional differences between the central and western
CLP, multiple climatic proxies were analyzed from the Xiaoshuizi red clay sequence. Our
results demonstrate the minimal weathering and pedogenesis from 6.7 to 4.8 Ma, which
implicates that the climate was sustained arid. We speculate that precipitation delivered by
the paleo-Asian Summer Monsoon (ASM) was limited during this period, and instead the
intensification of the westerlies circulation resulted in arid condition in the study region.
Subsequently, enhanced weathering and pedogenesis occurred during the interval of 4.8-3.6
Ma, which attests to increasing effective moisture. Thus, we ascribe the obvious arid-humid
climate transition near 4.8 Ma to the palaeo-ASM expansion. Increasing Arctic temperatures,
the vast poleward expansion of the tropical warm pool into the subtropical regions and water
freshening in the subtropical Pacific in response to the closure of the Panamanian Seaway
may have been responsible for the thermodynamical enhancement of the palaeo-ASM system,
which permitted more moisture to be carried to the NE Tibetan Plateau.
**Keywords:** Late Miocene-earlyPliocene; Xiaoshui Peneplain; Red Clay; Palaeo-ASM;



Westerly Circulation

**1. Introduction**

The Pliocene, including the Zanclean (5.33-3.60 Ma) and Piacenzian (3.60-2.58 Ma)

stages, is one of the most intensively studied intervals of the pre-Quaternary. The Zanclean
climate was generally warm and wet and it is analogous to the present day in terms of (i)
land-sea distribution, (ii) orbital configuration, (iii) carbon dioxide levels ranging from 280-
380 ppm (Raymo et al., 1996; Fedorov et al., 2013), and (iv) comparable temperatures in the
tropic region. In addition, both the Holocene and Zanclean are transitional periods from cold
to warm climatic condition. For these reasons, the early Pliocene climate is often used as  an
analogue for that of the Holocene and attracts much attention. On the other hand, Zanclean is
unique and some crucial transitions of the thermorhaline and atmospheric circulation  towards
modern conditions were undergoing. Temperatures at the high northern latitudes  were
considerably higher and therefore continental glaciers were almost absent from the Northern
Hemisphere (Ballantyne et al., 2010; Dowsett et al., 2010). The warm and wet climate
prevailed across the major continents and the warm Arctic is thought to have resulted from a
greenhouse effect caused by higher atmospheric moisture content (Abbot and Tziperman,
2008). The low meridional surface temperature gradient resulted in an "equable" climate
during this interval (Abbot and Tziperman., 2008; Fedorov et al., 2013). The east-west sea
surface temperature gradient in the tropical Pacific during this interval is also believed to  be
low, which is tightly linked with El Nino Southern Oscillation (Lawrence et al., 2006).
However, debate persists on whether permanent El Nino–like conditions were  sustained



during the Pliocene (Wara et al., 2005; Watanabe et al., 2011; Zhang et al., 2014).
Meanwhile, the most significant tectonic movements were the uplift of the TP (Li et al., 2015;
Zheng et al., 2000 ; Fang et al., 2005a, 2005b) and gradual closing of the Panama seaway
(Lunt et al., 2008; Haug et al., 1998, 2005). These tectonic movements resulted in major
changes in the global thermohaline and atmospheric circulation system which were thought to
make crucial preconditions for both appearing of ice sheet in northern hemisphere at ~3Ma
(Haug et al., 1998; Driscoll et al., 1998) and development of modern east-west
hydrographic gradient in the equatorial Pacific (Lawrence et al., 2006; Chaisson et al., 2000).
The ASM and meridional (westerlies) circulation systems, as major components of
atmospheric circulation, delivered moisture to Eurasia which might have prepared enough
moisture for long-term growth of ice sheet in northern hemisphere between 3 and 2 Ma
(Driscoll et al., 1998). Make clear the evolution of the palaeo-ASM and westerlies during
early Pliocene is critical to understanding formation mechanism of ice sheet at the Northern
high latitudes. Furthermore, the palaeo-ASM might be dynamically linked with the TP uplift,
changes in latitudinal and longitudinal heat gradients, global temperature and ice volume
during early Pliocene. Warm and wet climate background tends to yield wet climate condition
while reductions in the east-west sea surface temperature (SST) gradient in the tropical
Pacific results in a weakened summer monsoon (Wang et al., 2000). Several studies have
shown that a major atmospheric teleconnection links the ASM with both Arctic volume and
the TP uplift (Ding et al., 1990; Li et al., 1991; An et al., 2001; Clift et al., 2008; Sun et al.,
2015). Thus, it is crucial to make clear what the climate was like in East Asia under such
warm and equable climatic conditions in the Northern Hemisphere.



Previous research has revealed that since the late Miocene, red clay widely deposited
across the CLP, indicating that the onset of interior Asian aridification related to the uplift of
the TP occurred (Guo et al., 2001; Song et al., 2007; An et al., 2014; Ao et al., 2016; Li et al.,
2017). Element, strata and pollen evidence from the Qaidam and Tarim basin demonstrated
that the aridification had intensified since early Pliocene (Fang et al., 2008; Sun et al., 2006a,
2017; Chang et al., 2013; Liu et al., 2014). In eastern and central CLP, palaeontological
evidence, mineral magnetic parameters and geochemical records from the red clay  also
indicate a dry climate condition during late Miocene, however, aridification process  was
interrupted by a long interval of wet climate during the early Pliocene (Wang et al., 2006;
Guo et al., 2001; Wu et al., 2006; Song et al., 2007; Sun et al., 2010;  Ao et al., 2016). The
most controversial climate change occurred during the  interval  of  4.8-4.1  Ma,  for  which
climate reconstructions from different proxies reveal conflicting palaeoenvironmental trends.
For example, field observations and pollen records indicate an intensified monsoon system,
but low magnetic susceptibility values are more consistent with arid rather than wet climatic
conditions (Ding et al., 2001; Ma et al., 2005; Song et al., 2007; Sun et al., 2010). It's thought
to be substantial gleying resulted from large amount precipitation which made magnetic
susceptibility invalid over this period (Ding et al., 2001). Obviously, climate changes in
westerlies dominated regions and monsoon dominated regions are discrepancy.  The
inconsistent climate change may be related to different response of westerlies and the  palaeo-
ASM to global climate changes and the TP uplift during early Pliocene. To clarify the
evolution and dynamic of westerlies and the palaeo-ASM, requires accurate paleoclimatic
reconstructions in the CLP, especially in the western CLP.



Till now, early Pliocene paleoclimatic records from the western CLP red clay are
lacking. Recently, continuous red clay has been found on the uplifted Xiaoshuizi (XSZ)
peneplain in the NE Tibetan Plateau and well dated via high-resolution magnetostratigraphy
analysis (Li et al., 2017). The special gemorophological and climatic characteristic of the
Xiaoshuizi red clay makes it different from the main CLP red clay, and provides particular
opportunity to reveal the late Miocene-early Pliocene climate history in NE Tibetan Plateau
and discuss the climatic difference between the central and western CLP red clay. In this
study, multiple climatic proxies have been applied in the Xiaoshuizi late Miocene-Pliocene
red clay sequence. Then we reconstruct the detailed precipitation, chemical weathering and
pedogensis history in the Xiaoshuizi planation surface during the interval of 6.7-3.6 Ma.
Finally, the regional climate and possible mechanism have been further discussed.

**123    2. Regional background**

The XSZ planation surface locates in Yuzhong County in the western Chinese Loess
Plateau (Fig. 1). The main XSZ planation surface is at an altitude of 2800 m in the
Maxianshan Mountains where it has truncated Precambrian gneiss. The Maxianshan are
rejuvenated mountains which protrude into the broad Longzhong Basin, and are in a
climatically sensitive zone because of the combined influences of the Asian Monsoon and the
northern branch of the mid-latitude westerly circulation system. The planation surface is
mantled by over 30 m of loess and over 40 m of red clay. Our previous bio-
magnetostratigraphic study has demonstrated that the red clay sequence covered on the XSZ
peneplain is dated to ~6.9-3.6 Ma (Li et al., 2017), here we choose the Xiaoshuizi core to



discuss the regional climate because of its continuous deposit and whole timescale relative to
the Shangyantan core mentioned in Li et al (2017). Yuzhong County lies within the semi-arid
temperate climate zone at the junction of the eastern monsoon area, the arid area of northwest
china, and the Tibetan Plateau cold region. The East Asian Monsoon system and the westerly
circulation operate together. The mean annual temperature is about 6.7 ℃ and the
precipitation amount is 300-800 mm. The spatial distribution of precipitation is uneven,
decreasing from south to north in Yuzhong County. Precipitation amount increases with
elevation at a rate of 27 mm per 100 m, attaining a maximum of 800 mm.

**3. Material and methods**
The XSZ core (35.81154 °N;103.8623 °E and 2758.1 m above sea level) is composed of
42 m of pure red clay and ~3 m of red clay with an increasing angular gravel content. The red
clay is composed of brownish red and yellowish clay layers. The upper 20 m is impregnated
with many horizontal carbonate nodule horizons and most of these horizons underline the
brownish red layer; there are occasional carbonized plant root channels, elliptical worm
burrows and snail fossil fragments. Fe-Mn stains are more frequent in the brownish layers
than in the yellowish layers, which is also the case for horizons containing carbonized root
channels. The red clay over the Xiaoshuizi planation surface is similar to that of typical
eolian red clay in the CLP, all of which are characterized by many carbonate nodule-rich
horizons. Grain-size, carbonate content and magnetic susceptibility samples were taken at 5-
cm intervals, while samples for geochemical analysis were collected at 25-cm intervals. Each
sample age was modeled using linear interpolation to derive absolute ages, constrained by our



previous magnetostratigraphy study. The grain-size distribution of samples was measured
with a Malvern Mastersizer 2000 with a detection range of 0.02-2000μm.  Magnetic
susceptibility ($\chi$) was measured using a Bartington MS2 meter and MS2B dual-frequency
sensor at two frequencies (470 Hz and 4700 Hz, designated $\chi_{lf}$ and $\chi_{hf}$, respectively). Three
measurements were made at each frequency and the final results were averaged.  The
frequency-dependent magnetic susceptibility ($\chi_{fd}$) was calculated as $\chi_{lf}$–$\chi_{hf}$. Chemical
composition was measured using Panalytical Magix PW2403. The sample preparation
procedure for XRF analysis was as follows: first, the bulk sample was heated to 35℃ for 7
days, then each sample was ground to less than 75μm using an agate mortar, and finally about
4 g of powdered sample was pressed into a pellet with a borate coating using a semiautomatic
oil-hydraulic laboratory press (model YYJ-40). All the measurements were finished in the
MOE Key Laboratory of Western China's Environmental Systems, Lanzhou University. The
molar content of silicate Ca (CaO*) was calculated using the following equation:
$$CaO^*(mol) = CaO(mol) - CaCO_3(mol) - \frac{10}{3} * \frac{P_2O_5}{M(P_2O_5)}$$

The carbonate content was measured with a calcimeter using the volumetric method of

Avery and Bascomb (1974) in the Key Laboratory of Mineral Resources in Western China
(Gansu Province), Lanzhou University.

**4. Results**

**Carbonate content**

According to the fluctuations in carbonate content, the red clay sequence was divided

into two intervals: *Interval - I* is from 6.7-4.8 Ma, during which the carbonate content



fluctuates from 3.8-39.2% with an average of 17.4%; the amplitude of fluctuations is small
and the carbonate content decreases upwards. From 5.4-4.9 Ma, the carbonate content
fluctuations are of greater amplitude than during 6.7-5.4 Ma. *Interval - II* is from 4.8-3.6Ma,
during which the carbonate content fluctuates from 1.6-39.1% with an average of 13.8%.
From 4.8-3.9 Ma there are several leaching-accumulation layers with <7% carbonate content
in the leached loess layers and >20% carbonate content in the accumulation layers.
**Element geochemistry**
The XSZ red clay consists mainly of $SiO_2$, $Al_2O_3$, CaO and $Fe_2O_3$ with low
concentrations (<5%) of MgO, $K_2O$, $Na_2O$, Sr, Rb and Ba (Table 1). The variations in $Al_2O_3$
and $K_2O$ are synchronous and roughly opposite to that of CaO. The variations in CaO show
the same trend as carbonate content. When the carbonate content is high, CaO is high, while
$Al_2O_3$ and $K_2O$ are low. The contents of $Al_2O_3$ and $K_2O$ from 4.8-3.6 Ma are clearly higher
than those from 6.7-4.8 Ma. The variations in these element concentrations from 4.8-3.6 Ma
are also greater than those from 6.7-4.8 Ma. The changes in Sr are similar to those of CaO,
but opposite to those of Ba and Rb.
**Magnetic susceptibility**
During interval I (6.9-4.8 Ma), $\chi_{hf}$ changes from 9.6-33.3×$10^{-8}$ $m^3$/kg with an average of
19.4×$10^{-8}$$m^3$/kg. $\chi_{lf}$ ranges from 11.4-36.1×$10^{-8}$ $m^3$/kg with an average of 20.3×$10^{-8}$ $m^3$/kg,
whilst $\chi_{fd}$ fluctuates from 0-2.8×$10^{-8}$ $m^3$/kg with an average of 1.0×$10^{-8}$ $m^3$/kg. During interval
II (4.8-3.58 Ma), $\chi_{hf}$ ranges from 12.8-53.9×$10^{-8}$ $m^3$/kg with an average of 25.4×$10^{-8}$ $m^3$/kg, $\chi_{lf}$
ranges from 13.56-59.0×$10^{-8}$ $m^3$/kg with an average of 26.9×$10^{-8}$ $m^3$/kg and $\chi_{fd}$ ranges from 0-
4.7×$10^{-8}$ $m^3$/kg with an average of 1.2×$10^{-8}$ $m^3$/kg. Clearly, the average values of the three



parameters are larger during interval Ⅱ than during interval Ⅰ; the amplitudes and durations
of the fluctuations of the three parameters during interval Ⅱ are also larger and longer than
those during the interval Ⅰ. From 4.8-4.7 Ma, 4.6-4.25 Ma and from 4.1-3.9 Ma, the values
of the three parameters are high, and they exhibit peaks from 4.6-4.25 Ma.
**Grain-size analysis**
The average clay content (<2μm ) is 8.2% during interval Ⅰ and 8.0% during interval Ⅱ.
The fluctuations in clay content are minor, except for maxima at about 5 Ma, 4.6 Ma and 4.2
Ma. The coarse silt component (>43μm), mainly carried by the East Asian Winter Monsoon,
exhibits a different trend to that of the clay content. From 6.7-4.8 Ma, the >43μm curve is
characterized by low values and high-frequency fluctuations, while after 4.8 Ma it exhibits
high values and long-duration fluctuations.

**5. Discussion**
**5.1 Paleoenvironmental explanation of the proxies**
The carbonate content of aeolian sediments is sensitive to varying climatic conditions,
and can be readily remobilized and deposited. Previous studies demonstrated that carbonate
in the loess-red clay sequence on the CLP records varies with precipitation (Fang et al., 1999;
Sun et al., 2010). The carbonate is mainly derived from a mixture of airborne dusts (Fang et
al., 1999). Soil micromorphological evidence from the Lanzhou loess demonstrates that
carbonate grains in loess are little altered, while those in the palaeosols have undergone a
reduction in size as a result of leaching and reprecipitation in the lower Bk horizons as
secondary carbonate (Fang et al., 1994, 1999). Furthermore, seasonal alternations between





wet and dry conditions are thought to be a key factor in driving carbonate dissolution and
reprecipitation (Sun et al., 2010). Thus, changes in carbonate content are generally controlled
by the effective precipitation. When effective precipitation is high, carbonate leaching
increases, and vice versa. So the carbonate content is regarded as an effective precipitation
proxy for studying wet-dry oscillations as well as summer monsoon evolution (Fang et al.,
1999; Sun et al., 2010).
Chemical weathering intensity is generally evaluated by the ratio of mobile (i.e., K, Ca,
Sr and Na) to non-mobile elements (e.g., Al and Rb). In general, Sr shows analogous
geochemical behavior to Ca and is easily released into solution and mobilized in the course of
weathering, while Rb is relatively immobile under moderate weathering conditions due to
strong adsorption to clay minerals (Nesbitt et al., 1980; Liu et al., 1993). Thus, the Rb/Sr ratio
potentially reflects chemical weathering intensity. However, the initial Rb/Sr ratio can be
affected by the precipitation of secondary carbonate leached from overlying sediments during
pedogenesis (Chang et al., 2013; Buggle et al., 2011), which may limit its environmental
significance. The correlation between Sr and CaO* (silicate CaO) is significant in the XSZ
section, while the correlation between Sr and $CaCO_3$ is not significant (99% confidence
interval). Thus we speculate that the Rb/Sr ratio mainly reflects the weathering intensity in
our studied samples (Fig. 4 e and f). In addition, previous study has proposed that the
$K_2O/Al_2O_3$ ratio can also indicate the weathering intensity. $Al_2O_3$ is typically chosen to
measure the mobility of elements due to its high stability (Taylor et al., 1983), while $K_2O$
(mainly produced by the physical weathering of potash feldspar) is easily leached from
primary minerals and then absorbed by secondary clay minerals with ongoing weathering



(Yang et al., 2006; Liang et al., 2013). In the arid and semi-arid regions of Asia, $K_2O$ is
enriched in palaeosols compared to loess horizons (Yang et al., 2006), meaning that the
enrichment of $K_2O$ is positively related with the amount of secondary clay. Thus, to some
extent, $K_2O/Al_2O_3$ reflects the amount of secondary clay and hence weathering intensity.
Generally, the $K_2O/Na_2O$ ratio is used to evaluate the clay content in loess and is also a
measure of plagioclase weathering, avoiding biases due to uncertainties in separating
carbonate Ca from silicate Ca (Liu et al., 1993 ; Buggle et al., 2011). As the product of
plagioclase weathering, $Na_2O$ is easily leached by increasing precipitation. As mentioned
above, $K_2O$ is easily absorbed by secondary clay particles, meaning that high $K_2O/Na_2O$
ratio is indicative of intense chemical weathering.

In the red clay-loess sequence of the CLP, magnetic parameters and clay ($<2$ μm)

content are well correlated and thus are regarded as the proxies of the ASM strength (Liu et
al., 2004). Eolian particles usually have two distinct magnetic components consisting of
detrital and pedogenic material (Liu et al., 2004). $\chi_{lf}$ can reflect the combined susceptibility of
both two components, but changes in $\chi_{lf}$ are dominantly affected by changes in the
concentration of pedogenic grains (Liu et al., 2004). Grain size distribution of pedogenic
particles confining within the superparamagnetic (SP) and single-domain (SD) grain size has
been proven to be steady (Liu et al., 2004, 2005). $\chi_{fd}$ can detect superparamagnetic minerals
produced by pedogenesis and therefore the correlation coefficient between $\chi_{lf}$ and $\chi_{fd}$ can
measure the contribution of SP grains ($<0.03$ μm for magnetite) to the bulk susceptibility (Liu
et al., 2004; Xia et al., 2014). As shown in Figure 4A, $\chi_{lf}$ is positively correlated with $\chi_{fd}$,
which means that the magnetic susceptibility of the XSZ red clay mostly reflects pedogenic



enhancement of the primary eolian ferromagnetic content through the in situ formation of
fine-grained ferrimagnetic material. This means that the magnetic susceptibility of the red
clay on the XSZ planation surface reflects pedogenic intensity. Both the original and
pedogenic magnetic signals can be separated using a simple linear regression method (Liu et
al., 2004; Xia et al., 2014). We use this method to extract the original magnetic component
($\chi_0$) and the pedogenic magnetite/maghemite component ($\chi_{pedo}$). In this study, $\chi_{fd}$ explains 11%
of the susceptibility in terms of pedogenic magnetite/maghemite ($\chi_{pedo} = \chi_{fd} / 0.11$).
Pedogenesis results in enhanced secondary clay formation (Sun et al., 2006); however,
not all of the clay particles are derived from in situ pedogenesis, but rather are inherited from
aeolian transport and deposition. Clay particles can adhere to coarser silt and sand particles
(Sun et al., 2006b). In the western CLP, the coarse silt (>40 μm) content is regarded as a
rough proxy for the winter monsoon strength (Wang et al., 2002). Therefore, to eliminate this
signal from the primary clay particles, the <2 μm/>40 μm ratio is proposed to evaluate
pedogenic intensity. Furthermore, the similarity of the variations between the <2 μm/>40 μm
ratio and $\chi_{pedo}$ confirms that both proxies are sensitive to pedogenic intensity in the XSZ red
clay.
**5.2 Time domain and frequency domain analysis of the carbonate content and $\chi_{pedo}$**
Power spectral analyses of carbonate content and $\chi_{pedo}$ show different dominant cycles
(Fig. 5). In detail, $\chi_{pedo}$ is concentrated in the eccentricity (100 ky), obliquity (41 ky) and
precession (21 ky) bands and another periodicities (71 ky and 27 ky) are also evident. In
contrast, the carbonate signal is concentrated in the precession (21 ky) and obliquity (41 ky)
bands, but it also exhibits even more prominent periodicities at 56 ky and 30 ky. Furthermore,



Morlet wavelet transform analysis of both carbonate content and $\chi_{pedo}$ show that the orbital
signal increases since 4.8 Ma (Fig. 5 d).
As for the non-orbital cycles, King (1996) proposed that these may possibly originate
from harmonics or interactions of the orbital cycles, while Lu (2004) ascribed them to the
unstable dust deposition processes followed by varying pedogenesis in palaeosol units. Here
we speculate that they may be caused by the low deposition rate, which potentially resulted in
the incomplete preservation of the paleoclimatic signal, especially for short cycles of
precipitation change. Thus, the incomplete nature of the red clay time series may be
responsible for the presence of spurious cycles. In addition, the carbonate content at various
depths is affected by leaching which means that the record integrates soil polygenetic
processes, thus obscuring orbital forcing trends related to precipitation amount. Low
deposition rates, compaction and leaching processes would obscure the orbital cycles, and
spectral peaks that do not correspond to orbital cycles may reflect these processes.
To investigate the post-6.7 Ma evolution of the climate signals in the XSZ section in the
frequency domain, we filtered the carbonate content and $\chi_{pedo}$ time series at the 100, 41, and
21-kyr periods, using Gaussian band filters centered at frequencies of 0.01, 0.02439, and
0.04762, respectively, and compared them with the equivalent filtered components of the
stacked deep-sea benthic foraminiferal oxygen isotope record. Our results show that the
fluctuations of the three filtered components of both two proxies change rapidly from very
low amplitude from 6.7-4.8 Ma to a much larger amplitude from 4.8-4.1 Ma (Fig. 5). The
enhanced orbital-scale variability of the two proxies from 4.8-4.1 Ma implies an increased
seasonality and wet-dry contrasts. This shift is not observed in the earth orbital parameters





but is observed in the filtered 41-kyr component of the stacked deep-sea benthic foraminiferal
oxygen isotope record ($\delta^{18}O$). This means that the increased contrast in wet-dry oscillations at
the XSZ site was not driven directly by changes in solar radiation intensity but rather was
linked with changes in ice volume or global temperature.
**5.3 Late Miocene-Pliocene climate history revealed by the Xiaoshuizi red clay**
**5.3.1 Multiporxy evidence for the dry climate during the interval of 6.7-4.8 Ma**
Based on the previous mentioned proxies of pedogenesis and chemical weathering, we
reconstruct the late Miocene and early Pliocene climatic history of the Xiaoshuizi peneplain,
NE Tibetan Plateau. As shown in Figure 6, we observe that a significant change recorded by
the most of the multiproxy (carbonate, Rb/Sr, $K_2O/Al_2O_3$, $\chi_{pedo}$) occurred near 4.8-4.7 Ma,
and therefore the climatic record was generally divided into two intervals. During interval I
(6.7-4.8 Ma), the relatively high carbonate values with minor fluctuations indicate that the
climate was dry and low Rb/Sr, $K_2O/Al_2O_3$ and $K_2O/Na_2O$ ratios support the weak chemical
weathering. Importantly, both the Rb/Sr and $K_2O/Na_2O$ ratios show opposite trends with
carbonate content, meaning that low effective precipitation resulted in weak chemical
weathering intensity. Furthermore, the pedogenic proxies (<2 μm/>40 μm ratio, $\chi_{pedo}$ and $\chi_{lf}$),
which characterised by low values with minor fluctuations, generally supports the weak
pedogenesis under the arid climate. Thus, during this interval the Xiaoshuizi climate was
relative arid, which characterized by weak chemical weathering and pedogenesis intensity.
However, subtle differences exist when these proxies detailed climate changes especially
when climate is relative wet. It is evident that the carbonate content decreases with increased
variation amplitude after 5.5 Ma, which is consistent with the cycles of carbonate nodules



within paleososol horizons observed in the field (Li et al., 2017). It may be increased
precipitation which induced eluviation-redeposition of carbonate since 5.5 Ma. However,
from pedogenesis indicies we observe that the general arid climate was interrupted by two
enhanced pedogenesis events (occurred at 5.85-5.7 Ma and 5.5-5.35 Ma, respectively). The
subtle differences may result from different sensitivity of magnetic susceptibility and
carbonate content to precipitation variability when precipitation is low (Sun et al., 2010). In
addition, a record of mollusks from the western Liupanshan showed cold-aridiphilous species
dominating which also document the cold and dry climate condition on the western CLP
during late Miocene (Fig . 7 g ).
During this interval, pollen, mollusk and magnetic records from the central and eastern
CLP also indicate generally dry and cold climatic conditions (Wang et al., 2006; Wu et al.,
2006; Nie et al., 2014). However, the obvious difference is that the Xiaoshuizi arid climate is
relative stable, while the climate of central and eastern CLP was interrupted by several
obvious humid stages. For instance, two humid stages (6.2-5.8 Ma and 5.4-4.9 Ma) are
recorded by the magnetic susceptibility of red clay in the hinterland of the CLP, but are not
recorded by the Xiaoshuizi magnetic susceptibility (Fig. 7). It is worth noting that 41-kyr
filtered component of thermo-humidiphilous species from the Dongwan was damped in late
Miocene (Li et al., 2008). Similarly, the amplitude of the orbital periodicities, filtered from
the XSZ carbonate content and $\chi_{pedo}$, are obviously damped from 6.7-4.8 Ma. However, the
three periodicities in Summer Monsoon Index from the central CLP show no obvious
difference between the late Miocene and Pliocene, but only a slight reduction in variability
after 4.2 Ma (Sun et al., 2010). Therefore, we agree that a dry climate prevailed on the CLP



during the interval of 6.7-5.2 Ma. The only difference is that the climate in the CLP
hinterland fluctuated more significantly than that of the Xiaoshuizi red clay.

The particularly damped response of the western CLP wet-dry oscillations to obliquity

forcing may indicate the palaeo-ASM had a negligible influence on the western CLP. It is
widely known that the summer monsoon intensity decreased from southeast to northwest
across the CLP. A regional climate model experiment demonstrated that  the modern Asian
summer monsoon was not fully established in the late Miocene and had only a small impact
on the northern China (Tang et al., 2011). The weak palaeo-ASM intensity from 7.0-4.8 Ma
has been revealed by hematite/goethite and smectite/kaolinite ratios at ODP Site 1148 from
the South China Sea (SCS) (Fig 7 i and j). Therefore, we deduce that the Asian monsoon
was weak and put a small impact on the Xiaoshuizi climate. In addition, during late Miocene,
the TP was not intensively uplifted and thus it could not block the westerlies completely (Li
et al., 2015). Previous studies suggested that the red clay may have been transported by both
low-level northerly winds and upper-level westerlies (Sun et al., 2004; Vandenberghe et al.,
2004). This means the impact of the westerly circulation on the study region cannot be
ignored. Notably, pedogenesis proxies roughly parallel to the stacked deep- sea benthic
foraminiferal oxygen isotope curve (Fig. 6). It indicates when global temperature was low,
pedogenesis intensity increased. It is unreasonable if the precipitation was dominated by the
palaeo-ASM. Thus, we speculate from 6.7 to 4.8 Ma, the precipitation transported by the
palaeo- ASM was limited and the westerly circulation probably dominated the climate of our
study region.

The simultaneous reduction in amplitude of the 41-kyr filtered components from the



western CLP and the deep sea $\delta^{18}O$ record from 6.7-4.8 Ma likely indicates that the dry
climate was related to changes in global temperature and ice volume. Look around the globe,
a cooling climate would be witnessed in late Miocene. $\delta^{18}O$ records from DSDP and ODP
sites show an increase of ~1.0‰ during the late Miocene which resulted from the increased
ice volume and the associated decrease in global temperature (Zachos et al., 2001). Records
from high latitude regions of the northern Hemisphere show continuously decreasing
temperatures and increasing ice volume during the late Miocene (Jansen and Sjøholm, 1991;
Mudieand Helgason, 1983; Haug et al., 2005). In the Quaternary, the dry climate prevailed
during glacial periods when global average temperature (especially in summer) was low.
Cool summers would have resulted in a small land-sea thermal contrast which in turn
weakened the palaeo-ASM in the late Miocene. Furthermore, the increased ice volume in the
Northern Hemisphere resulted in an increased meridional temperature gradient, thus
strengthening the westerlies and driving them southward. This would have prevented the
northwestward penetration of the Asian Summer Monsoon, which was also proposed as the
driving mechanism for a weak EASM in northern China during glacial periods (Sun et al.,
2015). Thus, the southward shift of the westerlies had a significant impact on the XSZ region.
Global cooling and the growth of polar ice-sheets reduced the amount of atmospheric water
vapor; thus, relatively little moisture was carried by the westerlies, producing a dry and stable
climate in the XSZ region. In conclusion, global cooling and increasing ice volume in the
Northern Hemisphere contributed to dry climatic conditions in the study region.
**5.3.2 Humid climate with enhanced fluctuations during the interval of 4.8-3.6 Ma**
During interval Ⅱ (4.8-3.6 Ma), the available proxy evidence indicates that the



Xiaoshuizi climate turns into humid condition from previous arid climate. The carbonate
content was low on average but with large fluctuations, indicating that the climate was
generally humid with increased dry-wet oscillations, especially during the interval of 4.8-3.9
Ma. Several obvious eluvial-illuvial cycles are observed from 4.8 to 3.9 Ma. The carbonate
content in the eluvial horizons was less than 10%, whereas in illuvial horizons it exceeded 30%
(Fig. 6). The emergence of high frequency cycles of carbonate eluviation-redeposition
indicates that seasonal precipitation was increased during this interval. Furthermore, the
variations of Rb/Sr and $K_2O/Na_2O$ ratios are very similar to those of carbonate content, which
suggests that weathering intensity was related to precipitation amount. Generally, high <2 μm
/ >40 μm ratio, $\chi_{pedo}$ and $\chi_{lf}$ correspond to large contrasts in carbonate content between eluvial
and illuvial horizons; thus, increased precipitation had a significant influence on enhanced
pedogenic intensity. From 4.8 to 3.9 Ma, high precipitation persisted and the weathering and
pedogenesis intensity were strong. The $K_2O/Al_2O_3$ ratio also increased rapidly at about 4.8-
4.7 Ma and maintained relatively high values after 4.7 Ma. This may indicate that the overall
weathering intensity was sufficient to produce secondary clays, resulting in a spike in $K_2O$
concentration. From 4.60-4.25 Ma, pedogenesis and weathering intensity reach the maximum,
as was precipitation intensity, which was manifested by enhanced eluviation and carbonate
accumulation. From 3.9 to 3.6Ma, precipitation decreased, and then weathering and
pedogensis intensity weakened, which may indicate that the Xiaoshuizi climate is generally
humid toward arid direction. Consisting with XSZ records, Dongwan mollusk records also
indicate the warm and wet conditions on the western CLP during early Pliocene (Fig. 7 h).

Palynological and terrestrial mollusk records from the central CLP also indicate





relatively humid conditions during early Pliocene (Wang et al., 2006;Wu et al., 2006). The
magnetic susceptibility records from the CLP hinterland exhibit similar characteristics to the
XSZ records that both the magnitude and variability of magnetic susceptibility are large from
4.8-3.6 Ma. From 4.1-3.9 Ma, the enhancement of magnetic susceptibility indicates that
humid climatic conditions prevailed across the entire CLP (Fig. 7). Obviously, when
precipitation amount peaked from 4.6-4.25 Ma in the XSZ section, the $\chi_{lf}$ values at Xifeng,
Lingtai and Chaona were low. However, the Lingtai $Fe_2O_3$ ratio record showed an
extraordinary high value corresponding to abundant clay coating over the interval of about
4.8-4.1 Ma and this interval was interpreted as the strongest ASM intensity in the CLP
since 7.0 Ma (Ding et al., 2001). In addition, the relative intensity of pedogenic alteration of
the grain-size distribution was the strongest during the interval from 4.8-4.2 Ma in the Lingtai
section (Sun et al., 2006c). Pollen assemblages at Chaona indicate a considerably warmer and
more humid climate from 4.61-4.07 Ma (Ma et al., 2005). These evidences indicate climate
from 4.6-4.25 Ma is warm and wet in the central CLP. Gleying has been implicated in
reducing the value of magnetic susceptibility as a record of precipitation during this period
(Ding et al., 2001). When soil moisture regularly exceeds the critical value, dissolution of
ferrimagnetic minerals occurs and the susceptibility signal is negatively correlated with
pedogenesis (Liu et al., 2003). This by itself indicates that precipitation was likely to have
been very high during this interval.

In summary, a wet climate prevailed across the CLP in early Pliocene. At the same time,

hematite/goethite ratio from the SCS also shows enhanced precipitation amount and
Smectite/Kaolinite ratio there shows increased seasonality at about 4.8Ma (Fig. 7 i and j),



which indicate the enhancement of palaeo-ASM (Clift et al., 2006, 2014). Thus, we regard
climate change of Xiaoshuizi as the result of expansion of the Palaeo-ASM expressed in its
intensity and reach during this interval.

The remarkably increased amplitude of the 41-kyr filtered components from XSZ and

the deep sea $\delta^{18}O$ record at about 4.8 Ma indicates the expansion of palaeo-ASM may be
related to changes in global temperature and ice volume. Furthermore, decreasing input of ice
raft debris into subarctic northwest Pacific was synchronous with the expansion of palaeo-
ASM during early Pliocene (Fig. 6). In addition, from 4.8-4.7 Ma and 4.6-4.25 Ma, the high
values of the three pedogenic indices at the XSZ section indicate that strong pedogenic
intensity corresponded with high SSTs in the eastern equatorial Pacific (EEP). These
coincides imply that phases of enhanced precipitation may be correlated with changes in SST
and ice volume (or temperature) at northern high latitudes.

**5.4 Possible mechanism for the paleo-ASM expansion during early Pliocene**

Ding (2001) proposed that uplift of the TP to a critical elevation resulted in an enhanced

summer monsoon system during 4.8-4.1 Ma. The TP uplift was shown to have had profound
effects on the ASM initiation, having strengthened ASM intensity and changed the shape of
the precipitation band in East Asia (Li et al., 1991, 2014; An et al., 2001). A more detailed
modeling study demonstrated that the uplift of the northern TP mainly resulted in an
intensified summer monsoon and increased precipitation in northeast Asia (Zhang et al.,
2012). From 8.26-4.96 Ma, massive deltaic conglomerates were widely deposited and the
sediment deposition rate increased, indicating the uplift of the Qilian Mountains (Song et al.,
2001). At the same time, the Laji Mountains underwent a pronounced uplift by thrusting at





about 8 Ma, which resulted in the current basin-range pattern (Li et al., 1991; Fang et al.,
2005a; Zheng et al., 2000). However, geological and palaeontological records indicate that
the uplift of the eastern and northern margins of the TP was very small from late Miocene to
middle Pliocene (Li et al., 1991, 2015; Zheng et al., 2000; Fang et al., 2005a, 2005b). So we
speculate the TP uplift may be not the major contribution to the expansion of palaeo-ASM
occurred at ~4.8 Ma.

As mentioned above, the extremely wet climate across the CLP was synchronous with

the gradual closure of the Panama Seaway, which led to a larger reorganization of the global
thermohaline circulation pattern. Nie (2014) proposed that the freshening of the Eastern
Equatorial and North Pacific surface water, resulting from the closure of the Panama Seaway
since 4.8 Ma (Haug et al., 2001), led to sea ice formation in the North Pacific Ocean, which
enhanced the high-pressure cell over the Pacific and increased the strength of southerly and
southeasterly winds. However, there was a warming trend in the Northern Hemisphere at 4.6
Ma (Haug et al., 2005; Lawrence et al., 2006). The gradual closure of the Panama Seaway
resulted in the reorganization of surface currents in the Atlantic Ocean. In particular, the Gulf
Stream was enhanced and began to transport warm surface waters to high northern latitudes,
thus strengthening the Atlantic meridional overturning circulation and warming the Arctic
(Haug et al., 1998, 2005). This in turn resulted in higher global atmospheric water vapor
levels which promoted warm moist conditions during the Pliocene (Abbot and Tziperman,
2008; Dowsett et al., 2010). Three independent proxies from an early Pliocene peat deposit in
the Canadian High indicate that Arctic temperatures were 19 ℃ warmer during the early
Pliocene than at present (Ballantyne et al., 2010). Therefore, even freshening of the Pacific



led to sea ice formation in the North Pacific Ocean. However, this process would be delayed
(occurring during 3.2-2.7 Ma) and the extent of the sea ice in the early Pliocene was thus very
limited. In contrast, the warming of the northern high latitude region led to increases in
summer temperature in the mid-latitudes of Eurasia. On the other hand, equatorial SSTs
remained stable or cooled slightly (Brierley et al., 2009; Fedorov et al., 2013). This amplified
the land-ocean thermal contrast and was essential for enhancing the palaeo-ASM.
Furthermore, external heating derived from reduced planetary albedo also enhanced the
thermal contrast between the Pacific and Eurasian regions (Dowsett et al., 2010). On the
other hand, the unusually warm Arctic and small meridional heat gradient in the Northern
Hemisphere pushed the Intertropical Convergence Zone northward. This weakened the
westerly circulation and thus facilitated the northwestward expansion of the ASM.
Fig 6 indicated high values of pedogenic indices at the XSZ section correspond with
high SSTs in the EEP. It seems to be discrepancy with the modern ENSO cases (when the
EEP temperature is high, the precipitation amount of the western CLP is low). The
discrepancy may indicate sea-air interaction during early Pliocene is different from today.
From 4.8 to 4.0 Ma, the thermahaline circulation was reorganizing and creating a
precondition for the development of the modern equatorial Pacific cold tongue (Chaisson et
al., 2000). Some crucial changes linked with summer monsoon occurred. We noticed a vast
expansion of the western Pacific warm pool into subtropical regions occurred in early
Pliocene (Brierley et al., 2009; Fedorov et al., 2013). Temperatures at the edge of the warm
pool show a warming trend of ~2℃ from the latest Miocene to the early Pliocene (Karas et
al., 2011). The thermal state of the WEP warm pool significantly enhanced the summer



monsoon and its northward extension. In modern times, when the north of western pacific
warm pool was warm, the convection over and around the Philippine was enhanced.
Subsequently, the northern extent of the western Pacific subtropical high shifted northwards
from the Yangtze River valley to the Yellow River valley and moisture was introduced across
the entire CLP (Wang et al., 2000; Huang et al., 2003). Whether it is also the case for the
early Pliocene or not needs further researching. However, warming and freshening seawater
of subtropic Pacific would have been more readily evaporated which would have provided
enhanced moisture for the palaeo-ASM leading to increased rainfall across the CLP.
Thus, we deduce it may be warming of high northern latitudes, accompanied by the vast
poleward expansion of the tropical warm pool into the subtropical regions and freshening of
water in the subtropical Pacific facilitated the expansion of the palaeo-ASM during early
Pliocene.
**6. Conclusions**
Continuous late Miocene-Pliocene red clay preserved on the representative planation
surface in NE Tibetan Plateau provides particular opportunity to discuss the Asian monsoon
history. Multi-proxy records from the XSZ planation surface in the western CLP, together
with other palaeoclimatic records from the CLP, reveal two intervals of major climatic change
from 6.7 to 3.6 Ma. During the first interval (6.7-4.8 Ma), the XSZ records indicate that both
the amount and variability of precipitation were small; however, they were much greater in
the hinterland of the CLP. Thus, the palaeo-ASM had little influence on the climate of the
western CLP during this interval. During the second interval (4.8-3.6 Ma), the XSZ records
indicate that both the amount and variability of precipitation were large. From 4.8 and 3.6 Ma,



the climate was characterized by abrupt increases in the seasonality of precipitation, which
attests to a major northwestward extension and enhancement of the summer monsoon.
Obviously, multiple paleoclimatic proxies show that the strongest summer monsoon occurred
during the interval of 4.6-4.25 Ma. The expansion of palaeo-ASM may have been caused by
warming of the Arctic, the vast poleward expansion of the tropical warm pool into the
subtropical regions and freshening of water in the subtropical Pacific in response to the
closure of the Panamanian Seaway during early Pliocene.

**Author contribution:** Tingjiang Peng and Jijun Li supported fund and edited the article.
Zhenhua Ma provided age frame and participated in the most of field work. Meng Li,
Zhantao Feng, Benhong Guo, Xiyan Ye and Hao Yu participated in investigation and
experiment works. Chunhui Song and Zhengchuang guided and supervised the field work.
**Competing interests:** The authors declare that they have no conflict of interest.

**Acknowledgements**
We thank Ai Song for the field drilling and Fengxia Yu for her early experimental work.
This work was supported by National Natural Science Foundation of China (41330745,
41401214) and Key Laboratory of Continental Collision and Plateau Uplift, Institute of
Tibetan Plateau Research (LCP201602).

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



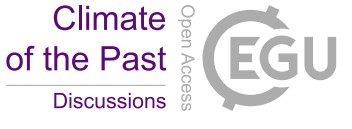

Figures and tables

Fig. 1. The location of the study area and atmospheric circulation patterns. (a) 850 mb vector wind averaged from June to August for 1982-2012 based on NOAA Earth System Research Laboratory reanalysis data (Compo et al., 2013). (b) Lithology and magnetostratigraphy of the XSZ drill core (Li et al., 2017). (c) The Chinese Loess Plateau with locations of the studied Xiaoshuizi site and other sections mentioned in the text.



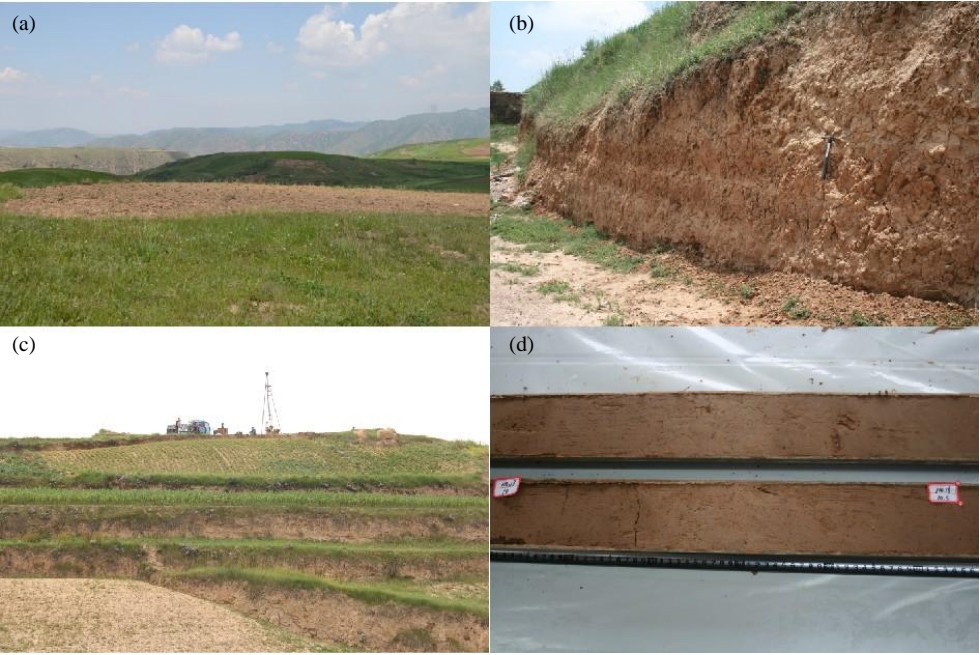

Fig. 2. Photos of the XSZ planation surface and the red clay. (a) XSZ planation surface.

(b) Red clay outcrop, XSZ. (c) Position of the XSZ drilling hole. (d) The XSZ drill core.



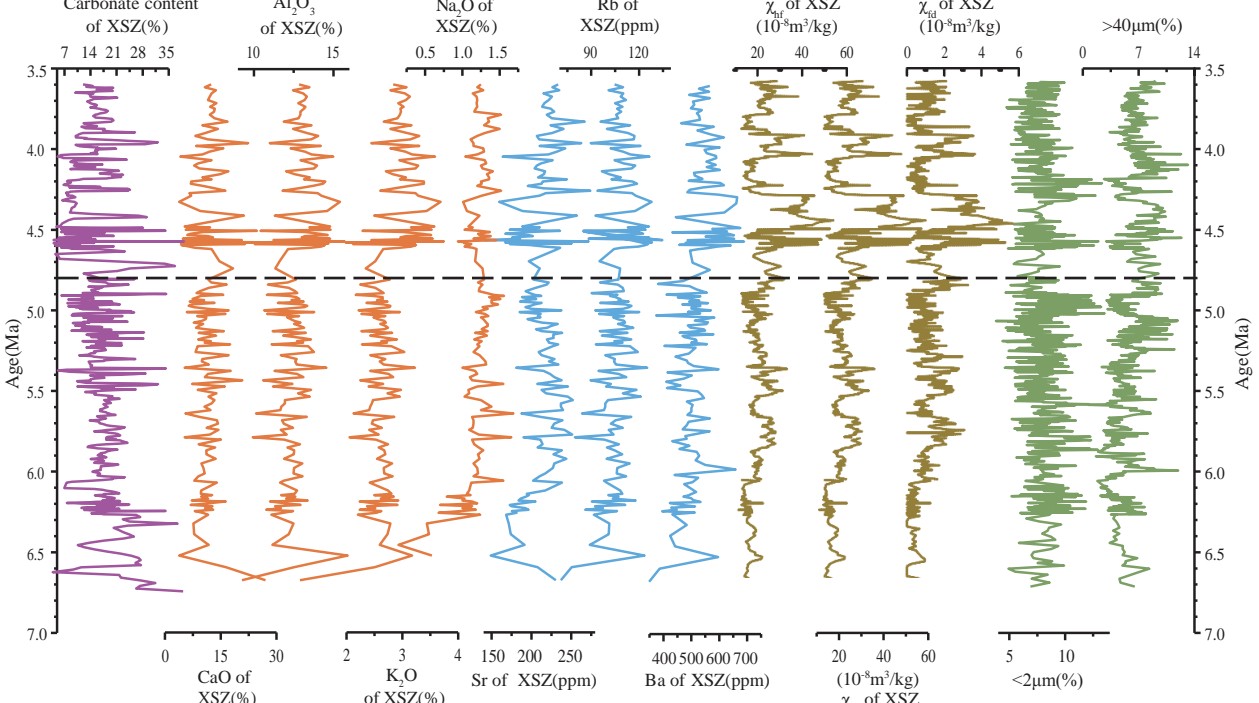

Fig. 3. Variations in carbonate content, major element concentration, minor element concentration, magnetic

susceptibility and grain size from the XSZ red clay section, spanning 6.7-3.6 Ma.





Fig. 4. (a) Scatter plots of $\chi_{lf}$ versus $\chi_{fd}$. (b) Separation of $\chi_{pedo}$ and $\chi_0$. (c) Scatter plot of $\chi_{lf}$ versus $\chi_{pedo}$ during

4.8-3.6 Ma. (d) Scatter plot of $\chi_{lf}$ versus $\chi_{pedo}$ during 6.7-4.8 Ma. (e) Scatterplot of Sr versus CaCO$_3$. (f) Scatter

plot of Sr versus CaO*. Solid squares and triangles are the average values during 4.8-3.6 Ma and 6.7-4.8 Ma,

respectively. $\chi_{pedo}$ is the magnetic susceptibility of pedogenic origin and $\chi_0$ is the magnetic susceptibility of the

detrital material.





Fig. 5. Spectrum analysis of the red clay. (a) $\chi_{pedo}$ and (b) carbonate content(blue) on original paleomagnetism

chronology. Dashed lines are 90% confidence limit lines. (c) Comparison of orbital parameters (i.e., eccentricity,

obliquity and precession,Laskar et al., 2004) with filtered components of the carbonate content, $\chi_{pedo}$ and $\delta^{18}O$

records (Zachos et al.,2001) at the 21-kyr, 41-kyr, and 100-kyr bands. Yellow shading denote the largest amplitude

of filtered components of carbonate and $\chi_{pedo}$ at the three orbital bands. Dashed lines indicate a large shift in the

East Asian monsoon circulation occurred around 4.8 Ma. (d) Results of the wavelet transform of $\chi_{pedo}$ and carbonate

content time series.

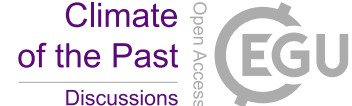

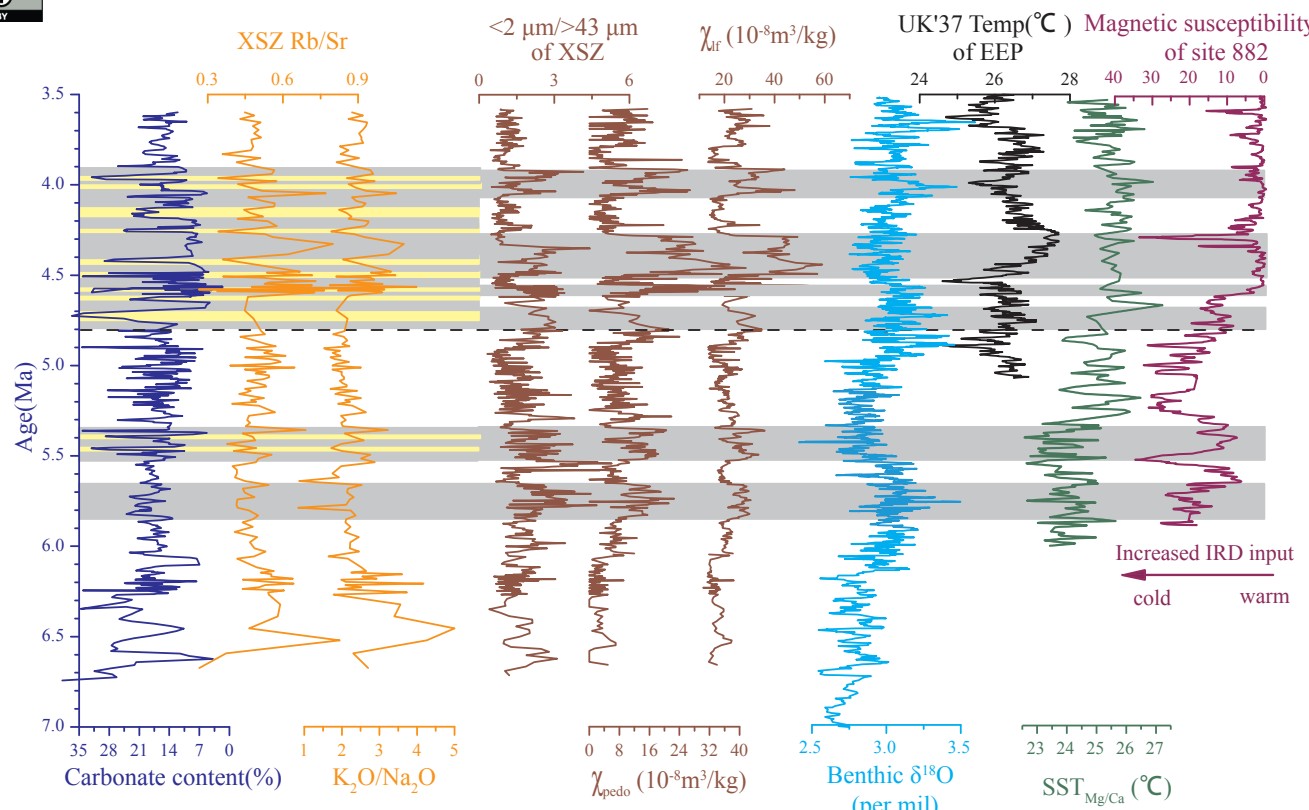

Fig. 6. Temporal evolution of the palaeo-ASM. The dark blue line represents changes in effective

precipitation at XSZ, the orange line represents changes in chemical weathering intensity, and the

brown lines represent changes in pedogenic intensity. The blue line is the stacked deep-sea benthic

foraminiferal oxygen isotope curve compiled from data from DSDP and ODP sites (Zachos et al.,

2001). The black line is a reconstruction of sea surface tempeature in the eastern equatorial Pacific

(EEP) from ODP Site 846 (Lawrence et al., 2006). Green line is a reconstruction temperature at the

edge of warm pool from southwest Pacific Ocean Site 590B (Karas et al., 2011). Purple line is

magnetic susceptibility from ODP Site 882 (Haug et al., 2005). Gray shading shows intervals of

strong palaeo-ASM and the light-yellow shading shows intervals of carbonate accumulation.

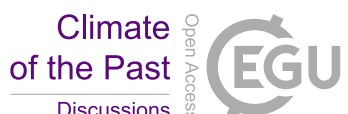

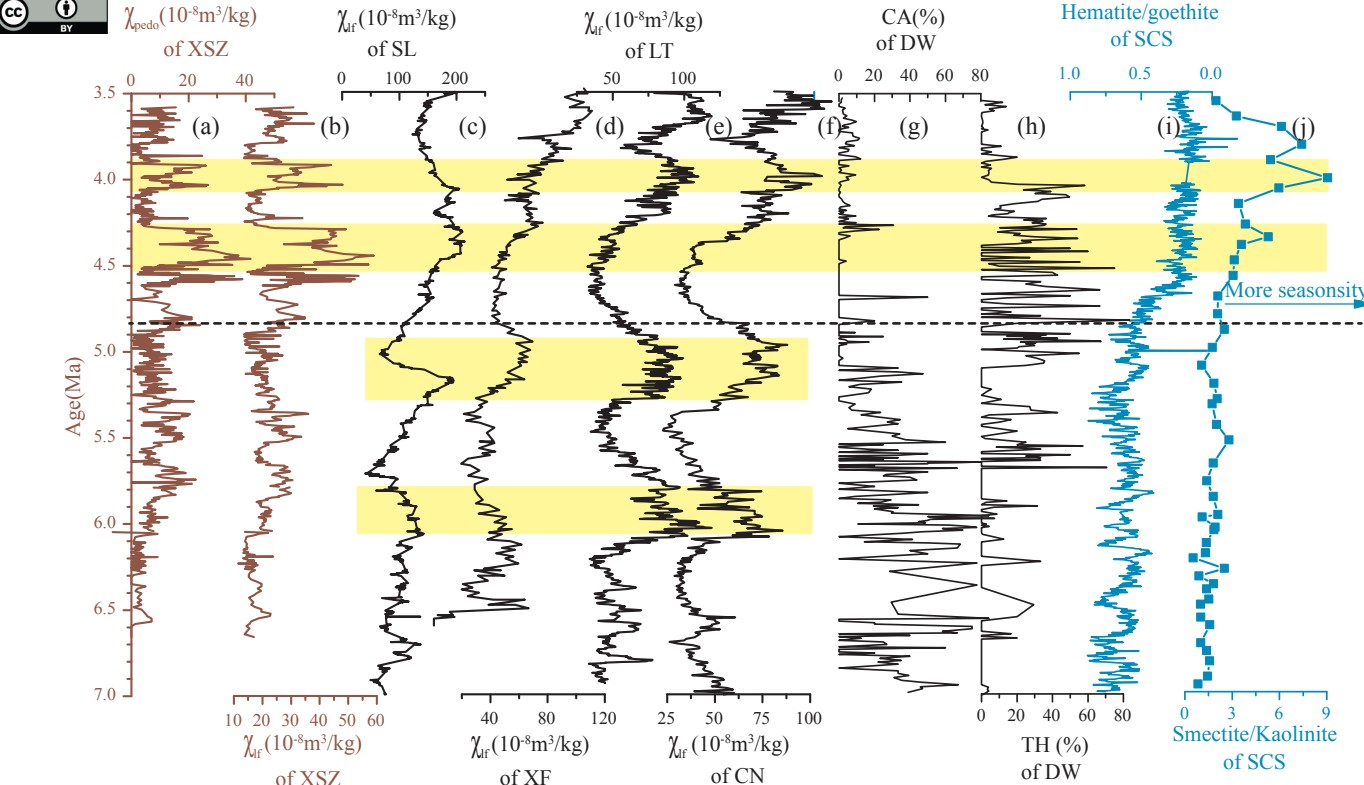

Fig. 7. Comparison of late Miocene-Pliocene paleoclimatic records from Asia. (a-b) $\chi_{pedo}$ and $\chi_{lf}$ from the

XSZ section. (c-f) $\chi_{lf}$ record from Shilou (Ao et al., 2016), Xifeng (Guo et al., 2001), Lingtai (Sun et al.,

2010) and Chaona (Song et al., 2007). (g-h) Percentages of cold-aridiphilous (CA) mollusk group and

thermo-humidiphilous (TH) mollusk group from Donwan(Li et al., 2008), (i) Hematite/goethite ratio

from the South China Sea (Clift, 2006). (j) Smectite/Kaolinite ratio from the South China Sea (Wan et al.,

2010; Clift et al., 2014).



**Table 1.** major element compositions of XSZ red clay.

| Content | SiO$_2$(%) | Al$_2$O$_3$(%) | Fe$_2$O$_3$(%) | CaO(%) | MgO(%) |
|---|---|---|---|---|---|
| Average | 49.16 | 12.61 | 5.38 | 11.36 | 2.76 |
| 6.7-4.8Ma | 48.85 | 12.22 | 5.18 | 11.20 | 3.06 |
| 4.8-3.6Ma | 49.50 | 13.22 | 5.69 | 11.60 | 2.30 |
| Content | K$_2$O(%) | Na$_2$O(%) | Rb(ppm) | Sr(ppm) | Ba(ppm) |
| Average | 2.76 | 1.22 | 106.2 | 212.8 | 519.0 |
| 6.7-4.8Ma | 2.59 | 1.20 | 103.9 | 211.7 | 494.3 |
| 4.8-3.6Ma | 3.03 | 1.23 | 109.9 | 214.6 | 558.0 |