# Peer review of "Late Miocene-Pliocene climate evolution recorded by the red clay covered on the Xiaoshuizi planation surface, NE Tibetan Plateau"

_Climate of the Past, 2018_

## Referee Comment (RC1) · Anonymous Referee #1 · 6 Aug 2018

The manuscript from Li, et al. presents new geochemical records and environmental reconstructions from the Xiaoshuizi section on the western Chinese Loess Plateau. These records are interesting because they help to constrain the westward reach of the East Asian Monsoon in the Late Miocene and Pliocene. When combined with records from the central and eastern plateau, the new Xiaoshuizi shows that significant increases in moisture availability did not occur until after 4.8 Ma, and coincided with the wettest periods observed on the eastern Loess Plateau. Generally, I found the manuscript to be interesting and well organized and have a few suggestions detailed below.

[Figure]

I think the manuscript needs to present the geochemistry data versus stratigraphic depth, in addition to just age. There also needs to be more discussion on the relationship between sedimentation rate and pedogenesis. For example, it would be helpful if Figure 3 was plotted vs. depth and there was also a column that plots sedimentation rate, and the presence of nodule horizons. This is important because the interval between 4.5 and 4.3 Ma, for example, shows a strong increase in magnetic evidence for pedogenesis and also coincides with a noticeable drop in deposition rate. Therefore, it needs to be discussed if this increase in pedogenesis was driven solely by wetter conditions, or was there also more time for soil formation and leaching of Ca. I do think more stratigraphic context will help some of the arguments presented in the text. For example, upon my initial reading of the text and figure, the division into the 2 primary intervals placed at 4.8 Ma seemed somewhat arbitrary looking at figure 3 (i.e. why not 4.6 or 5.1). But it makes much more sense in terms of the large decline in sedimentation rate around ~4.8, which accompanied by the deposition of a carbonate nodule layer, and then the noted increase up-section in nodule horizons underlaying leached zones. Also with deposition of loess being connected to regional wind patterns, is it significant that there was a notable ~200 kyr drop in sedimentation rates before a shift to generally wetter/more seasonal conditions?

I am somewhat confused by the explanation of K/Al ratios as a weathering proxy (lines 238-245). With time, Al can mobilize and become depleted at the top of a paleosol and enriched down profile. And in certain situations, you might expect K to be enriched at the soil surface, due to its biological importance. So, within the same well developed soil, you might expect a higher K/Al ratio at the top, and a lower ratio deeper in the profile. This is never plotted, so it might be worth eliminating this text?

The various magnetic susceptibility terms are well described in the discussion, but I think it would help readers if at least some of this information was moved up to either the results or methods. This would help provide context to all of the values presented in the results.

Minor suggestions:

Line 57: suggest "occurring" or "underway" instead of "ongoing"

Line 76: suggest "supplied" instead of "prepared"

Lines 75-88: I'm guessing the sentence beginning with "Make clear. . ." on line 78 was accidentally left in as a comment, which I still think needs to be addressed. I think I understand what the authors are going for within the paragraph, but I think the logic can be expressed more clearly. The strength/onset of the Asian monsoon is linked to these globally significant events (Tibetan uplift, northern hemisphere ice, etc). Therefore, by constraining paleoclimate across the Chinese Loess Plateau not only does this improve our understanding of regional climate, but it can also provide insight about the paleo-monsoon, and therefore changes in the global climate system during the Pliocene.

Line 96: suggest removing "condition" and changing "aridification process" to "regional aridification"

Line 104: change "to be" to "that", and I think it would be helpful for the future readers not just to say "gleying", but instead state briefly what that means (waterlogging, and iron reduction) and why it matters for the magnetic susceptibility record.

Line 105-106: This sentence does not make sense. Are you trying to say that climate in this region is influenced by the strength of both the westerlies and the monsoon, and that those two factors may not be directly related?

Lines 114-115: What makes the XSZ red clay different geomorphologically?

Line 118: suggest "are" instead of "have been"

Line 121: This sentence is slightly off.

Line 133: suggest "reconstruct and discuss" instead of "discuss'

Line 133-134: not sure exactly what is meant here. Is the XSZ core characterized by

more continuous deposition and records a longer time interval than the Shangyantan core?

Line 136: capitalize China

Lines 137-138: Not sure what is mean by the sentence beginning with "The East Asian Monsoon." Are you trying to explain how these two factors together control climate at the study site. This could be elaborated.

Line 144: Where in the section is the increase in gravel? From the strat column it looks like it is at the base. Say this in-text.

Lines 145-147: Clarify if most carbonate horizon are overlain by a brownish red-layer, or if the carbonate zone in its entirety underlies a larger brownish-red layer. Lines 148-150: It's not clear as written if carbonized root channels have more abundant Fe-Mn staining.

Line 168: Is all of the remaining Ca in silicate minerals? Won't a lot of it be loosely bound to clay minerals in the soils? Also, the correction for Phosphorous also needs to be explained. I'm guessing you are assuming some component of Ca-bearing phosphate minerals, but what is the basis for this assumption.

Line 199: What do you mean by durations? Are you saying there are some thicker intervals of high magnetic susceptibility?

Line 256: space between "susceptibility" and "of"

Line 257: suggest removing "two"

Line 314: Spelling of "Multiproxy"

Line 317-318: suggest "a significant change is recorded by most of the proxies that occurred"

Line 318: K/Al is not plotted, but K/Na is plotted. Based on the comment above, I think

this is probably a better choice.

Line 327: suggest "relatively" instead of "relative" and "and" instead of "which"

Line 328: Not sure what this sentence is trying to say.

Line 329: I suggest clarifying the beginning of this sentence to say something along the lines of "Carbonate content becomes more variable after 5.5 Ma, which is…"

Line 333: spelling of "indices"

Line 345: suggest "central and eastern" instead of "hinterland of the"

Line 377: suggest rewording the sentence beginning with: "Look around the globe,…"

Line 415: I'm not sure what "humid toward arid direction" means

Line 521: suggest "provides the opportunity constrain and discuss…"

Line 526: again suggest "central and eastern" instead of "hinterland of the"

Line 531: suggest removing "obviously"

Figure 1: I think it would help if you put a larger non-circle shape on panel A corresponding to the study site. Then you can remove the Xiashuizi label, which slightly obscures the vector. Then, match this symbol on panel C You are missing the white reversals between C3n.1n, C3n.2n, and C3n.3n on the Polarity plot for the XSZ section. These were included in the age model presented in Li et al. (2017).What do the black bars on the lithology column represent.

Figure 2: I think it would help if the line thicknesses were slightly thinner.

[Figure]

---

## Referee Comment (RC2) · Anonymous Referee #2 · 6 Aug 2018

Li et al. analyze a red clay sequence from Northeastern Tibet that spans from 6.7 to 3.6 Ma for carbonate content, major and minor element concentration, magnetic susceptibility, and grain size. The authors suggest that the proxies show a locally dry, westerlies-dominated climate from 6.7 to 4.8 Ma followed by a wetter, monsoonal climate from 4.8-3.6 Ma. Li et al. propose that closure of the Isthmus of Panama might be responsible for this climate shift. Although the proxy records are a valuable addition to our understanding of climate change and variability, the manuscript has several major flaws that should prevent publication in its current form.

General Comments:

Spelling and Grammar:

I have not edited this manuscript for spelling and grammar. I strongly encourage the authors to seek assistance from a very proficient or native English speaker. Also, please review the manuscript for organizational mistakes (e.g. Figures 2 and 3 are not cited in the text; incorrect citations).

Statistics:

The authors need to provide more information about the magnetostratigraphic ages. What is the temporal resolution of the records? What are the temporal uncertainties? Can the records accurately resolve all the cycles you discuss (e.g. precession)? Does variable deposition rate impact the signals?

How did the authors decide that 4.8 Ma was the appropriate transition point? It seems arbitrary to me. I see no clear transition in Figure 3. Are the two periods (6.7-4.8 Ma and 4.8-3.6 Ma) statistically distinct?

I find the signal filtering in Figure 5 questionable. First, the authors filter the data at frequencies with insignificant power (e.g. the 100 kyr filtering of carbonate content). Further, the most significant signals exist at frequencies that are difficult to explain, which the authors dismiss, and many of the discussed signals are barely significant at 90% confidence. The wavelet plots highlight the limited signal strength. Even if the filtered signals are sound, the filtered signals changes do not well align with the benthic d18O record.

Interpretation:

The potential drivers of the climate signals are often overstated. Connections are made with limited support. Many of the mechanisms discussed are still debated, particularly the Isthmus of Panama hypothesis and timing of Tibetan Plateau uplift. At the least, the authors need to do a better job citing recent literature and discussing the remaining uncertainties. Also, many of the citations are not primary sources for the associated

statements.

Specific Comments:

Line 51: Earth's orbital went through many cycles over this period, so the "orbital configuration" statement does not make much sense.

Lines 51-53: These statements, such as "comparable temperatures in the tropic region", require citations.

Line 65: Please clarify the link between mean tropical Pacific east-west gradient and ENSO.

Line 68: The timing of uplift of the Tibetan Plateau is heavily debated. . .

Line 70: Lunt et al. (2008) is not a direct source for the closure of the Isthmus of Panama. More recent works debate the timing of closure (e.g. Bacon et al., 2015; O'Dea et al., 2016).

Line 72: This statement is also not well supported. For example, Lunt et al. (2008), who are cited earlier, found closure of the Panama seaway to have little influence on NH glaciation. In general, the authors need to update their citations and discuss the literature more thoroughly. These ideas are far from settled, yet they are presented as facts.

Line 80-82: Citation?

Line 85: "Arctic volume" means "Arctic ice volume"?

Lines 105-106: This statement does not make sense.

Lines 136-140: Sources for these data?

Line 154: This requires more detail.

Line 175-182: How did you decide on these intervals? Did you test that they are statistically distinct?

Line 224-226: How are you sure that it relates to monsoon strength? Could it be seasonal or evaporative changes?

Lines 235-237: Both statements are significant at 99% confidence?

Lines 282-286: Are you sure these signals are real? If so, how might you explain the cycles not related to orbital variability?

Lines 294-295: Doesn't this "incomplete nature of the red climate time series" impact all of the frequency analyses? How can you distinguish real and fake signals?

Line 302: I believe that a 23 kyr filter makes more sense for the climate response to orbital change.

Line 304: What record? Lisiecki and Raymo (2005)?

Line 306: I do not observe this in the filtered record...Is this change significant? How much do these filter components contribute to the complete signal?

Line 309-310: Where is this shown? The 41 kyr signal in the benthic records do not well align with the data.

Lines 317-319: I see no clear changes in the records. You need statistical support.

Line 361: ODP source?

Line 368: "...roughly parallel..." I do not see a correlation. Please quantify.

Line 384: This is possible but not necessarily the case.

Lines 392-393: Cooler air can hold less vapor, but this statement is an extreme simplification.

Lines 402-403: You record captures seasonal variability?

Lines 469-471: Citation?

Line 480: Why global moisture and not local moisture?

Lines 484-487: This does not make sense.

Lines 491-492: Are you talking about regional or global albedo?

Lines 492-495: Citation?

Lines 497-498: Could this discrepancy relate to differences between short term variability and the mean climate state?

Line 502: "We noticed"? You mean the authors of these other publications noticed?

Lines 502-506: How close are these events in time?

Figures:

Figure 1a: The winds do not look correct. Also, 850 hPa winds do not exist over the Plateau...

Figure 2 and Figure 3 are not cited in the text.

Figure 3: It is difficult to see how the axes align with the lines

Figure 5d: Do the black lines represent significance?

---

## Referee Comment (RC3) · Anonymous Referee #3 · 26 Aug 2018

Ms. Cp-2018-73: Late Miocene-Pliocene climate evolution recorded by the red clay covered on the Xiaoshuizi planation surface, NE Tibetan Plateau   By Li Xiaomiao et al.

Overview:

This study provides multiple environmental proxies, including carbonate content, element concentration, magnetic susceptibility and grain size, from the red clay deposits of the Xiaoshuizi section to study the late Miocene-Pliocene climate evolution of the western Chinese Loess Plateau. The authors identify two time intervals with different climate patterns: 1) 6.7-4.8 Ma minimal weathering and pedogenesis representing arid

condition, which is a result of weakened paleo-Asian summer monsoon, and intensified Westerly circulation; and 2) 4.8-3.6 Ma enhanced weathering and pedogenesis indicating humid climate. This transition from arid to humid climate is considered to indicate enhancement of the paleo-Asian summer monsoon, which is inferred as the combined effects of: increasing Arctic temperatures, expansion of the tropical warm pool into the subtropical region, and water freshening in the subtropical Pacific.

This study presents many proxy data and provides detailed discussion, which are likely of interests to researchers studying Neogene climate changes of East Asia, especially evolution of the East Asia summer monsoon. Thus, this paper should be published. However, I have some concerns and questions that need to be solved before the manuscript can be accepted. A moderate to major revision is recommended.

Major concerns:

1) The introduction part is not well written as there are many ambiguity and in-accuracy (see detailed comments below). This section needs significant reworking.

2) The authors seem to preferentially pick 4.8 Ma as the boundary between the two climate intervals. However, most of the proxies exhibited in Fig. 3 seem has a distinct change at 4.6 Ma, but not 4.8 Ma, e.g., Al2O3, K2O, as well as the three magnetic susceptibility plots. In addition, for the grain size and carbonate content plots, there is no apparent difference below and above 4.8 Ma.

3) This manuscript is generally good written in English, but additional efforts are required to polish the language.

Line-to-line comments:

L1-2: I found the title is kind of misleading. The authors emphasize the Tibetan Plateau as the location of their section. However, throughout the manuscript, the Xiaoshuizi section is compared with other sections on the Chinese Loess Plateau, and reflects nothing of the Tibetan Plateau evolution. So it would be more appropriate to emphasize

the location as the "western Chinese Loess Plateau".

L49: one of the most intensively studied intervals of what? Climate I assume?

L51: in line 41, the authors state closure of the Panamanian Seaway at 4.8 Ma, and it seems that the seaway closure has significant climate effects. Thus, it would be inappropriate to state here that the Zanclean is similar as present due to similar land-sea distribution.

L52-53: references for "comparable temperatures in the tropical region" need to be added.

L54: Zanclean is a period from cold to warm?

L66-67: wired transition from the previous sentence. Not consistent.

L68: This is at least not accurate, if not wrong. Numerous studies have demonstrated that surface uplift of the Tibetan Plateau is stepwise and spatially diachronous. See reviews of Tapponnier et al., 2001, Wang et al., 2014 and many others. The south-central parts of the Tibetan Plateau were uplifted much earlier than the Zanclean, e.g., Paleogene. In the northern Tibetan Plateau, although there might be tectonic deformation in the margins of the Plateau (Li et al., 2015), the major part of the northern Plateau is probably uplifted during the Miocene, as evidenced by numerous other evidence, see review of Yuan et al., 2012. While it's OK to stick on the authors' own preference, it's necessary to discuss/reflect other research progress.

L72: 3 Ma or 2.6 Ma? Be accurate.

L 75: first appearance of ASM in the main text, need to define first. In addition, for summer monsoons in Asia, there is the East Asia Summer Monsoon and the South Asia (India) Summer Monsoon. Which one do you mean? I assume East Asia Summer Monsoon?

L76-77: In the abstract, the authors consider the ASM as moisture carrying, but the

Westerlies as moisture lacking. So it's not appropriate to list them together. In addition, moisture transport is short-time climate condition, how could it cause long-tern glaciation?

L82: "warm and wet" climate yield "wet" climate? Definitely!

L84: a weakened summer monsoon of where? Globally or East Asia only?

L89-91: onset of interior Asian aridification since the late Miocene? This is totally unjustified. Numerous studies indicate much earlier onset of Asian interior aridification, e.g., since 22 Ma (Guo et al., 2002), or much earlier at Eocene-Oligocene transition (Dupont-Nivet et al., 2007), or late Eocene (Bosboom et al., 2014).

L103-105: is this phenomenon also observed in other studies?

L107-108: what inconsistent? Need to clarify. For the evidence listed above, it's necessary to point out which region is dominated by westerlies, which is dominated by ASM.

L110: why the western CLP is especially important? Need to give reasons here.

L126-127: rejuvenated at what time?

L175-182: what are the criteria to divide the carbonate content plot into 6.7-4.8 Ma and 4.8-3.6 Ma. I do not see apparent difference between these two subdivisions. For the 6.7-4.8 Ma interval, the carbonate content is 3.8-39.2The 6.0-5.5 interval, with much smaller amplitude of fluctuation, seems to be more different from the other time intervals.

L185-187: looking at Fig. 3, it's pretty hard to determine whether two plots are of similar trend, or opposite trend. I would suggest to provide statistical evaluation to help readers understand the similarity between plots.

L188: provide the ranges of Al2O3 and K2O for the two time intervals.

L189: why choose 4.8 Ma as the boundary? The values between 4.8-4.6 seem be more similar as the 6.7-4.8 Ma interval.

L193: similar question, why group values between 6.9-4.8 together, but not include values between 4.8-4.6, which exhibit more similarities as the 6.9-4.8 Ma interval, which are of lower values and smaller amplitude of variation.

L204: there is no difference between these two intervals.

L206: in Fig. 3, it shows >40 m.

L251: I have a question here, maybe very basic in your discipline. If one wants to use K2O/Na2O values to determine the intensity of chemical weathering, a pre-assumption is that before weathering, all the samples have similar K2O/Na2O values. Right? How about if the original K2O/Na2O values are different? This question might also exist for other chemical proxies used here.

L287-288: Could you please explain this in more detail? Which feature in Fig. 5d denotes orbital signal increase since 4.8 Ma? As far as I can infer from Fig. 5d, in the carbonate content plot, the orbital parameters increase since 4.9-5.0 Ma. While, in the Xpedo plot, it seems the increasing timings are diachronous for different orbital parameters.

L292-295: Here the authors propose that the carbonate content and Xpedo signals reflect incomplete preservation of paleoclimate signals. Then the question is if the original paleoclimate signals are incomplete, how would you use these records to predict paleoclimate changes?

L308-312: According to the authors' statement, the rapid change from 6.7-4.8 Ma low amplitude to 4.8-4.1 Ma large amplitude is observed in all the three orbital parameters. But for the benthic foraminiferal d18O record, similar change is only observed in the 41-kyr component. Why? This does not read like strong evidence to infer that the wet-dry oscillations were driven by changes in ice volume or global temperature. An

associated question would be if the authors do not consider solar radiation intensity is the cause of the wet-dry cycles, but ice volume or global temperature, then what's the cause of ice volume and global temperature changes? Isn't solar radiation intensity a driving factor?

L317-318: I find this conclusion hard to believe. For the carbonate content signal, the authors state that they record incomplete paleoclimate signal (see comments for L292-295). For the K2O/Na2O and Rb/Sr record, a more apparent change seems to be at 4.6 Ma.

If higher carbonate content represents dry climate, and lower carbonate content represents humid climate, compared with 6.7-4.8 Ma, the 4.8-3.6 Ma would have more humid period, but also much drier period, because the 4.8-3.6 Ma has larger variability. While, I did not see a clear wetting trend.

L363-364: This is a false statement. Even at present, the Tibetan Plateau cannot block the Westerlies completely. The Westerlies can travel to the northeastern Tibet through valleys in the Tianshan.

L368: which plots are pedogenesis proxies? Cite the specific plots here. "roughly"? how rough? Better to give a quantitative value.

L391-392: is there evidence to suggest reduced amount of atmospheric water vapor? Weakening of the paleo-ASM and dominance of Westerlies can explain the aridity. This does not necessarily need reduced amount of atmospheric water vapor.

L469: "extremely wet"? wetter than any other period?

L528-530: I probably missed it, but how could your records reflect seasonality of precipitation? Which proxy records seasonal signals?

L532: why the strongest summer monsoon is between 4.6-4.25 Ma? What are the possible reasons for the decreasing strength after 4.25 Ma?

Figures:

Fig. 1: a, the present outline is too large, the wind vectors are too small to see. It's better to show a smaller region with more details; e.g., regions between 10N-50N, 70E-130E. c. highlights the Xiaoshuizi section. Hard to find now.

Fig. 2: These photos exhibit very few useful information.

Fig. 3: Between 4.8-4.6 Ma, most plots show a weird shape. Is this because there are limited samples compared with other time intervals?

Fig. 5: apparently the authors need to provide more information in the caption about their plots. For example, Fig. 5d, what does the color mean? What does the black curve represent? Also, the horizontal age scale is better to use Ma, but not ka, as Ma is used throughout the manuscript. In Fig. 5a-b, there are other strong periodicities denoted. How about these periodicities in Fig. 6d?

Fig. 6. It will be better to arrange all the proxies with the same logic, e.g., left-wet, right-dry.

References: Tapponnier, P., Xu, Z.Q., Roger, F., Meyer, B., Arnaud, N., Wittlinger, G., Yang, J.S., 2001. Oblique stepwise rise and growth of the Tibet Plateau. Science 294, 1671-1677. Wang, C.S., Dai, J.G., Zhao, X.X., Li, Y.L., Graham, S.A., He, D.F., Ran, B., Meng, J., 2014. Outward-growth of the Tibetan Plateau during the Cenozoic: A review. Tectonophysics 621, 1-43. Yuan, D.Y., Ge, W.P., Chen, Z.W., Li, C.Y., Wang, Z.C., Zhang, H.P., Zhang, P.Z., Zheng, D.W., Zheng, W.J., Craddock, W.H., Dayem, K.E., Duvall, A.R., Hough, B., Lease, R.O., Champagnac, J.D., Burbank, D.W., Clark, M.K., Farley, K.A., Garzione, C.N., Kirby, E., Molnar, P., Roe, G.H., 2013. The growth of northeastern Tibet and its relevance to large-scale continental geodynamics: A review of recent studies. Tectonics 32, 1358-1370. Li, J., Fang, X.M., Song, C.H., Pan, B.T., Ma, Y.Z., Yan, M.D., 2014. Late Miocene–Quaternary rapid stepwise uplift of the NE Tibetan Plateau and its effects on climatic and environmental changes. Quaternary

Research 81, 400-423. Guo, Z.T., Ruddiman, W.F., Hao, Q.Z., Wu, H.B., Qiao, Y.S., Zhu, R.X., Peng, S.Z., Wei, J.J., Yuan, B.Y., Liu, T.S., 2002. Onset of Asian desertification by 22 Myr ago inferred from loess deposits in China. Nature 416, 159-163. Dupont-Nivet, G., Krijgsman, W., Langereis, C.G., Abels, H.A., Dai, S., Fang, X., 2007. Tibetan plateau aridification linked to global cooling at the Eocene–Oligocene transition. Nature 445, 635-638. Bosboom, R.E., Dupont-Nivet, G., Grothe, A., Brinkhuis, H., Villa, G., Mandic, O., Stoica, M., Huang, W.T., Yang, W., Guo, Z.J., 2014. Linking Tarim Basin sea retreat (west China) and Asian aridification in the late Eocene. Basin Research 26, 621-640.

---

## Author Comment (AC1) · 2 Sep 2018

Dear reviewer, We would like thank you for having read and commented our manuscript and we would like to apologize for the delay in our answer. We are grateful for your questions and suggestions. It's very useful and enlightening. We will take consideration in the revised paper. Here, we provide some quick replies to your questions. Lines 114-115: What makes the XSZ red clay different geomorphologically? The Xiaoshuizi peneplain of the Maxian mountain occupies a critical transition position between the high-altitude TP and the low North China Craton (Li et al., 2017). Line 133-134: not sure exactly what is meant here. Is the XSZ core characterized by more continuous

deposition and records a longer time interval than the Shangyaotan core? Yes, SYT core is only covered the age from 6.4 Ma to 4.2 Ma. Line 168: Is all of the remaining Ca in silicate minerals? Won't a lot of it be loosely bound to clay minerals in the soils? Also, the correction for Phosphorous also needs to be explained. I'm guessing you are assuming some component of Ca-bearing phosphate minerals, but what is the basis for this assumption. Thanks for your questions and suggestions. No, not all of remaining Ca in silicate minerals and the Ca bound to clay minerals is also included. Silicate-bound CaO* is obtained, in theory, by the simple equation (Fedo et al., 1995): CaO*(mol) = CaO(mol) $-$CO2(calcite mol) $-$ 0.5 CO2(dolomite mol) $-$ 10/3 mol P2O5(apatite). It generally calculated based the assumption that all P2O5 is associated with apatite and all inorganic carbon is associated with carbonates. It may neglect the Ca bound to clay minerals and overestimate the component of Ca-bearing phosphate minerals (Garzanti and Resentini., 2016). The reason we use the equation to calculate the values is that we try to expel the possibility the variation of Sr is determined by the bound of secondary carbonate, but not by weathering intensity. For Sr can substitute Ca in secondary carbonates (Reeder et al., 2006; Buggle et al., 2011). We will explain it in the revised paper. Line 199: What do you mean by durations? Are you saying there are some thicker intervals of high magnetic susceptibility? Yes, it means the interval of strong pedogenesis sustained longer. Figure 1: I think it would help if you put a larger non-circle shape on panel A corresponding to the study site. Then you can remove the Xiashuizi label, which slightly obscures the vector. Then, match this symbol on panel C You are missing the white reversals between C3n.1n, C3n.2n, and C3n.3n on the Polarity plot for the XSZ section. These were included in the age model presented in Li et al. (2017). What do the black bars on the lithology column represent. Thank you for suggestions and pointing faults out. We have not noticed it in Fig. 1b. There is something wrong with this figure when we convert it into PDF format. Some thin white rectangles are missed. The black bars on the lithology column were the thin white rectangles representing the carbonate nodule layer. We give the figure1s of records versus stratigraphic depth. Reference Buggle B, Glaser B,

Hambach U, et al. (2011). An evaluation of geochemical weathering indices in loess–paleosol studies[J]. Quaternary International, 240(1–2):12-21 Fedo, C. M., Nesbitt, H. W., & Young, G. M. (1995). Unraveling the effects of potassium metasomatism in sedimentary rocks and paleosols, with implications for paleoweathering conditions and provenance. Geology, 23(10), 921-924. Garzanti, E., & Resentini, A. (2016). Provenance control on chemical indices of weathering (taiwan river sands). Sedimentary Geology, 336, 81-95. Li, J., Ma, Z., Li, X., Peng, T., Guo, B., & Zhang, J., et al. (2017). Late miocene-pliocene geomorphological evolution of the xiaoshuizi peneplain in the maxian mountains and its tectonic significance for the northeastern tibetan plateau. Geomorphology, 295. Reeder, S., Taylor, H., Shaw, R.A., Demetriades, A., (2006). Introduction to the chemistry and geochemistry of the elements. In: Tarvainen, T., de Vos, M. (Eds.), Geochemical Atlas of Europe. Part 2. Interpretation of Geochemical Maps, Additional Tables, Figures, Maps, and Related Publications. Geological Survey of Finland, Espoo, pp. 48-429

[Figure]

Fig. 1 s.  Variations in carbonate content, major element concentration, minor element concentration, magnetic susceptibility and grain size from the XSZ red clay section, spanning 6.7-3.6 Ma

**Fig. 1.**

---

## Author Comment (AC2) · 12 Sep 2018

**Dear reviewer,**

We would like thank you for having read and commented our manuscript and we would like to apologize for the delay in our answers. We are grateful for your questions and suggestions. It's very useful and enlightening. We will take consideration in the revised version. Here, we provide some quick replies to your questions.

I think the manuscript needs to present the geochemistry data versus stratigraphic depth, in addition to just age. There also needs to be more discussion on the relationship between sedimentation rate and pedogenesis. For example, it would be helpful if Figure 3 was plotted vs. depth and there was also a column that plots sedimentation rate, and the presence of nodule horizons. This is important because the interval between 4.5 and 4.3 Ma, for example, shows a strong increase in magnetic evidence for pedogenesis and also coincides with a noticeable drop in deposition rate. Therefore, it needs to be discussed if this increase in pedogenesis was driven solely by wetter conditions, or was there also more time for soil formation and leaching of Ca. I do think more stratigraphic context will help some of the arguments presented in the text. For example, upon my initial reading of the text and figure, the division into the 2 primary intervals placed at 4.8 Ma seemed somewhat arbitrary looking at figure 3 (i.e. why not 4.6 or 5.1). But it makes much more sense in terms of the large decline in sedimentation rate around 4.8, which accompanied by the deposition of a carbonate nodule layer, and then the noted increase up-section in nodule horizons underlaying leached zones. Also with deposition of loess being connected to regional wind patterns, is it significant that there was a notable \_200 kyr drop in sedimentation rates before a shift to generally wetter/more seasonal conditions?

**Our response:** Many thanks for your useful suggestions. We would use Fig 1s to replace the Fig 3 and we would add a brief statement "Profiles of the various proxies are illustrated in Fig 3 and there is an obvious difference in the character of the fluctuations above and below the depth of 16.5 m (~4.8 Ma). Above 16.5 m, the carbonate content fluctuates at a lower level but with greater amplitude, and the magnetic susceptibility also fluctuates at a greater amplitude. In addition, the CV of most of the records is greater above the boundary than below (Table 1). This suggests that the climate became more humid and variable after 4.8 Ma. Meanwhile, a noticeable drop in deposition rate around 4.8 Ma occurred (Li et al., 2017). Thus, the red clay sequence was divided into two intervals: *Interval I* (6.7-4.8 Ma) and *Interval II* (4.8-3.6 Ma). The characteristics of the individual proxy records are describe in detail below" in front of line 174. We will also add "We use the coefficient of variation (CV) to measure the variability of the records. The higher the CV, the more variable the record. The CV is defined as:

 $CV = 100 * \frac{Standard deviation,}{Mean}$  at the end of chapter 3( line 172).

I am somewhat confused by the explanation of K/Al ratios as a weathering proxy (lines 238-245). With time, Al can mobilize and become depleted at the top of a paleosol and enriched down profile. And in certain situations, you might expect K to be enriched at the soil surface, due to its biological importance. So, within the same well developed soil, you might expect a higher K/Al ratio at the top, and a lower ratio deeper in the profile. This is never plotted, so it might be worth eliminating this text? The various magnetic susceptibility terms are well described in the discussion, but I think it would help readers if at least some of this information was moved up to either the results or methods. This would help provide context to all of the values presented in the results.

**Our response:** We would consider removing the K2O/Al2O3 ratio and modify the statement "In addition, previous..." in lines 238-252 as "In addition, the K2O/Na2O ratio is used to evaluate the clay content in loess and is also a measure of plagioclase weathering, avoiding biases due to uncertainties in separating carbonate Ca from silicate Ca (Liu et al., 1993; Buggle et al., 2011). Na2O is mainly produced by plagioclase weathering and is easily lost during leaching as precipitation increases. By contrast, K2O (mainly produced by the weathering of potash feldspar) is easily leached from primary minerals and is then absorbed by secondary clay minerals with ongoing weathering (Yang et al., 2006; Liang et al., 2013). In the arid and semi-arid regions of Asia, K2O is enriched in palaeosols compared to loess horizons (Yang et al., 2006). Thus, high K2O/Na2O ratio also increased rapidly at about 4.8-4.7 Ma and maintained relatively high values after 4.7 Ma. This may indicate that the overall weathering intensity was sufficient to produce secondary clays, resulting in a spike in K2O concentration" in lines 409-312.

**Minor suggestions:**

Line 57: suggest "occurring" or "underway" instead of "ongoing" Our response: We would modify "ongoing" as "underway".

Line 76: suggest "supplied" instead of "prepared" Our response: We would modify "prepared" as "supplied".

Lines 75-88: I'm guessing the sentence beginning with "Make clear: : :" on line 78 was accidentally left in as a comment, which I still think needs to be addressed. I think I understand what the authors are going for within the paragraph, but I think the logic can be expressed more clearly. The strength/onset of the Asian monsoon is linked to these globally significant events (Tibetan uplift, northern hemisphere ice, etc). Therefore, by constraining paleoclimate across the Chinese Loess Plateau not only does this improve our understanding of regional climate, but it can also provide insight about the paleomonsoon, and therefore changes in the global climate system during the Pliocene. **Our response:** Many thanks for your suggestions. We would modify the statement of lines 75-88 as "The Asian summer monsoon (ASM) and the meridional (westerlies) circulation systems, as major components of the atmospheric circulation, delivered moisture to Eurasia. The onset and strength of the Asian monsoon during the early Pliocene was linked to the uplift of the Tibetan Plateau (TP), changes in latitudinal and longitudinal heat gradients, global temperature and ice volume (An et al., 2001; Ding et al., 2001; Li et al., 2008, 2010; Clift et al., 2008; Nie et al., 2014; Ao et al., 2016). Therefore, determining the range of climatic conditions across the Chinese Loess Plateau (CLP) during the Pliocene not only improves our understanding of the regional climate, but it can also provide insights into the paleomonsoon, and thus into changes in the global climate system at this time."

**Line 96: suggest removing "condition" and changing "aridification process" to "regional aridification"**

**Our response:** We would removing "condition" and modify "aridification process" as "regional aridification"

Line 104: change "to be" to "that", and I think it would be helpful for the future readers not just to say "gleying", but instead state briefly what that means (waterlogging, and iron reduction) and why it matters for the magnetic susceptibility record. Our response: We would take consideration in the revised version.

**Line 105-106: This sentence does not make sense. Are you trying to say that climate in this region is influenced by the strength of both the westerlies and the monsoon, and that those two factors may not be directly related?**

**Our response:** We would remove it and add "where climate is dominated by westerlies," after "Tarim basin" in line 92.

**Lines 114-115: What makes the XSZ red clay different geomorphologically?**

**Our response:** The Xiaoshuizi peneplain of the Maxian mountain occupies a critical transition position between the high-altitude TP and the low North China Craton (Li et al., 2017). The obvious difference between Xiaoshuizi deposit and the red clay in the Chinese Loess Plateau is the modern altitude, and this exactly results from the special geographical position of NE Tibetan Plateau.

Line 118: suggest "are" instead of "have been"

Our response: We would modify "have been" as "are"

**Line 121: This sentence is slightly off.**

**Our response:** We would modify it as "Finally, we consider the nature of the regional climate and its possible mechanisms"

Line 133: suggest "reconstruct and discuss" instead of "discuss " Our response: We would modify "discuss" as "reconstruct and discuss"

Line 133-134: not sure exactly what is meant here. Is the XSZ core characterized by more continuous deposition and records a longer time interval than the Shangyantan core?

Our response: Yes, SYT core is only covered the age from 6.4 Ma to 4.2 Ma.

Line 136: capitalize China Our response: We would modify it.

Lines 137-138: Not sure what is mean by the sentence beginning with "The East Asian Monsoon." Are you trying to explain how these two factors together control climate at the study site. This could be elaborated. Our response: We will remove it.

Line 144: Where in the section is the increase in gravel? From the strat column it looks like it is at the base. Say this in-text.

Our response: We would add "at the base" after "...gravel content" in line 144.

Lines 145-147: Clarify if most carbonate horizon are overlain by a brownish red-layer, or if the carbonate zone in its entirety underlies a larger brownish-red layer. Lines 148-150: It's not clear as written if carbonized root channels have more abundant Fe-Mn staining.

**Our response:** We would modify statement of lines 145-147 as "The upper 20 m contains numerous horizontal carbonate nodule horizons and most of these horizons underline the brownish red layer" and modify "horizons containing" in line149 as "the".

Line 168: Is all of the remaining Ca in silicate minerals? Won't a lot of it be loosely bound to clay minerals in the soils? Also, the correction for Phosphorous also needs to be explained. I'm guessing you are assuming some component of Ca-bearing phosphate minerals, but what is the basis for this assumption.

**Our response**: Thanks for your questions and suggestions. No, not all of remaining Ca in silicate minerals and the Ca bound to clay minerals is also included. Silicate-bound CaO\* is obtained, in theory, by the simple equation (Fedo et al., 1995): CaO\*(mol) = CaO(mol)  $-CO_2(\text{calcite mol}) - 0.5$ CO2(dolomite mol) - 10/3 mol P2O5(apatite). It generally calculated based the assumption that all P2O5 is associated with apatite and all inorganic carbon is associated with carbonates. It may neglect the Ca bound to clay minerals and overestimate the component of Ca-bearing phosphate minerals (Garzanti and Resentini., 2016). The reason we use the equation to calculate the values is that we try to expel the possibility the variation of Sr is determined by the bound of secondary carbonate, but not by weathering intensity. For Sr can substitute Ca in secondary carbonates (Reeder et al., 2006; Buggle et al., 2011). We will modify statement "The molar content of silicate Ca (CaO\*) was calculated using the following equation:" as "Silicate-bound CaO (CaO\*) can be estimated, in principle, by the equation:  $CaO^*(mol) = CaO(mol) - CO_2(calcite mol) - 0.5 CO_2(dolomite mol) - 10/3 mol P_2O_5(apatite) (Fedo et al., 1995). It is generally calculated based on the assumption that all the P_2O_5 is associated with apatite and all the inorganic carbon is associated with carbonates Thus, the CaO* of the XSZ red clay was calculated using the following equivalent equation".$

**Line 199: What do you mean by durations? Are you saying there are some thicker intervals of high magnetic susceptibility?**

Our response: Yes, it means the interval of strong pedogenesis sustained longer.

Line 256: space between "susceptibility" and "of" Our response: We would correct it.

Line 257: suggest removing "two" Our response: We would remove it.

Line 314: Spelling of "Multiproxy" Our response: We would modify it.

Line 317-318: suggest "a significant change is recorded by most of the proxies that occurred"

Our response: We would take consideration in the revised version.

Line 318: K/Al is not plotted, but K/Na is plotted. Based on the comment above, I think this is probably a better choice.

Our response: We would modify "K2O/Al2O3" as "K2O/Na2O".

Line 327: suggest "relatively" instead of "relative" and "and" instead of "which" Our response: We would modify "relative" as "relatively" and modify "which" as "and".

Line 328: Not sure what this sentence is trying to say. Our response: This sentence may be redundancy. We would remove it.

Line 329: I suggest clarifying the beginning of this sentence to say something along the lines of "Carbonate content becomes more variable after 5.5 Ma, which is..." Our response: We would modify the sentence "It is evident that the carbonate content decreases with increased variation amplitude after 5.5 Ma" as "Carbonate content becomes more variable after 5.5 Ma, which is..."

Line 333: spelling of "indices"

Our response: We would correct it.

Line 345: suggest "central and eastern" instead of "hinterland of the" Our response: We would modify "hinterland of the" as "central and eastern".

Line 377: suggest rewording the sentence beginning with: "Look around the globe,..." Our response: We would remove the sentence.

Line 415: I'm not sure what "humid toward arid direction" means Our response: It means climate tended to be dryer.

Line 521: suggest "provides the opportunity constrain and discuss.." Our response: We would take consideration in the revised version.

Line 526: again suggest "central and eastern" instead of "hinterland of the" Our response: We would take consideration in the revised version.

Line 531: suggest removing "obviously" Our response: We would remove it.

Figure 1: I think it would help if you put a larger non-circle shape on panel A corresponding to the study site. Then you can remove the Xiashuizi label, which slightly obscures the vector. Then, match this symbol on panel C You are missing the white reversals between C3n.1n, C3n.2n, and C3n.3n on the Polarity plot for the XSZ section. These were included in the age model presented in Li et al. (2017). What do the black bars on the lithology column represent.

**Our response:** Thank you for suggestions and pointing faults out. We have not noticed it in Fig. 1b. There is something wrong with this figure when we convert it into PDF format. Some thin white rectangles are missed. The black bars on the lithology column were the thin white rectangles representing the carbonate nodule layer. We would give the new figure (Fig 1).

**Figure 2: I think it would help if the line thicknesses were slightly thinner.**

Our response: You mean figure 3? We would modify it.

**Reference**

Buggle B, Glaser B, Hambach U, et al. (2011). An evaluation of geochemical weathering indices in loess–paleosol studies[J]. Quaternary International, 240(1–2):12-21

- Fedo, C. M., Nesbitt, H. W., & Young, G. M. (1995). Unraveling the effects of potassium metasomatism in sedimentary rocks and paleosols, with implications for paleoweathering conditions and provenance. Geology, 23(10), 921-924.
- Garzanti, E., & Resentini, A. (2016). Provenance control on chemical indices of weathering (taiwan river sands). Sedimentary Geology, 336, 81-95.
- Li, J., Ma, Z., Li, X., Peng, T., Guo, B., & Zhang, J., et al. (2017). Late miocene-pliocene geomorphological evolution of the xiaoshuizi peneplain in the maxian mountains and its tectonic significance for the northeastern tibetan plateau. Geomorphology, 295.
- Reeder, S., Taylor, H., Shaw, R.A., Demetriades, A., (2006). Introduction to the chemistry and geochemistry of the elements. In: Tarvainen, T., de Vos, M. (Eds.), Geochemical Atlas of Europe. Part 2. Interpretation of Geochemical Maps, Additional Tables, Figures, Maps, and Related Publications. Geological Survey of Finland, Espoo, pp. 48-429

**Figures and tables**

---

## Author Comment (AC3) · 12 Sep 2018

Dear reviewer,

We would like thank you for having read and commented our manuscript and we would like to apologize for the delay in our answers. We are grateful for your questions and suggestions. It's very useful and enlightening. We will take consideration in the revised version. Here, we provide some quick replies to your questions.

**Spelling and Grammar:**

**I have not edited this manuscript for spelling and grammar. I strongly encourage the authors to seek assistance from a very proficient or native English speaker. Also, please review the manuscript for organizational mistakes (e.g. Figures 2 and 3 are not cited in the text; incorrect citations).**

**Our response:** Thank you for your suggestions. We would take consideration in the revised version and we will cite the Figures2 in chapter 3(Material and methods) and Figure 3 in chapter 4(Results).

**Statistics:**

**The authors need to provide more information about the magnetostratigraphic ages. What is the temporal resolution of the records? What are the temporal uncertainties? Can the records accurately resolve all the cycles you discuss (e.g. precession)? Does variable deposition rate impact the signals?**

**Our response:** The average temporal resolution of records is 3.8 kyr. The resolution of records for detecting the precession signal needs to be 4 kyr or less (Luo et al., 2017). 80 % of sampling intervals satisfied the requirement with the resolution. Thus, the records can theoretically document the eccentricity and obliquity cycle of entire period and document the precession cycle of 80 % period. The variable deposition rate does impact on the conservation of the signals especially for precession signal.

**How did the authors decide that 4.8 Ma was the appropriate transition point? It seems arbitrary to me. I see no clear transition in Figure 3. Are the two periods (6.7-4.8 Ma and 4.8-3.6 Ma) statistically distinct?**

**Our response:** Figure 3 may not show the distinct of two periods clearly enough due to variable deposition. Thus, we present figure of records versus stratigraphic depth (Fig 1s). Synthesized the

values and variations of the carbonate content, elements content and magnetic susceptibility, a

transition period is presented at 16.5-15 m (4.8-4.6 Ma). There is an obvious difference in the character

of the fluctuations above and below the depth of 16.5-15 m. For example, above the 16.5 m, the

carbonate content fluctuates at a lower level but with larger amplitude accompanied by the noted

increase in nodule horizons underlaying leached zones in the field, and the magnetic susceptibility also

fluctuates at greater amplitude. Meanwhile, a noticeable drop in deposition rate around 4.8 Ma

occurred. Thus, we define the 4.8 Ma as the transition point.

The average value and coefficient of variation of the records during two periods (6.7-4.8 Ma and

4.8-3.6 Ma) have been given in Tables 1s. The coefficient of variation (CV) is defined as:

$$CV = 100 * \frac{\text{Standard deviation}}{\text{Mean}}$$

The higher the CV is, the more changeable the record is. It shows the average value and CV of the

most records show the obvious difference between two periods and most of the records are more

changeable during 4.8-3.6 Ma than 6.7-4.8 Ma.

**I find the signal filtering in Figure 5 questionable. First, the authors filter the data at**

**frequencies with insignificant power (e.g. the 100 kyr filtering of carbonate content).**

**Further, the most significant signals exist at frequencies that are difficult to explain,**

**which the authors dismiss, and many of the discussed signals are barely significant**

**at 90% confidence. The wavelet plots highlight the limited signal strength. Even if the**

**filtered signals are sound, the filtered signals changes do not well align with the benthic**

**d$^{18}$O record.**

**Our response:** The reason we filtered the carbonate at 100 kyr is that we observed that fluctuations of

$CaCO_3$ and weathering indices agree well with eccentricity orbital variations at 4.8-3.9 Ma (Fig 2s-c).

We perform the spectral analysis for carbonate content in two periods (6.7-4.8Ma and 4.8-3.6 Ma)

respectively. It shows 100 kyr, 41 kyr and 21 kyr periodic signals of carbonate are significant in the

period of 4.8-3.6 Ma (Fig 2s). However, most of the orbital periodic signals are insignificant in the

period of 6.7-4.8 Ma.

As for non-orbital periodic signals, we have not found the driving force and the signals recorded

by carbonate content are different from $\chi_{pedo}$. It may indicate non-orbital periodic signals are fake and

more significant non-orbital periodic signals of carbonate content are relate to the

dissolution-reprecipitation process of carbonate. Thus, we do not filter the records at these frequencies.

We consider removing wavelet plots.

**Interpretation:**

**The potential drivers of the climate signals are often overstated. Connections are made**

**with limited support. Many of the mechanisms discussed are still debated, particularly**

**the Isthmus of Panama hypothesis and timing of Tibetan Plateau uplift. At the least,**

**the authors need to do a better job citing recent literature and discussing the remaining**

**uncertainties. Also, many of the citations are not primary sources for the associated**

**statements.**

**Our response:** We would pay special attention to these problems in the revised version.

**Specific Comments:**

**Line 51: Earth's orbital went through many cycles over this period, so the "orbital configuration**

**statement does not make much sense.**

**Our response:** We would remove the statement "(ii) orbital configuration".

**Lines 51-53: These statements, such as "comparable temperatures in the tropic region",**

**require citations.**

**Our response:** We would add the citations (Herbert et al., 2010, 2016).

**Line 65: Please clarify the link between mean tropical Pacific east-west gradient and**

**ENSO.**

**Our response:** We would modify the statement of line 65 as "which may have given rise to permanent**

El Nino Southern Oscillation (Lawrence et al., 2006)."

**Line 68: The timing of uplift of the Tibetan Plateau is heavily debated…**

**Our response:** We consider removing the statement.

**Line 70: Lunt et al. (2008) is not a direct source for the closure of the Isthmus of Panama. More recent works debate the timing of closure (e.g. Bacon et al., 2015; O'Dea et al., 2016).**

**Our response:** We consider removing the statement.

**Line 72: This statement is also not well supported. For example, Lunt et al. (2008), who are cited earlier, found closure of the Panama seaway to have little influence on NH glaciation. In general, the authors need to update their citations and discuss the Literature more thoroughly. These ideas are far from settled, yet they are presented as facts.**

**Our response:** Thank you for your suggestions. We would modify the statement of lines 68-72 as "In addition, several major changes in global thermohaline and atmospheric circulation system occurred during the early Pliocene which are thought to be crucial preconditions for both the appearance of Northern Hemisphere ice sheets at ~2.6 Ma (Haug et al., 1998, 2005; Driscoll et al., 1998) and the development of the modern east-west hydrographic gradient in the equatorial Pacific (Lawrence et al., 2006; Chaisson et al., 2000)."

**Line 80-82: Citation?**

**Our response:** We would add the citations (An et al., 2001; Ding et al., 2001; Li et al., 2008, 2010; Clift et al., 2008; Nie et al., 2014; Ao et al., 2016).

**Line 85: "Arctic volume" means "Arctic ice volume"?**

**Our response:** Yes, we would correct it.

**Lines 105-106: This statement does not make sense.**

**Our response:** We would remove the statement.

**Lines 136-140: Sources for these data?**

**Our response:** We would modify the statement of lines 136-138 as "The mean annual temperature during 1986-2016 was ~7.0 ℃ and the annual precipitation was 260-550 mm. Most (80%) of the

precipitation is in summer and autumn. (The data were obtained from the National Meteorological Information Center (http://data.cma.cn/) of the Chinese Meteorological Administration (MCA)"

**Line 154: This requires more detail.**

**Our response:** We would add the statement "The average temporal resolution of the records is 3.8 kyr. Some 80 % of the sequence has a sampling resolution of 4 kyr or less."

**Line 175-182: How did you decide on these intervals? Did you test that they are statistically distinct?**

**Our response:** We would add a brief statement "Profiles of the various proxies are illustrated in Fig 3 and there is an obvious difference in the character of the fluctuations above and below the depth of 16.5 m (~4.8 Ma). Above 16.5 m, the carbonate content fluctuates at a lower level but with greater amplitude, and the magnetic susceptibility also fluctuates at a greater amplitude. In addition, the CV of most of the records is greater above the boundary than below (Table 1). This suggests that the climate became more humid and variable after 4.8 Ma. Meanwhile, a noticeable drop in deposition rate around 4.8 Ma occurred (Li et al., 2017). Thus, the red clay sequence was divided into two intervals: *Interval Ⅰ* (6.7-4.8 Ma) and *Interval Ⅱ (*4.8-3.6 Ma). The characteristics of the individual proxy records are described in detail below" in front of "Carbonate content" in line 174. We will also add "We use the coefficient of variation (CV) to measure the variability of the records. The higher the CV, the more variable the record. The CV is defined as:

$$CV = 100 * \frac{\text{Standard deviation,,}}{\text{Mean}}$$

at the end of chapter 3( line 172).

**Line 224-226: How are you sure that it relates to monsoon strength? Could it be seasonal or evaporative changes?**

**Our response:** We agree with you. However, on the condition moisture is carried by the monsoon and the monsoon is strong enough, $CaCO_3$ could indicate the monsoon strength (Fang et al., 1999; Sun et al., 2010).

**Lines 235-237: Both statements are significant at 99% confidence?**

**Our response:** The correlation between Sr and CaO* (silicate CaO) is significant at 99% confidence, while the correlation between Sr and CaCO$_3$ is not significant.

**Lines 282-286: Are you sure these signals are real? If so, how might you explain the cycles not related to orbital variability?**

**Lines 294-295: Doesn't this "incomplete nature of the red climate time series" impact all of the frequency analyses? How can you distinguish real and fake signals?**

**Our response:** The questions are really worth pondering. Firstly, our chronology is reliable. Secondly, all sampling intervals of XSZ red clay satisfies requirement to detect eccentricity and obliquity signals and 80 % of sampling intervals satisfies requirement to detect the precession signal. Thirdly, three orbital periodic signals were also detected in the other sites of the CLP from late Miocene to early Pliocene, which means changes of orbital parameters really had impact on climate of the CLP (Han et al., 2011; Li et al., 2008). Thus, 100 kyr, 41 kyr and 21 kyr periodic signals recorded by XSZ red clay are probably true.

Changes of Earth orbital parameters would dynamically lead to the variation of the climate. However, the change of Earth orbital parameters is just one of forcing factors and other factors (some internal process or feedback) could magnify or cover the orbital forcing, which means the climate changes probably show non-linear response to orbital forcing. In this specific case, it might be the expansion of the palaeo-ASM that enhanced the orbital periodic signals of XSZ red clay between 4.8 and 4.1 Ma. As for short cycles, the power of these cycles would be weakened by the low and uneven sedimentation accumulation rate (Luo et al., 2017). The incomplete nature of the red climate time series would also impact on the conservation of the signals especially for precession signal. Meanwhile, the age model has not been astronomically tuned. Thus, it's hard to completely match the filtered 41 kyr and 21 kyr components with the lagged obliquity and precession in phase and amplitude even these signals are real. Our results resemble to those of Han (2011) and Tian (2002) that three orbital periodic signals were significant while records and orbital variability were less matched from late Miocene to early Pliocene. On the other hands, at least to date, we have not found the driving force yielding these non-orbital periods. Thus, these non-orbital periodic signals are probably random or fake.

**Line 302: I believe that a 23 kyr filter makes more sense for the climate response to**

**orbital change.**

**Our response:** The 21kyr filtered component is filtered at 18-24kyr. The 23kyr filtered component was included.

**Line 304: What record? Lisiecki and Raymo (2005)?**

**Our response:** The data was the filtered components of the $\delta^{18}O$ record (Zachos et al., 2001) at the 21-kyr, 41-kyr, and 100-kyr bands.

**Line 306: I do not observe this in the filtered record…Is this change significant? How much do these filter components contribute to the complete signal?**

**Our response:** This shift may be not obvious in the 100-kyr filtered components but obvious in 41-kyr and 21-kyr filtered components especially in 41-kyr filtered components. We don't know how to measure the contribution. However, fig 2s shows 100 kyr, 41 kyr and 21 kyr periodic signals of carbonate content in the interval of 4.8-3.6 Ma are more significant than the interval of 6.7-4.8 Ma.

**Line 309-310: Where is this shown? The 41 kyr signal in the benthic records do not well align with the data.**

**Our response:** The shift may be not obvious in Fig 5 where we put all filtered curves together. Fig 4s shows 41 kyr signal in the benthic record and XSZ records enhanced between about 4.8 and 4.1 Ma. In my opinion, three curves have shown some similarities during the period of 4.8-3.6 Ma, with larger oscillation at the intervals of 4.7-4.4 Ma and 4.2-3.9 Ma, and damped oscillation at the interval of 3.9-3.6 Ma. On the other hands, the record has its own climatic significance and limitation, and even the 41 kyr filtered curves of $CaCO_3$ and $\chi_{pedo}$ show difference. Thus, the differences of the 41 kyr signal in benthic records and XSZ records are reasonable.

**Lines 317-319: I see no clear changes in the records. You need statistical support.**

**Our response:** Tables 2s has shown that from the period of 6.7-4.8 Ma to 4.8-3.6 Ma, average value of $CaCO_3$ decreased, weathering proxies and magnetic susceptibility increased. CV of the most proxies increased.

**Line 361: ODP source?**

**Our response:** Yes, ODP 1148.

**Line 368: "…roughly parallel…" I do not see a correlation. Please quantify.**

**Our response:** We interpolate the $d^{18}O$ with the age of Xiaoshuizi from 4.8 Ma to 6.7 Ma. Fig 4s-a shows that the extracted data (black line) match well with original $d^{18}O$ data (brown line). Fig 4s-b shows $\chi_{pedo}$ has a significant positive relationship with $d^{18}O$ at 80 % confidence during the period of 6.7- 4.8 Ma.

**Line 384: This is possible but not necessarily the case.**

**Our response:** We would modify "would have resulted" as "could result"

**Lines 392-393: Cooler air can hold less vapor, but this statement is an extreme simplification.**

**Our response:** We would modify the statement of lines 391-393 as "However, moisture sources for the westerly flow are distant from the CLP (Nie et al., 2014), and only a relatively small amount of moisture was carried to the CLP, resulting in a dry and stable climate in the XSZ region."

**Lines 402-403: You record captures seasonal variability?**

**Our response:** Research on migration process of carbonate indicated seasonally wet/dry climate is a key factor in driving carbonate dissolution and reprecipitation, and strong seasonally biased precipitation enhances the leaching process and produces thick leached horizons (Rossinsky and Swart, 1993; Zhao, 1995, 1998). We would add it in front of "The emergence of…"

**Lines 469-471: Citation?**

**Our response:** We would add the citations (Jackson and O'Dea, 2013; O'Dea et al., 2016).

**Line 480: Why global moisture and not local moisture?**

**Our response:** It referred to moisture at high northern latitudes. We would correct it.

**Lines 484-487: This does not make sense.**

**Our response:** We would remove the statement.

**Lines 491-492: Are you talking about regional or global albedo?**

**Our response:** It's reduced ice albedo at high northern latitudes. We would correct it.

**Lines 492-495: Citation?**

**Our response:** We would add the citations (Chang et al., 2011; Sun et al., 2015 ).

**Lines 497-498: Could this discrepancy relate to differences between short term variability and the mean climate state?**

**Our response:** Yes, it's one possibility.

**Line 502: "We noticed"? You mean the authors of these other publications noticed?**

**Our response:** We would modify it as "Several crucial changes linked with the summer monsoon occurred: There was a vast expansion of the western Pacific warm pool into subtropical regions in the early Pliocene (Brierley et al., 2009; Fedorov et al., 2013), and temperatures at the edge of the warm pool showed a warming trend of ~2℃ from the latest Miocene to the early Pliocene (Karas et al., 2011)"

**Lines 502-506: How close are these events in time?**

**Our response:** All of these events started at 5.2-4.8 Ma and developed at ~4.6 Ma.

**Figures:**

**Figure 1a: The winds do not look correct. Also, 850 hPa winds do not exist over the Plateau**

**Our response:** We would correct it and provide new figure (Fig 1).

**Figure 2 and Figure 3 are not cited in the text.**

**Our response:** We would cite the Figure 2 at the end of sentence "…clay is composed of brownish red and yellowish clay layers" in line 145 and Figure 3 in chapter 4(Results).

**Figure 3: It is difficult to see how the axes align with the lines**

**Our response:** We would use figure1s to replace the figure 3

**Figure 5d: Do the black lines represent significance?**

**Our response:** Yes, these lines are the 95% confidence limit line. We would provide new figure (Fig 5).

**References**

Han, W., Fang, X., Berger, A., & Yin, Q. (2011). An astronomically tuned 8.1 ma eolian record from the chinese loess plateau and its implication on the evolution of asian monsoon. Journal of Geophysical Research Atmospheres, 116(D24), -.

Li, F. J., Rousseau, D. D., Wu, N., Hao, Q., & Pei, Y. (2008). Late neogene evolution of the east asian monsoon revealed by terrestrial mollusk record in western chinese loess plateau: from winter to summer dominated sub-regime. Earth & Planetary Science Letters, 274(3–4), 439–447.

Luo, Z., Su, Q., Wang, Z., Heermance, R. V., Garzione, C., & Li, M., et al. (2017). Orbital forcing of plio-pleistocene climate variation in a qaidam basin lake based on paleomagnetic and evaporite mineralogic analysis. Palaeogeography Palaeoclimatology Palaeoecology.

Fang, X. M., Ono, Y., Fukusawa, H., Pan, B. T., Li, J. J., & Guan, D. H., et al. (1999). Asian summer monsoon instability during the past 60,000 years: magnetic susceptibility and pedogenic evidence from the western chinese loess plateau. Earth & Planetary Science Letters, 168(3–4), 219-232.

Rossinsky Jr., V., Swart, P.K., 1993. Influence of climate on the formation and isotopic composition of calcretes. In: Swart, P.K., Lohmann, K.C., McKenzie, J., Savin, S. (Eds.), Climate Change in Continental Isotopic Records, American GeophysicalUnion: Geophysical Monography, 78, pp. 67-75.

Sun, Y., An, Z., Clemens, S. C., Bloemendal, J., &Vandenberghe, J. (2010). Seven million years of wind and precipitation variability on the chinese loess plateau. Earth & Planetary Science Letters, 297(3–4), 525-535.

Zachos, J., Pagani, M., Sloan, L., Thomas, E., & Billups, K. (2001).Trends, rhythms, and aberrations in global climate 65 ma to present.Science, 292(5517), 686-93.

Zhao, J. B. (1995), A study of the $CaCO_3$ illuvial horizons of paleosols and permeated pattern far rain water, J Geogr Sci, 15(4), 344-350.

Zhao, J. B. (1998), Illuvial $CaCO_3$ layers of paleosol in loess and its environmental significance,

Journal of Xi'an Engineering University, 20(3), 46-49.

**Figures and tables**

[Figure]

Fig. 1. The location of the study area and atmospheric circulation patterns. (a) 850 mb vector wind averaged from June to August for 1982-2012 based on NOAA Earth System Research Laboratory reanalysis data (Compo et al., 2013). (b) Lithology and magnetostratigraphy of the XSZ drill core. (c) The Chinese Loess Plateau with locations of the studied Xiaoshuizi site and other sections mentioned in the text.

[Figure]

Fig. 1 s.  Variations in carbonate content, major element concentration, minor element concentration, magnetic susceptibility and

grain size from the XSZ red clay section, spanning 6.7-3.6 Ma

[Figure]

Fig. 2s.  Spectrum analysis of carbonate content during the period of (a) 4.8-3.6 Ma (b)  6.7-4.8 Ma on original

paleomagnetism chronology.  (d) Carbonate and chemical weathering intensity fluctuations linked to eccentricity

and obliquity orbital variations at 4.8–3.9 Ma.

[Figure]

Fig. 3s.  Comparison of obliquity (Laskar et al., 2004) with filtered components of the carbonate content, $\chi_{pedo}$ and $\delta\ ^{18}O$ records (Zachos et al.,2001) at the 41-kyr bands.

[Figure]

[Figure]

Fig. 4s. (a) The comparison of benthic $\delta^{18}O$ between extracted data (black line) and original $\delta^{18}O$ data (Zachos et al., 2001).

(b) Scatter plot of benthic $\delta^{18}O$ versus $\chi_{pedo}$ during 6.7-4.8 Ma

Table. 1s. The average value and coefficient of variation of the records during two periods of 6.7-4.8 Ma and 4.8-3.6 Ma.

| | | $SiO_2$(%) | $Al_2O_3$(%) | CaO(%) | $Fe_2O_3$(%) | $K_2O$(%) |
|---|---|---|---|---|---|---|
| 4.8-3.6 Ma | Average | 48.9 | 13.2 | 11.6 | 5.69 | 3 |
| | CV | 14.6 | 9.58 | 45.57 | 9.3 | 12.3 |
| 6.7-4.8 Ma | Average | 49.5 | 12.2 | 11.2 | 5.2 | 2.6 |
| | CV | 11.6 | 9.09 | 32.18 | 9.6 | 10.3 |
| | | $Na_2O$(%) | MgO(%) | Sr(ppm) | Rb(ppm) | Ba(ppm) |
| 4.8-3.6 Ma | Average | 1.23 | 2.3 | 214.6 | 109.9 | 558 |
| | CV | 24.4 | 9 | 12.5 | 10.9 | 11.5 |
| 6.7-4.8 Ma | Average | 1.2 | 3.1 | 211.7 | 103.9 | 494 |
| | CV | 10.2 | 61 | 10.04 | 10.8 | 13.2 |
| | | $CaCO_3$(%) | $\chi_{hf}$ | $\chi lf$ | $\chi fd$ | |
| 4.8-3.6 Ma | Average | 13.8 | 25.4 | 26.9 | 1.2 | |
| | CV | 45.6 | 38.3 | 36.2 | 78.2 | |
| 6.7-4.8 Ma | Average | 17.4 | 19.4 | 20.3 | 1 | |
| | CV | 29.3 | 23.8 | 21.2 | 72.8 | |

Table. 2s. The average value and coefficient of variation of the proxies during two periods of 6.7-4.8 Ma and 4.8-3.6 Ma.

| | | $CaCO_3$ | Rb/Sr | $K_2O/Na_2O$ | $\chi_{lf}$ | $\chi_{pedo}$ | <2um/>43um |
|---|---|---|---|---|---|---|---|
| 4.8 -3.6 Ma | Average | 13.8 | 0.52 | 2.49 | 26.9 | 10.9 | 1.52 |
| | CV | 45.59 | 23.1 | 19.4 | 36.17 | 78.24 | 55.7 |
| 6.7-4.8 Ma | Average | 17.4 | 0.5 | 2.35 | 20.3 | 9.1 | 1.33 |
| | CV | 29.31 | 14.6 | 21.3 | 21.93 | 72.79 | 47.55 |

---

## Author Comment (AC4) · 12 Sep 2018

**Dear reviewer,**

We would like thank you for having read and commented our manuscript and we would like to apologize for the delay in our answers. We are grateful for your questions and suggestions. It's very useful and enlightening. We will take consideration in the revised version. Here, we provide some quick replies to your questions.

**Major concerns:**

1) The introduction part is not well written as there are many ambiguity and in accuracy (see detailed comments below). This section needs significant reworking.

2) The authors seem to preferentially pick 4.8 Ma as the boundary between the two climate intervals. However, most of the proxies exhibited in Fig. 3 seem has a distinct change at 4.6 Ma, but not 4.8 Ma, e.g., Al2O3, K2O, as well as the three magnetic susceptibility plots. In addition, for the grain size and carbonate content plots, there is no apparent difference below and above 4.8 Ma.

3) This manuscript is generally good written in English, but additional efforts are required to polish the language.

**Our response:** Thank you for your suggestions. We would take consideration in the revised version. We would explain how we defined the boundary later.

**Line-to-line comments:**

L1-2: I found the title is kind of misleading. The authors emphasize the Tibetan Plateau as the location of their section. However, throughout the manuscript, the Xiaoshuizi section is compared with other sections on the Chinese Loess Plateau, and reflects nothing of the Tibetan Plateau evolution. So it would be more appropriate to emphasize the location as the "western Chinese Loess Plateau".

**Our response:** It's a very good question. The reason why we emphasize the NE Tibetan Plateau is that it significantly represents the location of the Xiaoshuizi planation surface. It locates the transition zone from the Tibetan Plateau to the Chinese Loess Plateau. The obvious difference between Xiaoshuizi deposit and the red clay in the Chinese Loess Plateau is the modern altitude, and this exactly results from the special geographical position of NE Tibetan Plateau. Compared with other sections on the Chinese Loess Plateau is the main method to address the evolution of palaeo-circulation over the Xiaoshuizi planation from late Miocene to early Pliocene. We would modify statement of lines 108-114 as "Clarification of the evolution and dynamics of the westerlies and the palaeo-ASM requires accurate paleoclimatic reconstructions in the East Asia especially in the NE Tibetan Plateau, which is at the junction of the westerlies and monsoonal influences.

Continuous red clay sequence was recently discovered on the uplifted Xiaoshuizi (XSZ) peneplain in the NE Tibetan Plateau and has been dated via high-resolution magnetostratigraphy analysis (Li et al., 2017)."

L49: one of the most intensively studied intervals of what? Climate I assume? Our response: We would add "in climate research" at the end of the sentence.

L51: in line 41, the authors state closure of the Panamanian Seaway at 4.8 Ma, and it seems that the seaway closure has significant climate effects. Thus, it would be inappropriate to state here that the Zanclean is similar as present due to similar land-sea distribution. Our response: We would remove the statement.

L52-53: references for "comparable temperatures in the tropical region" need to be added.

Our response: We would add the citations (Herbert et al., 2010, 2016).

**L54: Zanclean is a period from cold to warm?**

**Our response:** Zanclean (5.3-3.6 Ma) is a transition period from relative cold climate of Late Miocene to relative warm climate of late Pliocene (3.6-3 Ma).

**L66-67: wired transition from the previous sentence. Not consistent.**

**Our response:** We would modify it as "whether permanent El Nino–like conditions were sustained during the Pliocene is still controversial."

L68: This is at least not accurate, if not wrong. Numerous studies have demonstrated that surface uplift of the Tibetan Plateau is stepwise and spatially diachronous. See reviews of Tapponnier et al., 2001, Wang et al., 2014 and many others. The south-central parts of the Tibetan Plateau were uplifted much earlier than the Zanclean, e.g., Paleogene. In the northern Tibetan Plateau, although there might be tectonic deformation in the margins of the Plateau (Li et al., 2015), the major part of the northern Plateau is probably uplifted during the Miocene, as evidenced by numerous other evidence, see review of Yuan et al., 2012. While it's OK to stick on the authors' own preference, it's necessary to discuss/reflect other research progress.

**Our response:** Thank you for point it out. What we would like to express is the uplift of the Tibetan Plateau was underway. It may be inappropriate. We would modify the statement of lines 68-72 as "In addition, several major changes in global thermohaline and atmospheric circulation system occurred during the early Pliocene which are thought to be crucial preconditions for both the appearance of Northern Hemisphere ice sheets at ~2.6 Ma (Haug et al., 1998, 2005; Driscoll et al., 1998) and the development of the modern east-west hydrographic gradient in the equatorial Pacific (Lawrence et al., 2006; Chaisson et al., 2000)."

**L72: 3 Ma or 2.6 Ma? Be accurate.**

Our response: We would correct it.

L75: first appearance of ASM in the main text, need to define first. In addition, for summer monsoons in Asia, there is the East Asia Summer Monsoon and the South Asia (India) Summer Monsoon. Which one do you mean? I assume East Asia Summer Monsoon?

**Our response:** Actually, South Asia Summer Monsoon also put impact on the XSZ. However, the impact is not as significant as that put by East Asia summer Monsoon. We would consider modifying it.

L76-77: In the abstract, the authors consider the ASM as moisture carrying, but the Westerlies as moisture lacking. So it's not appropriate to list them together. In addition, moisture transport is short-time climate condition, how could it cause long-tern glaciation?

Our response: Thank you for point it out, we would modify it.

L82: "warm and wet" climate yield "wet" climate? Definitely!

Our response: We would modify it.

L84: a weakened summer monsoon of where? Globally or East Asia only?

**Our response:** It is East Asia Summer Monsoon. We would modify lines 75-88 as "The Asian summer monsoon (ASM) and the meridional (westerlies) circulation systems, as major components of the atmospheric circulation, delivered moisture to Eurasia. The onset and strength of the Asian monsoon during the early Pliocene was linked to the uplift of the Tibetan Plateau (TP), changes in latitudinal and longitudinal heat gradients, global temperature and ice volume (An et al., 2001; Ding et al., 2001; Li et al., 2008, 2010; Clift et al., 2008; Nie et al., 2014; Ao et al., 2016). Therefore, determining the range of climatic conditions across the East Asia during the Pliocene not only improves our understanding of the regional climate, but it can also provide insights into the palaeo-monsoon, and thus into changes in the global climate system at this time".

L89-91: onset of interior Asian aridification since the late Miocene? This is totally unjustified. Numerous studies indicate much earlier onset of Asian interior aridification, e.g., since 22 Ma (Guo et al., 2002), or much earlier at Eocene-Oligocene transition (Dupont-Nivet et al., 2007), or late Eocene (Bosboom et al., 2014).

Our response: We would modify the "onset" as the "enhancement".

**L103-105: is this phenomenon also observed in other studies?**

**Our response:** Yes, pollen assemblages at Chaona indicate a considerably warmer and more humid climate from 4.61-4.07 Ma which was not consistent with that the low magnetic susceptibility values (Ma et al., 2005).

L107-108: what inconsistent? Need to clarify. For the evidence listed above, it's necessary to point out which region is dominated by westerlies, which is dominated by ASM.

**Our response:** We would add "where climate is dominated by westerlies," after "Tarim basin" in line 92 and "where climate is dominated by East Asia Monsoon," after "In eastern and central CLP," in line 94.

**L110: why the western CLP is especially important? Need to give reasons here.**

**Our response:** We would modify it as "Clarification of the evolution and dynamics of the westerlies and the palaeo-ASM requires accurate paleoclimatic reconstructions in the East Asia especially in the NE Tibetan Plateau, which is at the junction of the westerlies and monsoonal influences."

**L126-127: rejuvenated at what time?**

**Our response:** The tectonic movement of the Pre-Sinian formed the inverse anticlinorium and several faults arranged as en echelon (or parallel) pattern, which lay the foundation for outline of Maxianshan (Li et al., 1990). The planation surface research indicates the tectonic movement since late Miocene made the area uplift with a great amplitude (Li et al., 2017, Ma et al., 2016).

L175-182: what are the criteria to divide the carbonate content plot into 6.7-4.8 Ma and 4.8-3.6 Ma. I do not see apparent difference between these two subdivisions. For the 6.7-4.8 Ma interval, the carbonate content is 3.8-39.2. The 6.0-5.5 interval, with much smaller amplitude of fluctuation, seems to be more different from the other time intervals.

**Our response:** Figure 3 may not show the distinct of two periods clearly enough due to variable deposition. Thus, we present figure of records versus stratigraphic depth (Fig 1s). Synthesized the values and variations of the carbonate content, elements content and magnetic susceptibility, a transition period is presented at 16.5-15 m (4.8-4.6 Ma). There is an obvious difference in the character of the fluctuations above and below the depth of 16.5-15 m. For example, above the 16.5 m, the carbonate content fluctuates at a lower level but with larger amplitude accompanied by the noted increase in nodule horizons underlaying leached zones in the field, and the magnetic susceptibility also fluctuates at greater amplitude. Meanwhile, a noticeable drop in deposition rate around 4.8 Ma occurred. Thus, we define the 4.8 Ma as the transition point.

The average value and coefficient of variation of the records during two periods (6.7-4.8 Ma and 4.8-3.6 Ma) have been given in Tables 1s. The coefficient of variation (CV) is defined as:

$$CV = 100 * \frac{\text{Standard deviation}}{\text{Mean}}$$

The higher the CV, the more changeable the record. It shows the average value and CV of the most records show the obvious difference between two periods and most of the records are more changeable during 4.8-3.6 Ma than 6.7-4.8 Ma.

We would explain it in revised version.

L185-187: looking at Fig. 3, it's pretty hard to determine whether two plots are of similar trend, or opposite trend. I would suggest to provide statistical evaluation to help readers understand the similarity between plots.

Our response: Would add table 2 in the revised version.

L188: provide the ranges of  $Al_2O_3$  and  $K_2O$  for the two time intervals. L189: why choose 4.8 Ma as the boundary? The values between 4.8-4.6 seem be more similar as the 6.7-4.8 Ma interval. Our response: Fig1s shows 16.5-15m (corresponding to 4.8-4.6 Ma) is a transition period. During this period,  $K_2O$  and Rb increased while CaO,  $Na_2O$  and Sr decreased. The transition period means climate translates from one condition to another. Meanwhile, a noticeable drop in deposition rate around 4.8 Ma occurred. Thus, we define the start of the transition period (4.8 Ma) as the transition point.

We would modify the statement of lines 184-191 as "The XSZ red clay consists mainly of SiO2, Al2O3, CaO and Fe2O3 with low concentrations (<5%) of MgO, K2O, Na2O, Sr, Rb and Ba (Table 1). During *Interval I*, K2O ranges from 1.9-3.3% with an average content of 2.6%. Na2O ranges from 0.14-1.52% with an average content of 1.2%. Rb ranges from 80-125 ppm with an average content of 103.9 ppm. Sr ranges from 150-252 ppm with an average content of 211.7 ppm. During *Interval II*, K2O ranges from 2-3.7% with an average content of 3%. Na2O ranges from 0.94-1.54 % with an average content of 1.23%. Rb ranges from 74-134 ppm with an average content of 109.9 ppm. Sr ranges from 141-281 ppm with an average content of 214.6 ppm. The variations in CaO show the same trend as carbonate content. The variations of Rb and K2O are synchronous and roughly opposite to that of CaO. The changes of Sr show some similarity with CaO. Accordingly, table 2 indicates CaO has positive correlation with CaCO3 and Sr while negative correlation with other elements. The variations in CaO, K2O, Sr and Rb content during 4.8-3.6 Ma are greater than those during 6.7-4.8 Ma, which is also indicated by the CV of these elements (Table 1)."

L193: similar question, why group values between 6.7-4.8 together, but not include values between 4.8-4.6, which exhibit more similarities as the 6.7-4.8 Ma interval, which are of lower values and smaller amplitude of variation.

Our response: Fig1s shows magnetic susceptibility start to increase from 16.5m (4.8 Ma).

**L204: there is no difference between these two intervals.**

**Our response:** >40 um content is a proxy for winter monsoon strength. <2um content partly adheres to coarser silt and sand particles. Thus, we do not take two contents into consideration here.

**L206: in Fig. 3, it shows >40 um.**

Our response: It's >40 um. We would correct it.

L251: I have a question here, maybe very basic in your discipline. If one wants to use  $K_2O/Na_2O$  values to determine the intensity of chemical weathering, a pre-assumption is that before weathering, all the samples have similar  $K_2O/Na_2O$  values. Right? How about if the original  $K_2O/Na_2O$  values are different? This question might also exist for other chemical proxies used here.

**Our response:** It's really a good question. Previous studies have indicated the rare earth element distribution patterns of both the loess and red clay are identical to those of upper continental crust (Ding et al., 2001; Jahn et al., 2001). It suggested the initial geochemical composition of well mixed loess and red clay is similar. Thus, changes in  $K_2O/Na_2O$  and Rb/Sr ratios are mainly determined by post-depositional weathering.

L287-288: Could you please explain this in more detail? Which feature in Fig. 5d denotes orbital signal increase since 4.8 Ma? As far as I can infer from Fig. 5d, in the carbonate content plot, the orbital parameters increase since 4.9-5.0 Ma. While, in the Xpedo plot, it seems the increasing timings are diachronous for different orbital parameters.

**Our response:** Thank you for pointing it out. The explanation for plot Fig 5d may be not appropriate. We consider removing the wavelet plots and the statement "Furthermore, Morlet wavelet transform analysis of both carbonate content and  $\chi$ pedo show that the orbital signal increases since 4.8 Ma (Fig. 5 d)" in lines 286-288.

L292-295: Here the authors propose that the carbonate content and Xpedo signals reflect incomplete preservation of paleoclimate signals. Then the question is if the original paleoclimate signals are incomplete, how would you use these records to predict paleoclimate changes?

**Our response:** It involves scale problem. These records cannot document the millennial-scale climate change completely but can document paleoclimate changes in ten thousand scale. The reason why we said the signals are incomplete is that 20% of sample intervals are not satisfied with the requirement for detecting the precession signal (with the resolution 4 kyr or less).

L308-312: According to the authors' statement, the rapid change from 6.7-4.8 Ma low amplitude to 4.8-4.1 Ma large amplitude is observed in all the three orbital parameters. But for the benthic foraminiferal d18O record, similar change is only observed in the 41-kyr component. Why? This does not read like strong evidence to infer that the wet-dry oscillations were driven by changes in ice volume or global temperature. An associated question would be if the authors do not consider solar radiation intensity is the cause of the wet-dry cycles, but ice volume or global temperature, then what's the cause of ice volume and global temperature changes? Isn't solar radiation intensity a driving factor?

**Our response:** "This may mean that the increased contrast in wet-dry oscillations at the XSZ site was not driven directly by changes in solar radiation intensity but rather was linked with changes in ice volume or global temperature." (lines 310-312). Sorry, the sentence is an ambiguous or misleading

expression. It's the enhancement for wet-dry contrast not wet-dry oscillation, which was linked with changes in ice volume or global temperature. We would correct it. Changes of Earth orbital parameters (linked to solar radiation intensity) would dynamically lead to the variation of the climate including changes in global temperature, ice volume and also wet-dry cycles of Xiaoshuizi. However, the change of Earth orbital parameters is just one of forcing factors and some internal process or feedback could magnify or cover the orbital forcing, which means the climate changes probably show non-linear response to orbital forcing (solar radiation intensity). In this specific case, it might be the expansion of the palaeo-ASM that enhanced the orbital periodic signals of XSZ red clay between 4.8 and 4.1 Ma.

In addition, glacial–interglacial climate cycles show different response to three orbital periodic changes in different time periods. For example, the LR04 benthic stack exhibits significant coherency with insolation in the obliquity band throughout the entire 5.3 Ma (Lisiecki, 2005) and glacial–interglacial cycles was dominated by 100-kyr eccentricity band approximately 0.8 Ma ago (Lisiecki et al., 2010). Thus, the change is only observed in the 41-kyr component of benthic foraminiferal  $\delta^{18}$ O record is reasonable. Furthermore, in our records, the most significant change is also observed in 41kyr filtered components. The remarkably increased amplitude of the 41kyr filtered components from XSZ and the deep sea  $\delta^{18}$ O record at about 4.8 Ma indicating common changes may have synchronously amplified the response of  $\delta^{18}$ O record and XSZ wet-dry oscillations to obliquity forcing. The variation of deep sea  $\delta^{18}$ O related to the changes in global temperature and ice volume (Zachos et al., 2001). The increased wet-dry oscillation of XSZ related to the expansion of palaeo-ASM. Thus, the expansion of palaeo-ASM may be related to changes in global temperature and ice volume. It is not the evidence but just one possibility.

L317-318: I find this conclusion hard to believe. For the carbonate content signal, the authors state that they record incomplete paleoclimate signal (see comments for L292-295). For the K2O/Na2O and Rb/Sr record, a more apparent change seems to be at 4.6 Ma. If higher carbonate content represents dry climate, and lower carbonate content representshumid climate, compared with 6.7-4.8 Ma, the 4.8-3.6 Ma would have more humid period, but also much drier period, because the 4.8-3.6 Ma has larger variability. While, I did not see a clear wetting trend. Our response: Firstly, as mentioned above, paleoclimate signal is incomplete in millennial scale. In ten thousand scale, it is complete. Secondly, carbonate content averaged with 13.8% during the interval of 4.8-3.6 Ma, which indicate climate became wetter after 4.8 Ma. In detail, during the interval of 4.8-3.6 Ma, the carbonate content fluctuates with greater amplitude but at a lower level and leached horizons is thicker than the interval of 6.7-4.8 Ma, which means leaching process enhanced after 4.8 Ma. Thirdly, the similar phenomenon was also observed in Xijin loess-paleosol series that in loess layers carbonate has a high average content with small fluctuation amplitude while in paleosol layers carbonate has a low average content with large fluctuation amplitude (Ye, 2017).

L363-364: This is a false statement. Even at present, the Tibetan Plateau cannot block the

**Westerlies completely. The Westerlies can travel to the northeastern Tibet through valleys in the Tianshan.**

Our response: Thank you for pointing it out. We would consider removing it.

**L368: which plots are pedogenesis proxies? Cite the specific plots here. "roughly"? how rough? Better to give a quantitative value.**

**Our response:** We would add " $\chi_{pedo}$  shows a significant positive relationship with  $\delta^{18}$ O at 80 % confidence (Fig. 4f)" after "pedogenesis proxies roughly parallel to the stacked deep- sea benthic foraminiferal oxygen isotope curve" of lines 368-369. We would also add the fig 4f

**L391-392: is there evidence to suggest reduced amount of atmospheric water vapor? Weakening of the paleo-ASM and dominance of Westerlies can explain the aridity. This does not necessarily need reduced amount of atmospheric water vapor.**

**Our response:** We would modify the statement of lines 391-393 as "However, moisture sources for the westerly flow are distant from the CLP (Nie et al., 2014), and only a relatively small amount of moisture was carried to the CLP, resulting in a dry and stable climate in the XSZ region."

**L469: "extremely wet"? wetter than any other period?**

**Our response:** The expression is inappropriate. We would modify "As mentioned above, the extremely wet climate across the CLP was synchronous with the gradual closure of the Panama Seaway, which led to a larger reorganization of the global thermohaline circulation pattern" as "The occurrence of humid climate across the CLP was synchronous with the gradual closure of the Panama Seaway (Jackson and O'Dea, 2013; O'Dea et al., 2016)."

**L528-530: I probably missed it, but how could your records reflect seasonality of precipitation? Which proxy records seasonal signals?**

**Our response:** Research on migration process of carbonate indicated seasonally wet/dry climate is a key factor in driving carbonate dissolution and reprecipitation, and strong seasonally biased precipitation enhances the leaching process and produces thick leached horizons (Rossinsky and Swart, 1993; Zhao, 1995, 1998). The emergence of high-frequency cycles of carbonate eluviation-redeposition and thicker leached horizons indicates that seasonal precipitation increased during the interval of 4.8-3.6 Ma.

**L532: why the strongest summer monsoon is between 4.6-4.25 Ma? What are the possible reasons for the decreasing strength after 4.25 Ma?**

**Our response:** From 4.60-4.25 Ma, pedogenesis and weathering intensity of XSZ reached a maximum, as did precipitation intensity, which is manifested by the enhanced eluviation and carbonate accumulation. These evidences indicate the strength of summer monsoon is strongest from 4.60-4.25 Ma. Meanwhile, the temperatures of both high northern latitude and subtropic Pacific were relatively

high, which may be responsible for enhancement of the paleo-ASM system. On the other hand, input of ice raft debris into subarctic northwest Pacific increased from 4.25 to 4.1 Ma which indicates temperature of the high northern latitudes decreased. Temperature at subtropic Pacific also decreased after 4.0 Ma. The decreased temperature of high northern latitude and subtropic Pacific may be the reason for the decreasing strength of palaeo-ASM after 4.25 Ma (Fig. 6).

**Figures:**

Fig. 1: a, the present outline is too large, the wind vectors are too small to see. It's better to show a smaller region with more details; e.g., regions between 10N-50N, 70E-130E. c. highlights the Xiaoshuizi section. Hard to find now.

Our response: Thank you for suggestions. We would modify it. (Fig 1)

**Fig. 2: These photos exhibit very few useful information.**

**Our response**: These photos visually exhibit information about Xiaoshuizi planation surface and red clay.

Fig. 3: Between 4.8-4.6 Ma, most plots show a weird shape. Is this because there are limited samples compared with other time intervals?

Our response: Yes, it is. We would use figure1s to replace the figure 3

Fig. 5: apparently the authors need to provide more information in the caption about their plots. For example, Fig. 5d, what does the color mean? What does the black curve represent? Also, the horizontal age scale is better to use Ma, but not ka, as Ma is used throughout the manuscript. In Fig. 5a-b, there are other strong periodicities denoted. How about these periodicities in Fig. 6d? Our response: We would modify it. (Fig. 5)

Fig. 6. It will be better to arrange all the proxies with the same logic, e.g., left-wet, right-dry. Our response: The reason why we put the benthic foraminiferal  $\delta^{18}$ O record with different logic from other records is that we would like to show the similarity of benthic foraminiferal  $\delta^{18}$ O record and  $\chi_{pedo}$  curve during 6.7 to 4.8 Ma.

**References**

- Ding, Z. L., Sun, J. M., Yang, S. L., Liu, T. S. (2001). Geochemistry of the pliocene red clay formation in the chinese loess plateau and implications for its origin, source provenance and paleoclimate change. Geochimica Et Cosmochimica Acta, 65(6), 901-913.
- Jahn, B. M., Gallet, S., & Han, J. (2001). Geochemistry of the xining, xifeng and jixian sections, loess plateau of china: eolian dust provenance and paleosol evolution during the last 140 ka. Chemical Geology, 178(1), 71-94.
- Li, Z. Q., Liu, Q. H., Sun, B. J.(1990). Germorphic features and types of Xinglong mountians. Journal

of Gansu Agricultural University, 25(3), 303-312.

- Lisiecki, L. E., & Raymo, M. E. (2005). A pliocene pleistocene stack of 57 globally distributed benthic  $\delta 180$  records. Paleoceanography, 20(2), -.
- Lisiecki, L. E. (2010). Links between eccentricity forcing and the 100,000-year glacial cycle. Nature Geoscience, 3(5), 349-352.
- Ma Yuzhen, Fuli, FANG, Xiaomin, Jijun, & Zhisheng, et al. (2005). Pollen record from red clay sequence in the central loess plateau between 8.10 and 2.60 ma. Chinese Science Bulletin, 50(19), 2234-2243.
- Ma Zhenhua(2016). The planation surfaces and their Late Cenozoic geomorphological evolution in Maxianshan Mountains, NE Tibetan Plateau. Lanzhou: Lanzhou University Masters dissertation.
- Rossinsky Jr., V., Swart, P.K., 1993. Influence of climate on the formation and isotopic composition of calcretes. In: Swart, P.K., Lohmann, K.C., McKenzie, J., Savin, S. (Eds.), Climate Change in Continental Isotopic Records, American GeophysicalUnion: Geophysical Monography, 78, pp. 67-75.
- Ye Xiyan (2017). The time series establishment and paleoclimatic period evolution of Xijin Loess. Lanzhou: Lanzhou University Masters dissertation.
- Zachos, J., Pagani, M., Sloan, L., Thomas, E., & Billups, K. (2001).Trends, rhythms, and aberrations in global climate 65 ma to present.Science, 292(5517), 686-93.
- Zhao, J. B. (1995), A study of the CaCO3 illuvial horizons of paleosols and permeated pattern far rain water, J Geogr Sci, 15(4), 344-350.
- Zhao, J. B. (1998), Illuvial CaCO3 layers of paleosol in loess and its environmental significance, Journal of Xi'an Engineering University, 20(3), 46-49.

**Figures and tables**

---

## Author Response (AR1)

Dear, editor Ran Feng,

We have carefully revised and edited the manuscript entitled " Late Miocene-Pliocene climate evolution recorded by the red clay covered on the Xiaoshuizi planation surface, NE Tibetan Plateau" based on the valuable comments and suggestions from three anonymous reviewers. Below please find the detailed responses. In addition, other major modifications are also listed.

**Responses to comment 1**

**I think the manuscript needs to present the geochemistry data versus stratigraphic depth, in addition to just age. There also needs to be more discussion on the relationship between sedimentation rate and pedogenesis. For example, it would be helpful if Figure 3 was plotted vs. depth and there was also a column that plots sedimentation rate, and the presence of nodule horizons. This is important because the interval between 4.5 and 4.3 Ma, for example, shows a strong increase in magnetic evidence for pedogenesis and also coincides with a noticeable drop in deposition rate. Therefore, it needs to be discussed if this increase in pedogenesis was driven solely by wetter conditions, or was there also more time for soil formation and leaching of Ca. I do think more stratigraphic context will help some of the arguments presented in the text. For example, upon my initial reading of the text and figure, the division into the 2 primary intervals placed at 4.8 Ma seemed somewhat arbitrary looking at figure 3 (i.e. why not 4.6 or 5.1). But it makes much more sense in terms of the large decline in sedimentation rate around 4.8, which accompanied by the deposition of a carbonate nodule layer, and then the noted increase up-section in nodule horizons underlaying leached zones. Also with deposition of loess being connected to regional wind patterns, is it significant that there was a notable _200 kyr drop in sedimentation rates before a shift to generally wetter/more seasonal conditions?**

**Our response:** Many thanks for your valuable suggestions. We would take these suggestions and use new figure to replace the Fig 3, and we would add a brief statement "Profiles of the various proxies are illustrated in Fig 3 and there is an obvious difference in the character of the fluctuations above and below the depth of 16.5 m (~4.8 Ma). Above 16.5 m, the carbonate content fluctuates at a lower level but with greater amplitude, and the magnetic susceptibility also fluctuates at a greater amplitude. In

addition, the CV of most of the records is greater above the boundary than below (Table 1). This

suggests that the climate became more humid and variable after 4.8 Ma. Meanwhile, a noticeable drop

in deposition rate around 4.8 Ma occurred (Li et al., 2017). Thus, the red clay sequence was divided

into two intervals: *Interval* Ⅰ (6.7-4.8 Ma) and *Interval* Ⅱ *(*4.8-3.6 Ma). The characteristics of the

individual proxy records are describe in detail below" in front of line 174. We will also add "We use

the coefficient of variation (CV) to measure the variability of the records. The higher the CV, the more

variable the record. The CV is defined as: $CV = 100 * \frac{Standard\ deviation}{Mean}$," at the end of chapter 3 (line

172).

**I am somewhat confused by the explanation of K/Al ratios as a weathering proxy (lines
238-245). With time, Al can mobilize and become depleted at the top of a paleosol and
enriched down profile. And in certain situations, you might expect K to be enriched at
the soil surface, due to its biological importance. So, within the same well developed
soil, you might expect a higher K/Al ratio at the top, and a lower ratio deeper in the
profile. This is never plotted, so it might be worth eliminating this text?**
**The various magnetic susceptibility terms are well described in the discussion, but I
think it would help readers if at least some of this information was moved up to either
the results or methods. This would help provide context to all of the values presented
in the results.**

**Our response:** We would consider removing the $K_2O/Al_2O_3$ ratio and modify the statement "In

addition, previous…" in lines 238-252 as "In addition, the $K_2O/Na_2O$ ratio is used to evaluate the clay

content in loess and is also a measure of plagioclase weathering, avoiding biases due to uncertainties in

separating carbonate Ca from silicate Ca (Liu et al., 1993; Buggle et al., 2011). $Na_2O$ is mainly

produced by plagioclase weathering and is easily lost during leaching as precipitation increases. By

contrast, $K_2O$ (mainly produced by the weathering of potash feldspar) is easily leached from primary

minerals and is then absorbed by secondary clay minerals with ongoing weathering (Yang et al., 2006;

Liang et al., 2013). In the arid and semi-arid regions of Asia, $K_2O$ is enriched in palaeosols compared

to loess horizons (Yang et al., 2006). Thus, high $K_2O/Na_2O$ ratios are indicative of intense chemical

weathering. " We would also remove "The $K_2O/Al_2O_3$ ratio also increased rapidly at about 4.8-4.7 Ma

and maintained relatively high values after 4.7 Ma. This may indicate that the overall weathering intensity was sufficient to produce secondary clays, resulting in a spike in $K_2O$ concentration" in lines 409-312. In lines 287-288, "Morlet wavelet transform analysis of both carbonate content and $\chi_{pedo}$ show that the orbital signal increases since 4.8 Ma (Fig. 5 d)."

**Minor suggestions:**

**Line 57: suggest "occurring" or "underway" instead of "ongoing"**

**Our response:** We would modify "ongoing" as "occurring".

**Line 76: suggest "supplied" instead of "prepared"**

**Our response:** Thank you for suggestion. We consider remove the sentence.

**Lines 75-88: I'm guessing the sentence beginning with "Make clear…" on line 78 was accidentally left in as a comment, which I still think needs to be addressed. I think I understand what the authors are going for within the paragraph, but I think the logic can be expressed more clearly. The strength/onset of the Asian monsoon is linked to these globally significant events (Tibetan uplift, northern hemisphere ice, etc). Therefore, by constraining paleoclimate across the Chinese Loess Plateau not only does this improve our understanding of regional climate, but it can also provide insight about the paleomonsoon, and therefore changes in the global climate system during the Pliocene.**

**Our response:** Many thanks for your suggestions. We would modify statement as "East Asia is one of the key regions for studying the aridification of the Asian interior and the Asian monsoon evolution which is tightly linked to the uplift of the TP, the regional climate change and global temperature and ice volume evolution (An et al., 2001; Ding et al., 2001; Li et al., 2008; Clift et al., 2008; Nie et al., 2014; Ao et al., 2016;  Sun et al., 2006a, 2017; Chang et al., 2013; Liu et al., 2014)" and add "Therefore, determining the climatic conditions of the NE TP during the early Pliocene not only improves our understanding of the regional climate change, but also provides insights into the responses of the palaeo-EASM and westerlies to TP uplift and changes in the global climate system at this time." before line 111.

**Line 96: suggest removing "condition" and changing "aridification process" to "regional aridification"**

**Our response:** We would modify " a dry climate condition" as "dry climatic conditions" and modify "however, aridification process was interrupted by a long interval of wet climate" as "but generally wet climatic conditions".

**Line 104: change "to be" to "that", and I think it would be helpful for the future readers not just to say "gleying", but instead state briefly what that means (waterlogging, and iron reduction) and why it matters for the magnetic susceptibility record.**

**Our response:** We would modify "It's thought to be substantial gleying resulted from large amount precipitation which made magnetic susceptibility invalid over this period" was replaced as "It is thought that waterlogging and iron reduction resulting from high precipitation significantly affected the climatic significance of magnetic susceptibility records during this period".

**Line 105-106: This sentence does not make sense. Are you trying to say that climate in this region is influenced by the strength of both the westerlies and the monsoon, and that those two factors may not be directly related?**

**Our response:** We would remove the statement.

**Lines 114-115: What makes the XSZ red clay different geomorphologically?**

**Our response:** The Xiaoshuizi peneplain of the Maxian mountain occupies a critical transition position between the high-altitude TP and the low North China Craton (Li et al., 2017). The obvious difference between Xiaoshuizi deposit and the red clay in the Chinese Loess Plateau is the modern altitude, and this exactly results from the special geographical position of NE Tibetan Plateau.

**Line 118: suggest "are" instead of "have been"**

**Our response:** We would modify "have been" as "are"

**Line 121: This sentence is slightly off.**

**Our response:** We would modify it as "Finally, the regional climate evolution and its possible mechanisms have been further discussed."

**Line 133: suggest "reconstruct and discuss" instead of "discuss "**

**Our response:** We would modify "discuss" as "reconstruct and discuss"

**Line 133-134: not sure exactly what is meant here. Is the XSZ core characterized by more continuous deposition and records a longer time interval than the Shangyantan core?**

**Our response:** Yes, SYT core is only covered the age from 6.4-4.2 Ma.

**Line 136: capitalize China**

**Our response:** We would modify it.

**Lines 137-138: Not sure what is mean by the sentence beginning with "The East Asian Monsoon." Are you trying to explain how these two factors together control climate at the study site. This could be elaborated.**

**Our response:** The means of this sentence is similar to statement of lines 127-128. We would remove it.

**Line 144: Where in the section is the increase in gravel? From the strat column it looks like it is at the base. Say this in-text.**

**Our response:** We would add "at the base (Fig. 1b)" after "…gravel content" in line 144.

**Lines 145-147: Clarify if most carbonate horizon are overlain by a brownish red-layer, or if the carbonate zone in its entirety underlies a larger brownish-red layer. Lines 148-150: It's not clear as written if carbonized root channels have more abundant Fe-Mn staining.**

**Our response:** We would modify "is impregnated with" as "contains numerous" and modify "horizons containing" in line149 as "the".

**Line 168: Is all of the remaining Ca in silicate minerals? Won't a lot of it be loosely bound to clay minerals in the soils? Also, the correction for Phosphorous also needs to be explained. I'm guessing you are assuming some component of Ca-bearing phosphate minerals, but what is the basis for this assumption.**

**Our response**: Thanks for your questions and suggestions. No, not all of remaining Ca in silicate minerals and the Ca bound to clay minerals is also included. Silicate-bound CaO* is obtained, in theory, by the simple equation (Fedo et al., 1995): CaO*(mol) = CaO(mol) $-CO_2$(calcite mol) $-$ 0.5 $CO_2$(dolomite mol) $-$ 10/3 $P_2O_5$ (apatite mol). It generally calculated based the assumption that all $P_2O_5$ is associated with apatite and all inorganic carbon is associated with carbonates. It may neglect the Ca bound to clay minerals and overestimate the component of Ca-bearing phosphate minerals (Garzanti and Resentini., 2016). The reason we use the equation to calculate the values is that we try to expel the possibility the variation of Sr is determined by the bound of secondary carbonate, but not by weathering intensity. For Sr can substitute Ca in secondary carbonates (Reeder et al., 2006; Buggle et al., 2011). We will modify statement "The molar content of silicate Ca (CaO*) was calculated using the following equation:" as "Silicate-bound CaO (CaO*) can be estimated, in principle, by the equation: CaO*(mol) = CaO(mol) $-$ $CO_2$(calcite mol) $-$ 0.5 $CO_2$(dolomite mol) $-$ 10/3 $P_2O_5$(apatite mol) (Fedo et al., 1995). It is generally calculated based on the assumption that all the $P_2O_5$ is associated with apatite and all the inorganic carbon is associated with carbonates Thus, the CaO* of the XSZ red clay was calculated using the following equivalent equation" .

**Line 199: What do you mean by durations? Are you saying there are some thicker intervals of high magnetic susceptibility?**

**Our response:** Yes, it means the interval of strong pedogenesis sustained longer.

**Line 256: space between "susceptibility" and "of"**

**Our response:** We would correct it.

**Line 257: suggest removing "two"**

**Our response:** We would remove "two".

**Line 314: Spelling of "Multiproxy"**

**Our response:** We would modify it.

**Line 317-318: suggest "a significant change is recorded by most of the proxies that**

**occurred"**

**Our response:** We would modify "we observe that a significant change recorded by the most of the multiproxy (carbonate, Rb/Sr, $K_2O/Al_2O_3$, $\chi_{pedo}$) occurred near 4.8-4.7 Ma" as "there is a significant change in most of the proxies (carbonate, Rb/Sr, $K_2O/Na_2O$ and $\chi_{pedo}$) near 4.8 Ma".

**Line 318: K/Al is not plotted, but K/Na is plotted. Based on the comment above, I think this is probably a better choice.**

**Our response:** We would modify "$K_2O/Al_2O_3$" as "$K_2O/Na_2O$".

**Line 327: suggest "relatively" instead of "relative" and "and" instead of "which"**

**Our response:** We would modify "relative" as "relatively" and modify "which" as "and".

**Line 328: Not sure what this sentence is trying to say.**

**Our response:** This sentence may be redundancy. We would remove "when these proxies detailed climate changes especially when climate is relative wet".

**Line 329: I suggest clarifying the beginning of this sentence to say something along the lines of "Carbonate content becomes more variable after 5.5 Ma, which is…"**

**Our response:** We would modify the sentence "It is evident that the carbonate content decreases with increased variation amplitude after 5.5 Ma" as "Carbonate content becomes more variable after 5.5 Ma, which is…"

**Line 333: spelling of "indices"**

**Our response:** We would correct it.

**Line 345: suggest "central and eastern" instead of "hinterland of the"**

**Our response:** We would modify "hinterland of the" as "central and eastern".

**Line 377: suggest rewording the sentence beginning with: "Look around the globe,.."**

**Our response:** We would modify the sentence of lines 376-377 as "A sustained cooling occurred in both hemispheres during late Miocene and the cooling culminated between 7 and 5.4 Ma (Herbert et al.,

2016)."

**Line 415: I'm not sure what "humid toward arid direction" means**

**Our response:** It means climate tended to become dry. We would modify the sentence as "During 3.9-3.6 Ma, precipitation decreased, and weathering and pedogenic intensity also weakened".

**Line 521: suggest "provides the opportunity constrain and discuss.."**

**Our response:** We would modify it as "provides the opportunity to elucidate the history of …"

**Line 526: again suggest "central and eastern" instead of "hinterland of the"**

**Our response:** We would modify "hinterland of the" as "central and eastern".

**Line 531: suggest removing "obviously"**

**Our response:** We would remove "obviously".

**Figure 1: I think it would help if you put a larger non-circle shape on panel A corresponding to the study site. Then you can remove the Xiashuizi label, which slightly obscures the vector. Then, match this symbol on panel C You are missing the white reversals between C3n.1n, C3n.2n, and C3n.3n on the Polarity plot for the XSZ section. These were included in the age model presented in Li et al. (2017). What do the black bars on the lithology column represent.**

**Our response:** Thank you for suggestions and pointing faults out. We have not noticed it in Fig. 1b. There is something wrong with this figure when we convert it into PDF format. Some thin white rectangles are missed. The black bars on the lithology column were the thin white rectangles representing the carbonate nodule layer. We would give the new figure (Fig 1).

**Figure 2: I think it would help if the line thicknesses were slightly thinner.**

**Our response:** You mean figure 3? We would modify it.

**Responses to comment 2**

**Spelling and Grammar:**

**I have not edited this manuscript for spelling and grammar. I strongly encourage the authors to seek assistance from a very proficient or native English speaker. Also, please review the manuscript for organizational mistakes (e.g. Figures 2 and 3 are not cited in the text; incorrect citations).**

**Our response:** Thanks for your suggestions. We would take consideration in the revised version and we would add"(Fig. 2)" after "yellowish clay layers" and add "(Fig. 2b)" after "…horizons" and cite Figure 3 in chapter 4(Results).

**Statistics:**

**The authors need to provide more information about the magnetostratigraphic ages. What is the temporal resolution of the records? What are the temporal uncertainties? Can the records accurately resolve all the cycles you discuss (e.g. precession)? Does variable deposition rate impact the signals?**

**Our response:** The average temporal resolution of records is 3.8 kyr. The resolution of records for detecting the precession signal needs to be 4 kyr or less (Luo et al., 2017). 80 % of sampling intervals satisfied the requirement with the resolution. Thus, the records can theoretically document the eccentricity and obliquity cycle of entire period and document the precession cycle of 80 % period. The variable deposition rate does impact on the conservation of the signals especially for precession signal.

**How did the authors decide that 4.8 Ma was the appropriate transition point? It seems arbitrary to me. I see no clear transition in Figure 3. Are the two periods (6.7-4.8 Ma and 4.8-3.6 Ma) statistically distinct?**

**Our response:** Figure 3 may not show the distinct of two periods clearly enough due to variable deposition. Thus, we present figure of records versus stratigraphic depth (Fig 1s). Synthesized the values and variations of the carbonate content, elements content and magnetic susceptibility, a transition period is presented at 16.5-15 m (4.8-4.6 Ma). There is an obvious difference in the character of the fluctuations above and below the depth of 16.5-15 m. For example, above the 16.5 m, the carbonate content fluctuates at a lower level but with larger amplitude accompanied by the noted

increase in nodule horizons underlaying leached zones in the field, and the magnetic susceptibility also fluctuates at greater amplitude. Meanwhile, a noticeable drop in deposition rate around 4.8 Ma occurred. Thus, we define the 4.8 Ma as the transition point.

The average value and coefficient of variation of the records during two periods (6.7-4.8 Ma and 4.8-3.6 Ma) have been given in Tables 1s. The coefficient of variation (CV) is defined as:

$$CV = 100 * \frac{\text{Standard deviation}}{\text{Mean}}$$

The higher the CV is, the more changeable the record is. It shows the average value and CV of the most records show the obvious difference between two periods and most of the records are more changeable during 4.8-3.6 Ma than 6.7-4.8 Ma.

**I find the signal filtering in Figure 5 questionable. First, the authors filter the data at frequencies with insignificant power (e.g. the 100 kyr filtering of carbonate content). Further, the most significant signals exist at frequencies that are difficult to explain, which the authors dismiss, and many of the discussed signals are barely significant at 90% confidence. The wavelet plots highlight the limited signal strength. Even if the filtered signals are sound, the filtered signals changes do not well align with the benthic d$^{18}$O record.**

**Our response:** The reason we filtered the carbonate at 100 kyr is that we observed that fluctuations of CaCO$_3$ and weathering indices agree well with eccentricity orbital variations at 4.8-3.9 Ma (Fig 1s-c). We perform the spectral analysis for carbonate content in two periods (6.7-4.8Ma and 4.8-3.6 Ma) respectively. It shows 100 kyr, 41 kyr and 21 kyr periodic signals of carbonate are significant in the period of 4.8-3.6 Ma (Fig 1s). However, most of the orbital periodic signals are insignificant in the period of 6.7-4.8 Ma.

As for non-orbital periodic signals, we have not found the driving force and the signals recorded by carbonate content are different from $\chi_{\text{pedo}}$. It may indicate non-orbital periodic signals are fake and more significant non-orbital periodic signals of carbonate content are relate to the dissolution-reprecipitation process of carbonate. Thus, we do not filter the records at these frequencies. We consider removing wavelet plots.

**Interpretation:**

**The potential drivers of the climate signals are often overstated. Connections are made with limited support. Many of the mechanisms discussed are still debated, particularly the Isthmus of Panama hypothesis and timing of Tibetan Plateau uplift. At the least, the authors need to do a better job citing recent literature and discussing the remaining uncertainties. Also, many of the citations are not primary sources for the associated statements.**

**Our response:** Thank you pointing it out. We would pay special attention to these problems in the revised version.

**Specific Comments:**

**Line 51: Earth's orbital went through many cycles over this period, so the "orbital configuration statement does not make much sense.**

**Our response:** We would remove the statement "(ii) orbital configuration".

**Lines 51-53: These statements, such as "comparable temperatures in the tropic region", require citations.**

**Our response:** We would add the citations "(Herbert et al., 2010, 2016)".

**Line 65: Please clarify the link between mean tropical Pacific east-west gradient and ENSO.**

**Our response:** We would modify the statement of line 65 as "…and low east-west sea surface temperature gradient in the tropical Pacific during this interval is believed to have given rise to a permanent El Nino Southern Oscillation."

**Line 68: The timing of uplift of the Tibetan Plateau is heavily debated…**

**Our response:** Sorry, the expression is misleading. What we would like to express is the uplift of the Tibetan Plateau was still underway.

**Line 70: Lunt et al. (2008) is not a direct source for the closure of the Isthmus of Panama. More recent works debate the timing of closure (e.g. Bacon et al., 2015;**

**O'Dea et al., 2016).**

**Our response:** Thank you for pointing it out. We would modify it as "(Keigwin et al., 1978; O'Dea et al., 2016)".

**Line 72: This statement is also not well supported. For example, Lunt et al. (2008), who are cited earlier, found closure of the Panama seaway to have little influence on NH glaciation. In general, the authors need to update their citations and discuss the Literature more thoroughly. These ideas are far from settled, yet they are presented as facts.**

**Our response:** Thank you for pointing these faults out. We would remove the statement

**Line 80-82: Citation?**

**Our response:** We would remove the statement.

**Line 85: "Arctic volume" means "Arctic ice volume"?**

**Our response:** Yes, we would correct it.

We would modify the statement of lines 68-88 as "Meanwhile, the episodic uplift of the TP (Li et al., 2015; Zheng et al., 2000; Fang et al., 2005a, 2005b) and gradual closing of the Panama seaway (Keigwin et al., 1978; O'Dea et al., 2016) were underway. The former substantially influenced the palaeoclimate change (An et al., 2001; Ding et al., 2001; Liu et al., 2014) and the later resulted in reorganization of the global thermohaline circulation (Haug et al., 1998, 2001)."

**Lines 105-106: This statement does not make sense.**

**Our response:** Thank you for pointing it out. We would remove the statement.

**Lines 136-140: Sources for these data?**

**Our response:** We would modify the statement of lines 136-138 as "The mean annual temperature during 1986-2016 was ~7.0 ℃ and the annual precipitation was 260-550 mm. Most (80%) of the precipitation is in summer and autumn. (The data were obtained from the National Meteorological Information Center (http://data.cma.cn/) of the Chinese Meteorological Administration (MCA)"

**Line 154: This requires more detail.**

**Our response:** We would remove the statement in lines 153-155 and add "Each sample age was estimated using linear interpolation to derive absolute ages, constrained by our previous magnetostratigraphic study (Fig. 1b). The average temporal resolution of the records is 3.8 kyr. Some 80 % of the sequence has a sampling resolution of 4 kyr or less" at the end of the chapter 3.

**Line 175-182: How did you decide on these intervals? Did you test that they are statistically distinct?**

**Our response:** We would add a brief statement "Profiles of the various proxies are illustrated in Fig 3 and there is an obvious difference in the character of the fluctuations above and below the depth of 16.5 m (~4.8 Ma). Above 16.5 m, the carbonate content fluctuates at a lower level but with greater amplitude, and the magnetic susceptibility also fluctuates at a greater amplitude. In addition, the CV of most of the records is greater above the boundary than below (Table 1). This suggests that the climate became more humid and variable after 4.8 Ma. Meanwhile, a noticeable drop in deposition rate around 4.8 Ma occurred (Li et al., 2017). Thus, the red clay sequence was divided into two intervals: *Interval Ⅰ* (6.7-4.8 Ma) and *Interval Ⅱ (*4.8-3.6 Ma). The characteristics of the individual proxy records are described in detail below" in front of "Carbonate content" in line 174. We will also add "We use the coefficient of variation (CV) to measure the variability of the records. The higher the CV, the more variable the record. The CV is defined as:

$$CV = 100 * \frac{Standard\ deviation,,}{Mean}$$

at the end of chapter 3(line 172).

**Line 224-226: How are you sure that it relates to monsoon strength? Could it be seasonal or evaporative changes?**

**Our response:** We agree with you. However, on the condition moisture is carried by the monsoon and the monsoon is strong enough, $CaCO_3$ could indicate the monsoon strength (Fang et al., 1999; Sun et al., 2010).

**Lines 235-237: Both statements are significant at 99% confidence?**

**Our response:** The correlation between Sr and CaO* (silicate CaO) is significant at 99% confidence, while the correlation between Sr and $CaCO_3$ is not significant.

**Lines 282-286: Are you sure these signals are real? If so, how might you explain the cycles not related to orbital variability?**

**Our response:** The questions are really worth pondering. Firstly, our chronology is reliable. Secondly, all sampling intervals of XSZ red clay satisfies requirement to detect eccentricity and obliquity signals and 80 % of sampling intervals satisfies requirement to detect the precession signal. Thirdly, three orbital periodic signals were also detected in the other sites of the CLP from late Miocene to early Pliocene, which means changes of orbital parameters really had impact on climate of the CLP (Han et al., 2011; Li et al., 2008). Thus, 100 kyr, 41 kyr and 21 kyr periodic signals recorded by XSZ red clay are probably true.

Changes of Earth orbital parameters would dynamically lead to the variation of the climate. However, the change of Earth orbital parameters is just one of forcing factors and other factors (some internal process or feedback) could magnify or cover the orbital forcing, which means the climate changes probably show non-linear response to orbital forcing. In this specific case, it might be the expansion of the palaeo-EASM that enhanced the orbital periodic signals of XSZ red clay between 4.8 and 4.1 Ma. As for short cycles, the power of these cycles would be weakened by the low and uneven sedimentation accumulation rate (Luo et al., 2017). Meanwhile, the age model has not been astronomically tuned. Thus, it's hard to completely match the filtered 41 kyr and 21 kyr components with the lagged obliquity and precession in phase and amplitude even these signals are real. Our results resemble to those of Han (2011) and Tian (2002) that three orbital periodic signals were significant while records and orbital variability were less matched from late Miocene to early Pliocene.

**Lines 294-295: Doesn't this "incomplete nature of the red climate time series" impact all of the frequency analyses? How can you distinguish real and fake signals?**

**Our response:**

The incomplete nature of the climate time series would also impact on the conservation of the signals especially for precession signal. At least to date, we have not found the driving force yielding

these non-orbital periods. Thus, these non-orbital periodic signals are probably random or fake.

**Line 302: I believe that a 23 kyr filter makes more sense for the climate response to orbital change.**

**Our response:** The 21kyr filtered component is filtered at 18-24kyr. The 23kyr filtered component was included.

**Line 304: What record? Lisiecki and Raymo (2005)?**

**Our response:** The data was the filtered components of the $\delta^{18}O$ record (Zachos et al., 2001) at the 21-kyr, 41-kyr, and 100-kyr bands.

**Line 306: I do not observe this in the filtered record…Is this change significant? How much do these filter components contribute to the complete signal?**

**Our response:** This shift may be not obvious in the 100-kyr filtered components but obvious in 41-kyr and 21-kyr filtered components especially in 41-kyr filtered components. We don't know how to measure the contribution. However, fig 1s shows 100 kyr, 41 kyr and 21 kyr periodic signals of carbonate content in the interval of 4.8-3.6 Ma are more significant than the interval of 6.7-4.8 Ma.

**Line 309-310: Where is this shown? The 41 kyr signal in the benthic records do not well align with the data.**

**Our response:** The shift may be not obvious in Fig 5 where we put all filtered curves together. Fig 4s shows 41 kyr signal in the benthic record and XSZ records enhanced between about 4.8 and 4.1 Ma. In my opinion, three curves have shown some similarities during the period of 4.8-3.6 Ma, with larger oscillation at the intervals of 4.7-4.4 Ma and 4.2-3.9 Ma, and damped oscillation at the interval of 3.9-3.6 Ma. On the other hands, the record has its own climatic significance and limitation, and even the 41 kyr filtered curves of $CaCO_3$ and $\chi_{pedo}$ show difference. Thus, the differences of the 41 kyr signal in benthic records and XSZ records are reasonable.

**Lines 317-319: I see no clear changes in the records. You need statistical support.**

**Our response:** Tables 2s has shown that from the period of 6.7-4.8 Ma to 4.8-3.6 Ma, average value of

CaCO$_3$ decreased, weathering proxies and magnetic susceptibility increased. CV of the most proxies increased.

**Line 361: ODP source?**

**Our response:** Yes, ODP 1148.

**Line 368: "…roughly parallel…" I do not see a correlation. Please quantify.**

**Our response:** We would add "$\chi_{pedo}$ shows a significant positive relationship with $\delta^{18}O$ at 80 % confidence interval (Fig. 4f)" after "pedogenesis proxies roughly parallel to the stacked deep sea benthic foraminiferal oxygen isotope curve" in lines 368-369. We would also add the fig 4f.

**Line 384: This is possible but not necessarily the case.**

**Our response:** Thank you for pointing it out. We would modify "would have resulted" as "could result"

**Lines 392-393: Cooler air can hold less vapor, but this statement is an extreme simplification.**

**Our response:** We would modify the statement of lines 391-393 as "However, moisture sources for the westerly flow are distant from the CLP (Nie et al., 2014), and only a relatively small amount of moisture was carried to the CLP, resulting in a dry and stable climate in the XSZ region."

**Lines 402-403: You record captures seasonal variability?**

**Our response:** Research on migration process of carbonate indicated seasonally wet/dry climate is a key factor in driving carbonate dissolution and reprecipitation, and strong seasonally biased precipitation enhances the leaching process and produces thick leached horizons (Rossinsky and Swart, 1993; Zhao, 1995, 1998). We would add this statement before "The emergence of…"

**Lines 469-471: Citation?**

**Our response:** We would add the citations "(Keigwin et al., 1978; O'Dea et al., 2016)".

**Line 480: Why global moisture and not local moisture?**

**Our response:** It referred to moisture at high northern latitudes. We would remove sentence.

**Lines 484-487: This does not make sense.**

**Our response:** We would remove the statement.

**Lines 491-492: Are you talking about regional or global albedo?**

**Our response:** It's reduced ice albedo at high northern latitudes. We would add "at high northern latitudes" after "…reduced ice albedo".

**Lines 492-495: Citation?**

**Our response:** We would add the citations "(Chang et al., 2011; Sun et al., 2015 )".

**Lines 497-498: Could this discrepancy relate to differences between short term variability and the mean climate state?**

**Our response:** Yes, it's one possibility.

**Line 502: "We noticed"? You mean the authors of these other publications noticed?**

**Our response:** The expression is inappropriate. We would modify it as "Several crucial changes linked with the summer monsoon occurred: There was a vast expansion of the western Pacific warm pool into subtropical regions in the early Pliocene (Brierley et al., 2009; Fedorov et al., 2013), and temperatures at the edge of the warm pool showed a warming trend of ~2℃ from the latest Miocene to the early Pliocene (Karas et al., 2011)".

**Lines 502-506: How close are these events in time?**

**Our response:** All of these events started at 5.2-4.8 Ma and developed at ~4.6 Ma.

**Figures:**

**Figure 1a: The winds do not look correct. Also, 850 hPa winds do not exist over the Plateau**

**Our response:** Thank you for pointing it out. We would correct it and provide new figure (Fig 1).

**Figure 2 and Figure 3 are not cited in the text.**

**Our response:** We would cite the Figure 2 at the end of sentence "…clay is composed of brownish red

and yellowish clay layers" in line 145 and Figure 3 in chapter 4(Results).

**Figure 3: It is difficult to see how the axes align with the lines**

**Our response:** We would use new figure 3 to replace the figure 3

**Figure 5d: Do the black lines represent significance?**

**Our response:** Yes, these lines are the 95% confidence limit line. We would provide new figure (Fig 5).

**Responses to comment 3**

**Major concerns:**

**1) The introduction part is not well written as there are many ambiguity and in accuracy (see detailed comments below). This section needs significant reworking.**

**2) The authors seem to preferentially pick 4.8 Ma as the boundary between the two climate intervals. However, most of the proxies exhibited in Fig. 3 seem has a distinct change at 4.6 Ma, but not 4.8 Ma, e.g., Al$_2$O$_3$, K$_2$O, as well as the three magnetic susceptibility plots. In addition, for the grain size and carbonate content plots, there is no apparent difference below and above 4.8 Ma.**

**3) This manuscript is generally good written in English, but additional efforts are required to polish the language.**

**Our response:** Thanks for your valuable suggestions. We would rework the introduction and would modify and polish the language. We would explain how we defined the boundary later.

**Line-to-line comments:**

**L1-2: I found the title is kind of misleading. The authors emphasize the Tibetan Plateau as the location of their section. However, throughout the manuscript, the Xiaoshuizi section is compared with other sections on the Chinese Loess Plateau, and reflects nothing of the Tibetan Plateau evolution. So it would be more appropriate to emphasize the location as the "western Chinese Loess Plateau".**

**Our response:** It's a very good question. The reason why we emphasize the NE Tibetan Plateau is that it significantly represents the location of the Xiaoshuizi planation surface. It locates the transition zone from the Tibetan Plateau to the Chinese Loess Plateau. The obvious difference between Xiaoshuizi deposit and the red clay in the Chinese Loess Plateau is the modern altitude, and this exactly results from the special geographical position of NE Tibetan Plateau. Compared with other sections on the Chinese Loess Plateau is the main method to address the evolution of palaeo-circulation over the Xiaoshuizi planation from late Miocene to early Pliocene.

**L49: one of the most intensively studied intervals of what? Climate I assume?**

**Our response:** Thank you for pointing it out. We would add "on climate change research" at the end of the sentence.

**L51: in line 41, the authors state closure of the Panamanian Seaway at 4.8 Ma, and it seems that the seaway closure has significant climate effects. Thus, it would be inappropriate to state here that the Zanclean is similar as present due to similar land-sea distribution.**

**Our response:** Thank you for pointing it out. We would remove "(i) land-sea distribution".

**L52-53: references for "comparable temperatures in the tropical region" need to be added.**

**Our response:** We would add the citations "(Herbert et al., 2010, 2016)".

**L54: Zanclean is a period from cold to warm?**

**Our response:** The statement may be inappropriate. We would remove it.

**L66-67: wired transition from the previous sentence. Not consistent.**

**Our response:** We would modify it as "whether permanent El Nino–like conditions were sustained during the Pliocene is still controversial."

**L68: This is at least not accurate, if not wrong. Numerous studies have demonstrated that surface uplift of the Tibetan Plateau is stepwise and spatially diachronous. See reviews of Tapponnier et al., 2001, Wang et al., 2014 and many others. The south-central parts of the Tibetan Plateau were**

uplifted much earlier than the Zanclean, e.g., Paleogene. In the northern Tibetan Plateau, although there might be tectonic deformation in the margins of the Plateau (Li et al., 2015), the major part of the northern Plateau is probably uplifted during the Miocene, as evidenced by numerous other evidence, see review of Yuan et al., 2012. While it's OK to stick on the authors' own preference, it's necessary to discuss/reflect other research progress.

**Our response:** Thank you for pointing it out. What we would like to express is the uplift of the Tibetan Plateau was still underway. We would modify the statement of lines 68-88 as "Meanwhile, the episodic uplift of the TP (Li et al., 2015; Zheng et al., 2000; Fang et al., 2005a, 2005b) and gradual closing of the Panama seaway (Keigwin et al., 1978; O'Dea et al., 2016) were underway. The former substantially influenced the palaeoclimate change (An et al., 2001; Ding et al., 2001; Liu et al., 2014) and the later resulted in reorganization of the global thermohaline circulation (Haug et al., 1998, 2001)".

**L72: 3 Ma or 2.6 Ma? Be accurate.**

**Our response:** It's 2.6 Ma. We would correct it.

**L75: first appearance of ASM in the main text, need to define first. In addition, for summer monsoons in Asia, there is the East Asia Summer Monsoon and the South Asia (India) Summer Monsoon. Which one do you mean? I assume East Asia Summer Monsoon?**

**Our response:** Thank you for pointing it out. We would correct it. Actually, the South Asia Summer Monsoon also put an impact on the XSZ. However, the impact is not as significant as that put by East Asia summer Monsoon. We would modify it.

**L76-77: In the abstract, the authors consider the ASM as moisture carrying, but the Westerlies as moisture lacking. So it's not appropriate to list them together. In addition, moisture transport is short-time climate condition, how could it cause long-term glaciation?**

**Our response:** Thank you for pointing it out. We would rework this part.

**L82: "warm and wet" climate yield "wet" climate? Definitely!**

**Our response:** Thank you for pointing it out. We would remove the sentence.

**L84: a weakened summer monsoon of where? Globally or East Asia only?**

**Our response:** It is East Asia Summer Monsoon. We would remove sentence.

**L89-91: onset of interior Asian aridification since the late Miocene? This is totally unjustified. Numerous studies indicate much earlier onset of Asian interior aridification, e.g., since 22 Ma (Guo et al., 2002), or much earlier at Eocene-Oligocene transition (Dupont-Nivet et al., 2007), or late Eocene (Bosboom et al., 2014).**

**Our response:** We would modify the "..onset of interior Asian aridification related to the uplift of the

**L103-105: is this phenomenon also observed in other studies?**

**Our response:** Yes, pollen assemblages at Chaona indicate a considerably warmer and more humid climate from 4.61-4.07 Ma which was not consistent with that the low magnetic susceptibility values (Ma et al., 2005).

**L107-108: what inconsistent? Need to clarify. For the evidence listed above, it's necessary to point out which region is dominated by westerlies, which is dominated by ASM.**

**Our response:** We would add "where climate is dominated by East Asia Monsoon," after "In eastern and central CLP," in line 94.

**L110: why the western CLP is especially important? Need to give reasons here.**

   **Our response:** We would rework this part. And modify statement of lines 105-110 as "In addition to the East Asian Monsoon, the westerlies also had an impact on climate of East Asia. However, patterns of climate change in westerlies dominated regions were different from the eastern and central CLP during the early Pliocene. Geochemical, stratigraphic and pollen evidence from the Qaidam and Tarim basins has demonstrated that aridification had intensified since the early Pliocene (Fang et al., 2008; Sun et al., 2006a, 2017; Chang et al., 2013; Liu et al., 2014). Although the general climatic trends of the main CLP and northern TP during this period were well recorded, palaeoclimatic change in the NE TP which is at the junction of the westerlies and monsoonal influences remains unclear. Therefore, determining the climatic conditions of the NE TP during the early Pliocene not only improves our understanding of the regional climate change, but also provides insights into the responses of the palaeo-EASM and westerlies to TP uplift and changes in the global climate system at this time."

**L126-127: rejuvenated at what time?**

**Our response:** The tectonic movement of the Pre-Sinian formed the inverse anticlinorium and several faults arranged as en echelon (or parallel) pattern, which lay the foundation for outline of Maxianshan (Li et al., 1990). The planation surface research indicates the tectonic movement since late Miocene made the area uplift with a great amplitude (Li et al., 2017; Ma et al., 2016).

**L175-182: what are the criteria to divide the carbonate content plot into 6.7-4.8 Ma and 4.8-3.6 Ma. I do not see apparent difference between these two subdivisions. For the 6.7-4.8 Ma interval, the carbonate content is 3.8-39.2.The 6.0-5.5 interval, with much smaller amplitude of fluctuation, seems to be more different from the other time intervals.**

**Our response:** Figure 3 may not show the distinct of two periods clearly enough due to variable deposition. Thus, we present figure of records versus stratigraphic depth (Fig 1s). Synthesized the values and variations of the carbonate content, elements content and magnetic susceptibility, a transition period is presented at 16.5-15 m (4.8-4.6 Ma). There is an obvious difference in the character of the fluctuations above and below the depth of 16.5-15 m. For example, above the 16.5 m, the carbonate content fluctuates at a lower level but with larger amplitude accompanied by the noted increase in nodule horizons underlaying leached zones in the field, and the magnetic susceptibility also fluctuates at greater amplitude. Meanwhile, a noticeable drop in deposition rate around 4.8 Ma occurred. Thus, we define the 4.8 Ma as the transition point.

The average value and coefficient of variation of the records during two periods (6.7-4.8 Ma and 4.8-3.6 Ma) have been given in Tables 1s. The coefficient of variation (CV) is defined as:

$$CV = 100 * \frac{\text{Standard deviation}}{\text{Mean}}$$

The higher the CV, the more changeable the record. It shows the average value and CV of the most records show the obvious difference between two periods and most of the records are more changeable during 4.8-3.6 Ma than 6.7-4.8 Ma.

We would explain it in revised version.

**L185-187: looking at Fig. 3, it's pretty hard to determine whether two plots are of similar trend, or opposite trend. I would suggest to provide statistical evaluation to help readers understand the similarity between plots.**

**Our response:** Would add table 2 in the revised version.

**L188: provide the ranges of $Al_2O_3$ and $K_2O$ for the two time intervals. L189: why choose 4.8 Ma as the boundary? The values between 4.8-4.6 seem be more similar as the 6.7-4.8 Ma interval.**

**Our response:** Fig1s shows 16.5-15m (corresponding to 4.8-4.6 Ma) is a transition period. During this period, $K_2O$ and Rb increased while CaO, $Na_2O$ and Sr decreased. The transition period means climate translates from one condition to another. Meanwhile, a noticeable drop in deposition rate around 4.8 Ma occurred. Thus, we define the start of the transition period (4.8 Ma) as the transition point.

We would modify the statement of lines 184-191 as "The XSZ red clay consists mainly of $SiO_2$, $Al_2O_3$, CaO and $Fe_2O_3$ with low concentrations (<5%) of MgO, $K_2O$, $Na_2O$, Sr, Rb and Ba (Table 1). During Interval Ⅰ, $K_2O$ ranges from 1.9-3.3% with an average content of 2.6%. $Na_2O$ ranges from 0.14-1.52% with an average content of 1.2%. Rb ranges from 80-125 ppm with an average content of 103.9 ppm. Sr ranges from 150-252 ppm with an average content of 211.7 ppm. During Interval Ⅱ, $K_2O$ ranges from 2-3.7% with an average content of 3%. $Na_2O$ ranges from 0.94-1.54 % with an average content of 1.23%. Rb ranges from 74-134 ppm with an average content of 109.9 ppm. Sr ranges from 141-281 ppm with an average content of 214.6 ppm. The variations in CaO show the same trend as carbonate content. The variations of Rb and $K_2O$ are synchronous and roughly opposite to that of CaO. The changes of Sr show some similarity with magnetic susceptibility before 4.8 Ma but with CaO after 4.8 Ma. Accordingly, table 2 indicates CaO has positive correlation with $CaCO_3$ and Sr while negative correlation with other elements. The variations in CaO, $K_2O$, Sr and Rb content during 4.8-3.6 Ma are greater than those during 6.7-4.8 Ma, which is also indicated by the CV of these elements (Table 1)."

**L193: similar question, why group values between 6.7-4.8 together, but not include values between 4.8-4.6, which exhibit more similarities as the 6.7-4.8 Ma interval, which are of lower values and smaller amplitude of variation.**

**Our response:** Fig1s shows magnetic susceptibility start to increase from 16.5 m (4.8 Ma).

**L204: there is no difference between these two intervals.**

**Our response:** >40 um content is a proxy for winter monsoon strength. <2um content partly adheres to coarser silt and sand particles. Thus, we do not take two contents into consideration here.

**L206: in Fig. 3, it shows >40 um.**

**Our response:** It's >40 um. We would correct ">43 um" as ">40 um".

**L251: I have a question here, maybe very basic in your discipline. If one wants to use $K_2O/Na_2O$ values to determine the intensity of chemical weathering, a pre-assumption is that before weathering, all the samples have similar $K_2O/Na_2O$ values. Right? How about if the original $K_2O/Na_2O$ values are different? This question might also exist for other chemical proxies used here.**

**Our response:** It's really a good question. Previous studies have indicated the rare earth element distribution patterns of both the loess and red clay are identical to those of upper continental crust (Ding et al., 2001; Jahn et al., 2001). It suggested the initial geochemical composition of well mixed loess and red clay is similar. Thus, changes in $K_2O/Na_2O$ and Rb/Sr ratios are mainly determined by post-depositional weathering.

**L287-288: Could you please explain this in more detail? Which feature in Fig. 5d denotes orbital signal increase since 4.8 Ma? As far as I can infer from Fig. 5d, in the carbonate content plot, the orbital parameters increase since 4.9-5.0 Ma. While, in the Xpedo plot, it seems the increasing timings are diachronous for different orbital parameters.**

**Our response:** Thank you for pointing it out. The explanation for plot Fig 5d may be not appropriate. We consider removing the wavelet plots and the statement "Furthermore, Morlet wavelet transform analysis of both carbonate content and χpedo show that the orbital signal increases since 4.8 Ma (Fig. 5 d)" in lines 286-288.

**L292-295: Here the authors propose that the carbonate content and Xpedo signals reflect incomplete preservation of paleoclimate signals. Then the question is if the original paleoclimate**

**signals are incomplete, how would you use these records to predict paleoclimate changes?**

**Our response:** It involves scale problem. These records cannot document the millennial-scale climate change completely but can document palaeoclimate changes in ten thousand scale. The reason why we said the signals are incomplete is that 20% of sample intervals are not satisfied with the requirement for detecting the precession signal (with the resolution 4 kyr or less).

**L308-312: According to the authors' statement, the rapid change from 6.7-4.8 Ma low amplitude to 4.8-4.1 Ma large amplitude is observed in all the three orbital parameters. But for the benthic foraminiferal d$^{18}$O record, similar change is only observed in the 41-kyr component. Why? This does not read like strong evidence to infer that the wet-dry oscillations were driven by changes in ice volume or global temperature. An associated question would be if the authors do not consider solar radiation intensity is the cause of the wet-dry cycles, but ice volume or global temperature, then what's the cause of ice volume and global temperature changes? Isn't solar radiation intensity a driving factor?**

**Our response:** "This may mean that the increased contrast in wet-dry oscillations at the XSZ site was not driven directly by changes in solar radiation intensity but rather was linked with changes in ice volume or global temperature."(lines 310-312). Sorry, the sentence is an ambiguous or misleading expression. It's the enhancement for wet-dry contrast not wet-dry oscillation, which was linked with changes in ice volume or global temperature. We would correct it. Changes of Earth orbital parameters (linked to solar radiation intensity) would dynamically lead to the variation of the climate including changes in global temperature, ice volume and also wet-dry cycles of XSZ section. However, the change of Earth orbital parameters is just one of forcing factors and some internal process or feedback could magnify or cover the orbital forcing, which means the climate changes probably show non-linear response to orbital forcing (solar radiation intensity). In this specific case, it might be the expansion of the palaeo-EASM that enhanced the orbital periodic signals of XSZ red clay between 4.8 and 4.1 Ma.

In addition, for our records, the most significant change is also observed in 41kyr filtered components. The remarkably increased amplitude of the 41kyr filtered components from XSZ and the deep sea $\delta^{18}$O record at about 4.8 Ma indicating common changes may have synchronously amplified the response of $\delta^{18}$O record and XSZ wet-dry oscillations to obliquity forcing. The variation of deep sea $\delta^{18}$O related to the changes in global temperature and ice volume (Zachos et al., 2001). The

increased wet-dry oscillation of XSZ related to the expansion of palaeo-EASM. Thus, the expansion of palaeo-EASM may be related to changes in global temperature and ice volume. It is not the evidence but just one possibility.

**L317-318: I find this conclusion hard to believe. For the carbonate content signal, the authors state that they record incomplete paleoclimate signal (see comments for L292-295). For the K$_2$O/Na$_2$O and Rb/Sr record, a more apparent change seems to be at 4.6 Ma. If higher carbonate content represents dry climate, and lower carbonate content represents humid climate, compared with 6.7-4.8 Ma, the 4.8-3.6 Ma would have more humid period, but also much drier period, because the 4.8-3.6 Ma has larger variability. While, I did not see a clear wetting trend.**

**Our response:** Firstly, as mentioned above, palaeoclimate signal is incomplete in millennial scale. In ten thousand scale, it is complete. Secondly, carbonate content averaged with 13.8% during the interval of 4.8-3.6 Ma and 17.4% during the interval of 6.7-4.8 Ma, which indicate climate became wetter after 4.8 Ma. In detail, during the interval of 4.8-3.6 Ma, the carbonate content fluctuates with greater amplitude but at a lower level and leached horizons is thicker than the interval of 6.7-4.8 Ma, which means leaching process enhanced after 4.8 Ma. Thirdly, the similar phenomenon was also observed in Xijin loess-paleosol series that in loess layers carbonate has a high average content with small fluctuation amplitude while in paleosol layers carbonate has a low average content with large fluctuation amplitude (Ye, 2017).

**L363-364: This is a false statement. Even at present, the Tibetan Plateau cannot block the Westerlies completely. The Westerlies can travel to the northeastern Tibet through valleys in the Tianshan.**

**Our response:** Thank you for pointing it out. We would remove the statement.

**L368: which plots are pedogenesis proxies? Cite the specific plots here. "roughly"? how rough? Better to give a quantitative value.**

**Our response:** We would add "$\chi_{pedo}$ shows a significant positive relationship with $\delta^{18}$O at 80 % confidence interval (Fig. 4f)" after "pedogenesis proxies roughly parallel to the stacked deep sea benthic foraminiferal oxygen isotope curve" in lines 368-369. We would also add the fig 4f

**L391-392: is there evidence to suggest reduced amount of atmospheric water vapor? Weakening of the palaeo-ASM and dominance of Westerlies can explain the aridity. This does not necessarily need reduced amount of atmospheric water vapor.**

**Our response:** Thanks for your suggestion. We would modify the statement of lines 391-393 as "However, moisture sources for the westerly flow are distant from the CLP (Nie et al., 2014), and only a relatively small amount of moisture was carried to the CLP, resulting in a dry and stable climate in the XSZ region."

**L469: "extremely wet"? wetter than any other period?**

**Our response:** The expression is inappropriate. We would modify "As mentioned above, the extremely wet climate across the CLP was synchronous with the gradual closure of the Panama Seaway, which led to a larger reorganization of the global thermohaline circulation pattern" as "The occurrence of humid climate across the CLP was synchronous with the gradual closure of the Panama Seaway (Keigwin et al., 1978; O'Dea et al., 2016)."

**L528-530: I probably missed it, but how could your records reflect seasonality of precipitation? Which proxy records seasonal signals?**

**Our response:** Research on migration process of carbonate indicated seasonally wet/dry climate is a key factor in driving carbonate dissolution and reprecipitation, and strong seasonally biased precipitation enhances the leaching process and produces thick leached horizons (Rossinsky and Swart, 1993; Zhao, 1995, 1998). The emergence of high-frequency cycles of carbonate eluviation-redeposition and thicker leached horizons indicates that seasonal precipitation increased during the interval of 4.8-3.6 Ma.

**L532: why the strongest summer monsoon is between 4.6-4.25 Ma? What are the possible reasons for the decreasing strength after 4.25 Ma?**

**Our response:** From 4.60-4.25 Ma, pedogenesis and weathering intensity of XSZ reached a maximum, as did precipitation intensity, which is manifested by the enhanced eluviation and carbonate accumulation. These evidences indicate the strength of summer monsoon is strongest from 4.60-4.25

Ma. Meanwhile, the temperatures of both high northern latitude and subtropic Pacific were relatively high, which may be responsible for enhancement of the palaeo-EASM. On the other hand, input of ice raft debris into subarctic northwest Pacific increased from 4.25 to 4.1 Ma which indicates temperature of the high northern latitudes decreased. Temperature at subtropic Pacific also decreased after 4.0 Ma. The decreased temperature of high northern latitude and subtropic Pacific may be the reason for the decreasing strength of palaeo-EASM after 4.25 Ma (Fig. 6 ).

**Figures:**

**Fig. 1: a, the present outline is too large, the wind vectors are too small to see. It's better to show a smaller region with more details; e.g., regions between 10N-50N, 70E-130E. c. highlights the Xiaoshuizi section. Hard to find now.**

**Our response**: Thank you for suggestions. We would modify it. (Fig. 1)

**Fig. 2: These photos exhibit very few useful information.**

**Our response**: These photos visually exhibit information about Xiaoshuizi planation surface and red clay.

**Fig. 3: Between 4.8-4.6 Ma, most plots show a weird shape. Is this because there are limited samples compared with other time intervals?**

**Our response**: Yes, it is. We would use new figure 3 to replace the figure 3

**Fig. 5: apparently the authors need to provide more information in the caption about their plots. For example, Fig. 5d, what does the color mean? What does the black curve represent? Also, the horizontal age scale is better to use Ma, but not ka, as Ma is used throughout the manuscript. In Fig. 5a-b, there are other strong periodicities denoted. How about these periodicities in Fig. 6d?**

**Our response:** We would modify it. (Fig. 5)

**Fig. 6. It will be better to arrange all the proxies with the same logic, e.g., left-wet, right-dry.**

**Our response:** The reason why we put the benthic foraminiferal $\delta^{18}O$ record with different logic from other records is that we would like to show the similarity of benthic foraminiferal $\delta^{18}O$ record and $\chi_{pedo}$

curve during 6.7 to 4.8 Ma.

**Other Modifications:**

**1.    Modifications in abstract**

The mentioned line number refers to that in the discussion manuscript. **In lines** 24-27, "As an analogue for predicting the future climate, Pliocene climate and its driving mechanism attract much attention for a long time. Late Miocene-Pliocene red clay sequence on the main Chinese Loess Plateau (CLP) has been widely applied to reconstruct the history of interior aridification and Asian monsoon climate." was replaced with "The Pliocene climate and its driving mechanisms have attracted substantial scientific interest because of its potential as an analog for near-future climates. The late Miocene-Pliocene red clay sequence of the main Chinese Loess Plateau (CLP) has been widely used to reconstruct the history of interior Asian aridification and Asian monsoon" **In line** 27, "the typical" was removed. **In line** 28, "the" and "(TP)" were added before and after "…Tibetan Plateau…" respectively and "Recently," was replaced with "A". **In line** 29, " has been found" was modified as "sequence was recently discovered". **In line** 30, "Tibetan Plateau (TP)" was modified as "TP". **In lines** 30-32, "To reconstruct the late Miocene-early Pliocene climate history of NE Tibetan Plateau and to assess the regional differences between the central and western CLP, multiple climatic proxies were analyzed from the XSZ red clay sequence" was replaced with "In this study, we analyzed multiple climatic proxies from the Xiaoshuizi red clay sequence to reconstruct the late Miocene-early Pliocene climate history of the NE TP and to assess regional climatic differences between the central and western CLP". **In line** 33, "occurrence of" was added before "...minimal weathering…" and "from 6.7 to 4.8 Ma" was modified as "during 6.7-4.8 Ma". **In line** 34, "implicate" was modified as "indicated" and "sustained" was removed. **In line** 35, "paleo-Asian Summer Monsoon (ASM)" was modified as "palaeo-East Asian Summer Monsoon (EASM)" and "instead" was modified as "that". **In line** 36, "condition" was modified as "conditions". **In line** 37, "the interval of" was removed. **In line** 38, "increasing" was modified as "an increase in", "Thus, we" was replaced with "We" and "obvious" was removed. **In line** 39, "climate transition near 4.8 Ma to the paleo-ASM expansion" was replaced with "climatic transition near 4.8 Ma to the expansion of the palaeo-EASM". **In line** 40, "vast" was removed. "the

subtropical regions and water" was replaced with "subtropical regions and the" . **In line** 41 "freshening in" was replaced with "freshening of". **In line** 43, "carried" was replaced with "transported" and "Tibetan Plateau" was replaced with "TP". **In line** 44, "Xiaoshui Peneplain" was replaced with "Xiaoshuizi Planation surface" and "palaeo-ASM" was modified as "palaeo-EASM".

**2.    Modifications in introduction**

**In line** 49, "on climate change research" was added after "…pre-Quaternary". **In lines** 50-51, "it is analogous to the present day" was replaced with "often used as an analogue for near-future climate conditions" and " (i) land-sea distribution, (ii) orbital configuration, (iii)" was removed. **In line** 52, "(Raymo et al., 1996; Fedorov et al., 2013)" was modified as "(Tripati et al., 2009; Pagani et al., 2010)" and "(iv)" was removed. **In line** 53, "(Herbert et al., 2010, 2016)" was added after "…tropic region". Statement of **lines** 53-55 was removed. **In line** 56, "...unique and some crucial transitions of the" was modified as "markedly different from today and several critical changes in". **In line** 57 "undergoing" was modified as "occurring (Haug et al., 1998, 2005; Lawrence et al., 2006; Chaisson and Ravelo, 2000)" , "For example, the early-Pliocene global mean temperature was approximately 4 ℃ warmer (Brierley and Fedorov, 2010) and the sea levels estimated to have been ~25 m higher than today (Dowsett et al., 2010)" was added before "Temperatures at…" and "the" was removed. The sentence of **lines** 59-61 was replaced with "The zonal and meridional sea surface temperature gradients in the Northern Hemisphere was weak and gradually changed toward a modern much more pronounced spatial temperature contrasts (Fedorov et al., 2013;Brierley et al., 2009, 2010)". **In line** 62, "an "equable" climate" was replaced with "weaker meridional circulation". **In line** 63, "Abbot and Tziperman., 2008" was replaced with "Brierley et al., 2009, 2010" and "the" was modified as "and low". **In lines** 64-65 "is also believed to be low, which is tightly linked with" was replaced with "is believed to have given rise to a". **In lines** 66-67, "debate persists on" was removed and "is still controversial" was added after "Pliocene". The statement of **lines** 68-88 was replaced with "Meanwhile, episodic uplift of the TP (Li et al., 2015; Zheng et al., 2000; Fang et al., 2005a, 2005b) and gradual closing of the Panama seaway (Keigwin et al., 1978; O'Dea et al., 2016) were underway. The former substantially influenced the palaeoclimate change (An et al., 2001; Ding et al., 2001; Liu et al., 2014) and the later resulted in reorganization of the global thermohaline circulation (Haug et al., 1998, 2001). Together these observations imply a structural change in global climate from the early Pliocene to

present. We have to ask what the regional climate like under such special climatic and tectonic settings." The sentence of **lines** 89-91 was replaced with "East Asia is one of the key regions for studying the aridification of the Asian interior and the Asian monsoon evolution which is tightly linked to the uplift of the TP, the regional climate change and global temperature and ice volume evolution (An et al., 2001; Ding et al., 2001; Li et al., 2008; Clift et al., 2008; Nie et al., 2014; Ao et al., 2016;    Sun et al., 2006a, 2017; Chang et al., 2013; Liu et al., 2014). Previous research has revealed that the red clay was widely deposited since the late Miocene across the CLP, indicating that the Asian aridification related to the uplift of the TP enhanced". Statement of **lines** 92-94 was removed. **In line** 94, "the" was added after "in" and "where climate is dominated by East Asian Monsoon" was added after "CLP". **In line** 95, "also" was removed. **In lines** 96-97, "a dry climate condition during late Miocene" was replaced with "dry climatic conditions during the late Miocene" and "however, aridification process was interrupted by a long interval of wet climate" was replaced with "but generally wet climatic conditions". **In line** 99, "of" was replaced with "from". **In line** 100, "reveal" was replaced with "indicate". **In line** 101, "monsoon system" was modified as "summer monsoon intensity". **In line** 103-105, "It's thought to be substantial gleying resulted from large amount precipitation which made magnetic susceptibility invalid over this period" was replaced as "It is thought that waterlogging and iron reduction resulting from high precipitation significantly affected the climatic significance of magnetic susceptibility records during this period". Statement of **lines** 105-110 was replaced with "In addition to the East Asian Monsoon, the westerlies also had an impact on climate of East Asia. However, patterns of climate change in westerlies dominated regions were different from the eastern and central CLP during the early Pliocene. Geochemical, stratigraphic and pollen evidence from the Qaidam and Tarim basins has demonstrated that aridification had intensified since the early Pliocene (Fang et al., 2008; Sun et al., 2006a, 2017; Chang et al., 2013; Liu et al., 2014). Although the general climatic trends of the main CLP and northern TP during this period were well recorded, palaeoclimatic change in the NE TP which is at the junction of the westerlies and monsoonal influences remains unclear. Therefore, determining the climatic conditions of the NE TP during the early Pliocene not only improves our understanding of the regional climate change, but also provide insights into the responses of the palaeo-EASM and westerlies to TP uplift and changes in the global climate system at this time." Statement of **lines** 111-114 was modified as "Continuous red clay sequence was recently discovered on the uplifted Xiaoshuizi (XSZ) planation surface in the NE TP and has been dated via high-resolution

magnetostratigraphy analysis (Li et al., 2017)." The sentence of **lines** 114-117, was modified as "The distinctive geomorphological and climatic characteristics of the XSZ red clay sequence differentiates it from the main CLP red clay, and provides the opportunity to reveal the late Miocene-early Pliocene climate history of the NE TP and to discuss the climatic differences between the central and western CLP". Statement of "multiple climatic proxies have been applied in the Xiaoshuizi…" **lines** 118-120, was modified as "we measured multiple climatic proxies from the late Miocene-Pliocene XSZ red clay core and then the detailed history of precipitation, chemical weathering and pedogenesis during 6.7-3.6 Ma are reconstructed". **In line** 121, "and" was modified as "evolution and its" and "mechanism" was modified as "mechanisms".

3. **Modifications in Regional background**

In line 124, "locates" was modified as "is located". **In line** 25, "the" was removed. **In line** 25, "Maxianshan" was modified as "Maxian". **In line** 131, "covered on" was modified as "covering". **In line** 132, "peneplain" was modified as "planation surface". **In lines** 132-134, "here we choose the Xiaoshuizi core to discuss the regional climate because of its continuous deposit and whole timescale relative to the Shangyaotan core mentioned in Li et al (2017)" as "Here, we use the Xiaoshuizi drill core to reconstruct and discuss the regional climate during the Miocene-Pliocene. The long, continuous well-dated record of the drill core is superior to that of the Shangyaotan core mentioned in Li et al. (2017)". **In line** 136, "china, and the Tibetan Plateau" was modified as "China, and the TP" and "The East Asian Monsoon system and the westerly circulation operate together " was removed.

4. **Modifications in material and methods**

In line 144 "at the base (Fig. 1b)" was added after "…gravel content". **In line** 145, "(Fig. 2)" was added after "yellowish clay layers" and "impregnated with many" was modified as "contains numerous". **In line** 147, "..layer; there are" was modified as "layer. There are also". **In line** 148, "snail fossil" was modified as "fossil snail shell". **In line** 149, "horizons containing" was removed. **In line** 150, "over the Xiaoshuizi", was modified as "across the XSZ". **In line** 151, "all of which are" was modified as "both are, " and "many" was modified as "numerous". **In line** 152, "(Fig. 2b)" was added after "…horizons" and "Grain-size" was modified as "Samples for grain-size" and "samples" was modified as "measurements". **In line** 153, "while" was modified as "and". The sentence of **lines** 153-155 was removed. **In line** 157, "(χ)" was removed. **In line** 161, "using" was modified as "via X-ray fluorescence using a". **In line** 162, "first, the bulk sample was" was modified as "Bulk samples

were". **In line** 163, "…, then each sample was" was modified as " and then", ", and " was modified as

";" and "about" was modified as "~". **In line** 166, "finished" was modified as "conducted". **In line** 167,

"The molar content of silicate Ca (CaO*) was calculated using the following equation" was modified

as "Silicate-bound CaO (CaO*) can be estimated, in principle, by the equation: CaO*(mol) = CaO(mol)

− $CO_2$(calcite mol) − 0.5 $CO_2$(dolomite mol) − 10/3 $P_2O_5$(apatite mol) (Fedo et al., 1995). It is

generally calculated based on the assumption that all the $P_2O_5$ is associated with apatite and all the

inorganic carbon is associated with carbonates Thus, the CaO* of the XSZ red clay was calculated

using the following equivalent equation". **In line** 171 statement of "We use the coefficient of variation

(CV) to measure the variability of the records. The higher the CV, the more variable the record. The

CV is defined as:

$$CV = 100 * \frac{\text{Standard deviation}}{\text{Mean}}$$

Each sample age was estimated using linear interpolation to derive absolute ages, constrained by

our previous magnetostratigraphic study (Fig. 1). The average temporal resolution of the records is 3.8

kyr. Some 80 % of the sequence has a sampling resolution of 4 kyr or less." was added.

5.  **Modifications in results**

Statement of "Profiles of the various proxies are illustrated in Fig 3 and there is an obvious

difference in the character of the fluctuations above and below the depth of 16.5 m (~ 4.8 Ma). Above

16.5 m, the carbonate content fluctuates at a lower level but with greater amplitude accompanied by the

noted increase in nodule horizons underlaying leached zones in the field, and the magnetic

susceptibility also fluctuates at greater amplitude. In addition, the CV of most of the records is greater

above the boundary than below (Table 1). This suggests that the climate became more humid and

variable after 4.8 Ma. Meanwhile, a noticeable drop in deposition rate around 4.8 Ma occurred (Li et al.,

2017). Thus, the red clay sequence was divided into two intervals: *Interval* Ⅰ (6.7-4.8 Ma) and

*Interval* Ⅱ *(*4.8-3.6 Ma). The characteristics of the individual proxy records are described in detail

below." was added before **line** 174. **In lines** 175-176, "According to the fluctuations in carbonate

content, the red clay sequence was divided into two intervals: *Interval -* Ⅰ is from 6.7-4.8 Ma, during

which…" was modified as "During *Interval* Ⅰ,". **In line** 177, "with an average of 17.4%;" was

modified as "and has a high average value (17.4%)" and "the amplitude of fluctuations is small" was

modified as "the carbonate content contrast between leach layers and accumulation layers is generally low". **In line** 178, "(Fig. 3)" was added after "upwards" and "From 5.4-4.9 Ma, the carbonate content" was modified as "From 29-16.5 m, the". **In line** 179, "6.7-5.4 Ma" was modified as "6.7-5.5 Ma" and "*Interval - Ⅱ* is from 42-29 m, during which…" was modified as "During *Interval Ⅱ*,". **In line** 180, "fluctuates from 1.6-39.1% with an average of 13.8%" was modified as "fluctuations have a large amplitude (from 1.6-39.1%) but a low average value (13.8%)." **In line** 181, "From 4.8-3.9 Ma" was modified as "From 16.5-4.5 m ". **In lines** 185-186, "The variations in $Al_2O_3$ and K2O are synchronous and roughly opposite to that of CaO" was replaced with "During *Interval Ⅰ*, $K_2O$ ranges from 1.9-3.3% with an average content of 2.6%. $Na_2O$ ranges from 0.14-1.52% with an average content of 1.2%. Rb ranges from 80-125 ppm with an average content of 103.9 ppm. Sr ranges from 150-252 ppm with an average content of 211.7 ppm. During *Interval Ⅱ*, $K_2O$ ranges from 2-3.7% with an average content of 3%. $Na_2O$ ranges from 0.94-1.54 % with an average content of 1.23%. Rb ranges from 74-134 ppm with an average content of 109.9 ppm. Sr ranges from 141-281 ppm with an average content of 214.6 ppm." The statement of **lines** 187-191was replaced with "The variations of Rb and $K_2O$ are synchronous and roughly opposite to that of CaO. The changes of Sr show some similarity with magnetic susceptibility before 4.8 Ma but with CaO after 4.8 Ma. Accordingly, table 2 indicates CaO shows positive correlation with $CaCO_3$ and Sr, while negative correlation with other elements. The variations in CaO, $K_2O$, Sr and Rb content during 4.8-3.6 Ma are greater than those during 6.7-4.8 Ma, which is also indicated by the CV of these elements (Table 1)." "Magnetic susceptibility also shows pronounced differences between the two intervals (Fig. 3)." was added at start of **line** 193 and "changes" was modified as "ranges". **In line** 195, "whilst" was modified as "and" and "fluctuates" was modified as "ranges". **In line** 199, "larger" was modified as "higher" and "…; the amplitudes and durations" was modified as "... The amplitudes". **In line** 200, "of the three" was modified as "in the three" and "and longer" was removed. **In line** 201, "From 4.8-4.7 Ma, 4.6-4.25 Ma and from 4.1-3.9 Ma" was modified as "From 16-15 m, 13-11 m and 7-5 m". **In line** 202, "…the values of the three parameters are high, and they exhibit peaks from 4.6-4.25 Ma" was removed. **In line** 203, "Grain-size analysis" was modified as "Grain size". **In line** 205, "about 5 Ma, 4.6 Ma and 4.2 Ma" was modified as "about 15m, 12m and 6m". **In line** 206, ">43um" was modified as ">40um" and "Winter Monsoon" was modified as "winter monsoon". **In line** 207, ", as well as to other proxies described above" was added after "…clay content" and ">43um curve" was modified as "variation of the >40 μm fraction". **In line** 208,

"while" was modified as "whereas". **In line** 209, "long-duration" was modified as "lower frequency".

**6. Modifications in discussions**

In line 212, "explanation" was modified as "interpretation". **In line** 213, "varying" was modified as "changing". **In line** 214, "in responses to changes in precipitation and evaporation intensity" was added after "…deposited". **In line** 215, "sequence on the CLP records" was modified as "sequences of the CLP". **In line** 218, "the" was added before "carbonate" and "while" was modified as "whereas". **In line** 224, "So" was modified as "Thus,". **In line** 225, "studying" was modified as "characterizing". **In line** 227, "i.e.," was modified as "e.g." . **In line** 230, "while" was modified as "whereas" and "its" was added after "due to". modify the statement "In addition, previous…". The statement of **lines** 232-234 was modified as "Sr may substitute for Ca in carbonates, which may limit the environmental significance of the Rb/Sr ratio". **In lines** 235-237, "…in the XSZ section, while the correlation between Sr and CaCO3 is not significant (99% confidence interval)" was replaced with "at the 99% confidence interval, while the correlation between Sr and $CaCO_3$ is not significant. This means that the variation of Sr is determined by weathering intensity." **In lines** 237-238, "in our studied samples" was added after "…speculate that" and "in our studied samples (Fig. 4 e and f)." was modified as "(Fig. 4 d and e)". **In lines** 238-252, the statement of "In addition, previous study…" was modified as "In addition, the $K_2O/Na_2O$ ratio is used to evaluate the clay content in loess and is also a measure of plagioclase weathering, avoiding biases due to uncertainties in separating carbonate Ca from silicate Ca (Liu et al., 1993; Buggle et al., 2011). $Na_2O$ is mainly produced by plagioclase weathering and is easily lost during leaching as precipitation increases. By contrast, $K_2O$ (mainly produced by the weathering of potash feldspar) is easily leached from primary minerals and is then absorbed by secondary clay minerals with ongoing weathering (Yang et al., 2006; Liang et al., 2013). In the arid and semi-arid regions of Asia, $K_2O$ is enriched in palaeosols compared to loess horizons (Yang et al., 2006). Thus, high $K_2O/Na_2O$ ratios are indicative of intense chemical weathering. ". **In line** 253, "the" was added before "clay (<2 µm)". **In line** 254, "the ASM" was modified as "EASM". **In line** 255, "Eolian" was modified as "Aeolian". **In line** 258, "Grain size" was modified as "The grain-size". **In line** 259, "confining" was removed , "and" was modified as "to" and "grain size" was modified as "size range". **In line** 260, "proven to be steady" was modified as "shown to be constant" and "χfd can detect" was modified as "Thus, $\chi_{fd}$ can be used". **In lines** 261-262, "can measure" was modified as "is a measure of". **In line** 263, "Figure 4A" was modified as "Figure 4a". **In line** 264, "mostly" was modified as "mainly". **In**

line 265, "eolian" was modified as "aeolian". **In lines** 266-267, "..of the red clay on the XSZ planation surface reflects" was modified as "primarily reflects". The statement of "We use this method…" **in lines** 269-271 was modified as "…, which we use to extract the lithogenic ($\chi_0$) and pedogenic magnetite/maghemite ($\chi_{pedo}$) components. In this study, pedogenic magnetite/maghemite accounts for 11% of the susceptibility ($\chi_{pedo} = \chi_{fd} / 0.11$)." **In line** 272, "Sun et al., 2006" was modified as "Sun and Huang, 2006b" **In line** 276, "the" was removed. **In line** 277, "is proposed to" was modified as "can be use". **In line** 278, "between the" was modified as "of". **In line** 280, "(Fig 6)" was added after "clay".

In line 281, the first "domain" and "the" were removed. **In line** 283-285, "ky" was modified as "kyr". "Morlet wavelet transform analysis of both carbonate content and χpedo show that the orbital signal increases since 4.8 Ma" was modified as "the fluctuations in $CaCO_3$, weathering and pedogenesis indices agree well with orbital eccentricity variations during 4.8-3.9 Ma (Fig. 5). Three orbital periodic signals were also detected in the other sites of the CLP from late Miocene to early Pliocene, which means changes of orbital parameters really had impact on climate of the CLP (Han et al., 2011)". **In line** 289, "As for the non-orbital cycles," was removed and "these" was modified as "non-orbital cycles". **In line** 290, the second "the" was removed. **In line** 291, "deposition" was modified as "depositional" and "degrees of" was added before "pedogenesis". Statement of **lines** 292-295 was replaced with "In the XSZ section, deposition rate was low and uneven, which potentially resulted in the incomplete preservation of the paleoclimatic signal, especially for short precession cycles. Meanwhile, pedogenic and post-depositional compaction would also weaken the orbital signals and produce spurious cycles". Statement of **lines** 297-299 was modified as "Therefore, we speculate that uneven and low deposition rates combined with compaction and leaching processes may weaken the orbital signals and be responsible for presence of non-orbital cycles in XSZ section". **In line** 304, "foraminiferal" was removed and "Our" was modified as "The". **In lines** 305-306, "(especially the 41-kyr component)" was added after "components", "two" was removed and "…change rapidly from very low amplitude from 6.7-4.8 Ma to a much larger amplitude from 4.8-4.1 Ma" was modified as "changes from a low amplitude during 6.7-4.8 Ma to a relatively high amplitude during 4.8-3.9 Ma". **In line** 306, "earth" was modified as "Earth". **In line** 310, "This means that the increased contrast in wet-dry oscillations" was modified as "This may mean that the enhancement for wet-dry contrast".

In line 314, "Multiporxy evidence for the dry climate during the interval of 6.7-4.8 Ma" was modified as "Multi proxy evidence for a dry climate during 6.7-4.8 Ma". **In line** 315, "Based on the

previous mentioned…" was modified as "We used the aforementioned" and "…, we" was modified as "to". **In lines** 316-317, "peneplain, NE Tibetan Plateau" was modified as "planation surface", " and Table 3" was added after "Figure 6", "we observe that" was modified as "there is" and "recorded by" was modified as "in". **In line** 318, "$K_2O/Al_2O_3$" was replaced with "$K_2O/Na_2O$" and "occurred" was removed. **In line** 319, "was generally" was modified as "can be". **In line** 321, "..,$K_2O/Al_2O_3$" was removed, "also" was added before "support" and "occurrence of" was added before "weak chemical". **In line** 322, "Importantly" was modified as "Notably," and "with" was modified as "to". **In line** 324, "intensity" was removed. **In line** 325, "which" was removed and "supports the" was modified as "support the occurrence of". **In line** 326, "the" was modified as "an", "Thus, during this interval the Xiaoshuizi climate was relative arid" was replaced with "the climate at the XSZ site during this interval was relatively arid". **In line** 327, "which" was removed and "pedogenesis" was modified as "pedogenic". **In lines** 328-329, "when these proxies detailed climate changes especially when climate is relative wet" was replaced with "between the carbonate and pedogenic indexes". **In line** 331, "may be" was modified as "is possible that". **In line** 332, "which" was modified as "since 5.5 Ma" and "since 5.5 Ma" at the end of sentence was removed. **In lines** 333-335, the sentence of "However, from…" was modified as "However, the pedogenic indexes indicate that the generally arid climate was interrupted by two episodes of enhanced pedogenesis, at 5.85-5.7 Ma and 5.5-5.35 Ma". **In line** 335, "different" was modified as "differences in the". **In line** 337, "a record of mollusks" was modified as "a coeval mollusk record" and "that" was added after "showed". **In line** 338, "dominating" was modified as "dominated" , "document the" was modified as "indicates that" and "climate condition on…" was modified as "climatic conditions occurred in…". **In line** 339, "the" was added after "during". **In line** 340, "During this interval" was modified as "coeval". **In line** 342, "obvious" was modified as "principal", "the Xiaoshuizi" was modified as "at XSZ the" and "is relative" was modified as "was relatively". **In line** 343, "the" was added before "central". **In line** 344, "obvious" was removed, "instance" was modified as "example" and "at" was added before "6.2-5.8 Ma". **In lines** 345-346, "…in the hinterland of the CLP, but are not recorded by the Xiaoshuizi magnetic susceptibility" was replaced with "in the central and eastern CLP, but are absent in the magnetic susceptibility record from XSZ" and "It is worth noting that" was replaced with "Notably,". **In line** 347, "the" before "Dongwan" was removed before "late Miocene". **In line** 349, "records" was added after "χpedo", "are" was modified as "was" and "from" was modified as "from". **In line** 350, "the" was added before "Summer"

and "Index" was modified as "index". **In line** 353, "6.7-5.2" was modified as "6.7-4.8". **In lines** 353-354, "The only difference is that the climate in the CLP hinterland fluctuated more significantly than that of the Xiaoshuizi red clay" was modified as "However, the only difference was that the climate in the central and eastern CLP fluctuated more substantially than was the case in the vicinity of the XSZ red clay section". **In line** 355, "particularly" was modified as "especially", "western CLP" was removed and "oscillations" was replaced with "climatic oscillations in the western CLP". **In line** 356, "that" was added after "indicate", "ASM" was modified as "EASM" and "on" was modified as "in". **In line** 357, "decreased" was modified as "decreases". **In line** 360, the first "the" was removed and "The weak palaeo-ASM" was modified as "A weak palaeo-EASM". **In line** 361, "has been" was modified as "was" and "from" was modified as "in". **In line** 362, "we deduce that the Asian monsoon" was modified as "we infer that the palaeo-EASM". **In line** 363, "put a small impact on the Xiaoshuizi climate" was replaced with "had only a minor impact on the climate in the study region." **In lines** 363-365 "during late Miocene, the TP was not intensively uplifted and thus it could not block the westerlies completely (Li et al., 2015)" was removed, "Previous" was modified as "previous" and "suggestion" was modified as "indicated". **In line** 367, "This means" was modified as "and thus". **In line** 368, "pedogenesis" was modified as "the variation of the pedogenic" and "parallel to" was modified as "parallels to that of". **In line** 369, ".., that $\chi_{pedo}$ shows a significant positive relationship with $\delta^{18}O$ at 80 % confidence interval (Fig. 4 f)." was added after "(Fig. 6)". **In line** 370, "pedogenesis" was modified as "pedogenic" and "if the precipitation" was modified as "to conclude that precipitation in the study area". **In line** 371, "palaeo-ASM. Thus, we speculate from 6.7 to 4.8 Ma, the precipitation" was modified as "palaeo-EASM and thus, we speculate that from 6.7-4.8 Ma precipitation". **In line** 372, "ASM" was modified as "EASM" and "the climate of our study region" was modified as "regional climate". The sentence of **lines** 376-377 was replaced with "A sustained cooling occurred in both hemispheres during late Miocene and the cooling culminated between 7 and 5.4 Ma (Herbert et al., 2016)." **In line** 379, statement of "In the Northern Hemisphere, transient glaciations appeared when the cooling culminated (Herbert et al., 2016)" was added before "Records". **In line** 380, "northern" was modified as "Northern" and "show" was modified as "indicate". **In line** 382, "In" was modified as "During" and "the" was modified as "a". **In line** 384, "would have resulted" was modified as "could result". **In line** 385, "ASM in the late Miocene" was modified as "EASM". **In line** 386, "(Herbert et al., 2016)," was added after "gradient". **In line** 388, "was" was modified as "is". **In lines** 391-392, "Global

cooling and the growth of polar ice-sheets reduced the amount of atmospheric water vapor; thus, relatively little moisture was carried by the westerlies, producing…" was replaced with "However, moisture sources for the westerly flow are distant from the CLP (Nie et al., 2014), and only a relatively small amount of moisture was carried to the CLP, resulting in…" **In line** 394, "the" was added before "dry climatic condition".

In **line** 395, "enhanced" was modified as "pronounced" and "the interval of" was removed. **In line** 396, "available" was removed. **In line** 397, "Xiaoshuizi climate turns into humid condition from previous arid climate" was modified as "previously arid climate of the XSZ area became humid". **In line** 399, "the interval of" was removed. **In line** 400, "obvious" was removed, "observed from 4.8 to 3.9 Ma. The carbonate…" was replaced with "evident during 4.8-3.9 Ma; the carbonate". **In line** 401, "30%" was replaced with "20%". **In line** 402, "Research on migration process of carbonate indicated seasonally wet/dry climate is a key factor in driving carbonate dissolution and reprecipitation, and strong seasonally biased precipitation enhances the leaching process and produces thick leached horizons (Rossinsky and Swart, 1993; Zhao, 1995, 1998)" was added before "The emergence…" **In line** 403, "was" was removed. **In line** 407, "enhanced" was removed. **In line** 408, "From 4.8 to 3.9 Ma, high" was replaced with "High" and "and the" was replaced with "from 4.8-3.9 Ma and". **In lines** 409-412, "pedogenesis" was modified as "pedogenic", "The K2O/Al2O3 ratio also increased rapidly at about 4.8- 4.7 Ma and maintained relatively high values after 4.7 Ma. This may indicate that the overall weathering intensity was sufficient to produce secondary clays, resulting in a spike in K2O concentration" was removed and "reach the" was replaced with "reached a". **In line** 413, "…as was precipitation intensity, which was manifested by…" was replaced with "..as did precipitation intensity, which is manifested by the…". **In line** 414, "From 3.9 to 3.6Ma" was replaced with "During 3.9-3.6 Ma" and "then" was removed. **In line** 415, "pedogensis intensity weakened" was replaced with "pedogenic intensity also weakened" and ".., which may indicate that the Xiaoshuizi climate is generally humid toward arid direction" was removed. The sentence of **lines** 415-417 was replaced with "Consistent with the records of the XSZ section, mollusk records from Dongwan also indicate the occurrence of warm and humid conditions in the western CLP during the early Pliocene." **In line** 419, "the" was added before "early Pliocene". The sentence of **lines** 419-422 was modified as "Magnetic susceptibility records from the central and eastern CLP are similar to that from the XSZ section in that both the magnitude and the variability are high during 4.8-3.6 Ma" and "enhancement of" was

modified as "increased". **In line** 423, "Obviously" was replaced with "Evidently". **In line** 424, "from 4.6-4.25 Ma in the XSZ section, the χlf" was replaced with "during 4.60-4.25 Ma in the vicinity of the XSZ section, the magnetic susceptibility". The sentence of **lines** 425-428 was modified as "However, a record of $Fe_2O_3$ ratio from Lingtai reveals extremely high values, corresponding to the presence of abundant clay coatings, during 4.8-4.1 Ma and this interval was interpreted as experiencing the strongest EASM intensity in the CLP since 7.0 Ma". **In line** 430, "considerably" was replaced with "substantially". The sentence of **lines** 431-432 was replaced with "These various **lines** of evidence indicate that during 4.60- 4.25 Ma the climate was warm and humid in the central CLP". **In line** 438, "the" was added before "early Pliocene". **In line** 439, "the" was added before "hematite/goethite". **In line** 440, "Smectite/Kaolinite ratio there shows" was replaced with "the smectite/kaolinite ratio indicates" and "about" was modified as "~". **In line** 441, "which indicate the enhancement of palaeo-ASM" was modified as "and thus the enhancement of the palaeo-EASM". The sentence of **lines** 441-443 was replaced with "Therefore, we regard the climatic change evident in XSZ section as the result of the expansion of the palaeo-EASM". **In line** 444, "XSZ" was replaced with "the XSZ section". **In line** 445, "palaeo-ASM may be" was replaced with "the palaeo-EASM may have been". **In lines** 446-447, "decreasing input of ice raft debris into" was modified as "a decrease in the input of ice-rafted debris to the sediments of the". **In lines** 447-448, "palaeo-ASM during early.." was modified as "the palaeo-EASM during the early". **In lines** 450-452, "eastern equatorial Pacific" was replaced with "Eastern Equatorial Pacific" and "These coincides imply that phases of enhanced precipitation may be correlated" was replaced with "This coherence between the record of the XSZ section and marine records implies that phases of enhanced precipitation were correlative".

In line 453, "mechanism for the paleo-ASM" was replaced with "driver of palaeo-EASM". **In line** 454, "the" was added before "uplift". **In line** 455, "The" was removed. **In lines** 456-457, "ASM initiation, having strengthened ASM intensity and changed the shape of the precipitation band in East Asia" was replaced with "EASM in terms of its initiation and strength as well as changing the distribution of the band of high precipitation in East Asia" and "more" was removed. **In line** 465, "very small from" was replaced with "minor from the". **In line** 466, "the" was added before "middle". The sentence of **lines** 465-467 was replaced with "Therefore, we speculate that uplift of the TP was not the major cause of the expansion of the palaeo-EASM at ~4.8 Ma." The sentence of **lines** 469-471 was replaced with "The occurrence of humid climate across the CLP was synchronous with the gradual closure of the Panama

Seaway (Keigwin et al., 1978; O'Dea et al., 2016)". **In line** 471, "the" before "freshening" was removed. **In line** 477, "In particular" was replaced with "Notably". The sentence of "This in turn…" **in lines** 469-471 was removed. **In line** 483, "Arctic" was added before "indicate". **In line** 484, "at present" was replaced with "today". The sentence of "Therefore, even…" **in lines** 484-487 was replaced with "This warmth is also confirmed by other records from high northern latitude regions: diatom abundances and assemblages, pollen data, magnetic susceptibility and sedimentological evidence from Siberia all indicate that the climate was warm and wet in the early Pliocene (Memb B. D. P., 1997, 1999)" and "In contrast, the" was replaced with "The". The sentence of **lines** 489-490 was replaced with "…, and thus the land-ocean thermal contrast was intensified." **In line** 491, "reduced planetary albedo" was replaced with "a reduced ice albedo at high northern latitudes" and "This large land-ocean thermal contrast was essential for enhancing the palaeo-EASM" was added before "On the…" **In line** 494, "(Chang et al., 2011; Sun et al., 2015)" was added after "northward" and "This" was modified as "…, which". **In line** 495, "ASM" was replaced with "palaeo-EASM". **In line** 496, "indicated" was replaced with "shows that" and "at" was replaced with "in". **In line** 497, "It seems to be discrepancy with" was modified as "This appears to be contradictory to the case of the" and "cases" was removed. **In line** 498, "of" was replaced with "in". **In line** 499, "sea-air interaction during early Pliocene is…" was replaced with "that the nature of sea-air interactions during the early Pliocene was…" **In line** 500, "From 4.8 to 4.0 Ma, the thermahaline" was replaced with "From 4.8-4.0 Ma, the thermohaline". **In line** 501, "Chaisson et al., 2000" was modified as "Chaisson and Ravelo, 2000". **In line** 502, "Some" was replaced with "several", "the" was added before "summer" and "We noticed" was replaced with "…: There was". **In line** 503, "occurred in early" was replaced with "in the early". **In line** 504, "….Temperatures" was replaced with "…, and temperatures". **In line** 505, "show" was modified as "showed". **In line** 506, "The thermal" was replaced with "This enhanced thermal". **In lines** 507-508, "In modern times, when the north of western pacific warm pool was" was replaced with "Today, when the northern part of the western pacific warm pool is", the first "the" was removed and "Philippine was" was replaced with "Philippine is". **In line** 509, "Subsequently" was replaced with "…; and subsequently" and "shifted" was replaced with "shifts". **In line** 510, "was" was replaced with "is". The sentence of **lines** 511-512 was replaced with "Further research is needed to determine if this was also the case during the early Pliocene", "the" was added before "warming" and "seawater" was removed. **In line** 511, "Wang et al., 2000;" was removed. **In line** 513, "subtropic" was replaced with "the

subtropical" and "been more readily evaporated" was replaced with "promoted increased evaporation". **In line** 514, "palaeo-ASM leading to" was replaced with "palaeo-EASM, resulting in". **In line** 515, "Thus, we deduce it may be" was replaced with "In conclusion, we infer that the". **In line** 516, "the subtropical" was replaced with "subtropical" and "freshening of" was replaced with "the freshening of the". **In line** 517, "palaeo-ASM during" was replaced with "palaeo-EASM during the".

**7.  Modifications in conclusions**

**In lines** 520-522, "Continuous late Miocene-Pliocene red clay preserved on the representative planation surface in NE Tibetan Plateau provides particular opportunity to discuss the Asian monsoonhistory. Multi-proxy records from the XSZ planation surface in the western CLP,…" was modified as "The continuous late Miocene-Pliocene red clay sequence preserved on the planation surface in the NE Tibetan Plateau provides the opportunity to elucidate the history of the Asian monsoon in the western CLP. Multi-proxy records from the XSZ section,…" **In line** 524, "the XSZ records indicate that" was removed. **In line** 525, "over the XSZ section" was added after "precipitation". **In line** 526, "the hinterland of the" was replaced with "the central and eastern CLP" and "ASM" was replaced with "EASM". **In line** 527, "during this interval" was replaced with "at this time" and "the XSZ records" was replaced with "the records from the XSZ section". **In line** 528, "From 4.8 and 3.6 Ma, the" was replaced with "The". **In line** 532, "the interval of 4.6-4.25 Ma" was replaced with "4.60-4.25 Ma" and "palaeo-ASM" was replaced with "the palaeo-EASM". **In line** 533, "region" was added after "Arctic", "the vast" was replaced with "A vast" and "into the" was replaced with "into". **In line** 534, "water in" was removed. **In line** 535, "the" was added before "early Pliocene".

**8.  Modifications in references**

The citation style was changed. References **in line** 550-551, 581-582, 586-587, 662-663, 672-673 674-676 and 710-711 was removed. The following references were added:

[revised manuscript text omitted]

Li, J. J., Ma, Z. H, Li, X. M., Peng, T. J., Guo, B. H., Zhang, J., Song, C. H., Liu, J. Hui, Z. C., Yu, H., Ye, X. Y., Liu, S. P., Wang, X. X.: Late Miocene-Pliocene geomorphological evolution of the Xiaoshuizi peneplain in the Maxian Mountains and its tectonic significance for the northeastern

Tibetan Plateau. Geomorphology. 295 393-405, 2017.

Li, Z. Q., Liu, Q. H., Sun, B. J.: Germorphic features and types of Xinglong mountians. Journal of Gansu Agricultural University,25(3),303-312, 1990.

Luo, Z., Su, Q., Wang, Z., Heermance, R. V., Garzione, C., Li, M. Ren, X. P., Song, Y. G., and Nie, J. S.: Orbital forcing of plio-pleistocene climate variation in a qaidam basin lake based on paleomagnetic and evaporite mineralogic analysis. Palaeogeography Palaeoclimatology Palaeoecology, doi:org/10.1016/j.palaeo.2017.09.022, 2017.

Ma, Y. Z., Wu, F. L., Fang, X. M., Li, J. J., An, Z. S., and Wei, W.: Pollen record from red clay sequence in the central Loess plateau between 8.10 and 2.60 Ma. Chinese Science Bulletin, 50(19), 2234-2243, 2005.

Ma Zhenhua.: The planation surfaces and their Late Cenozoic geomorphological evolution in Maxianshan Mountains, NE Tibetan Plateau. Lanzhou: Lanzhou University Masters dissertation, 2016.

Reeder, S., Taylor, H., Shaw, R.A., Demetriades, A.: Introduction to the chemistry and geochemistry of the elements. In: Tarvainen, T., de Vos, M. (Eds.), Geochemical Atlas of Europe. Part 2, 2006. Interpretation of Geochemical Maps, Additional Tables, Figures, Maps, and Related Publications. Geological Survey of Finland, Espoo, pp. 48-429

Rossinsky Jr., V., Swart, P. K.: Influence of climate on the formation and isotopic composition of calcretes. In: Swart, P.K., Lohmann, K.C., McKenzie, J., Savin, S. (Eds.), Climate Change in Continental Isotopic Records, American GeophysicalUnion: Geophysical Monography, 78, pp. 67-75, 1993.

Sun, Y. B., An, Z. S., Clemens, S. C., Bloemendal, J., andVandenberghe, J.: Seven million years of wind and precipitation variability on the Chinese Loess plateau. Earth & Planetary Science Letters, 297(3–4), 525-535, 2010.

Ye, X. Y.: The time series establishment and paleoclimatic period evolution of Xijin Loess. Lanzhou: Lanzhou University Masters dissertation, 2017.

Zachos, J., Pagani, M., Sloan, L., Thomas, E., and Billups, K.: Trends, rhythms, and aberrations in global climate 65 Ma to present. Science, 292(5517), 686-93, 2001.

Zhao, J. B.: A study of the CaCO$_3$ illuvial horizons of paleosols and permeated pattern far rain water, J Geogr Sci, 15(4), 344-350, 1995.

Zhao, J. B.: Illuvial CaCO3 layers of paleosol in loess and its environmental significance, Journal of Xi'an Engineering University, 20(3), 46-49, 1998.

Hopefully the revised version is now satisfactory for publication in Climate of the Past.

Best Regrads,

Jijun Li

[Figure]

Fig. 1s.   Spectrum analysis of carbonate content during the period of (a) 4.8-3.6 Ma (b)  6.7-4.8 Ma on original

[revised manuscript text omitted]

 high value, corresponding to the presence of abundant clay coating

 4.8  -4.1 Ma and this interval was interpreted as experiencing

the strongest EASM intensity in the CLP since 7.0 Ma (Ding et al., 2001). In addition, the

relative intensity of pedogenic alteration of the grain-size distribution was the strongest

during the interval from 4.8-4.2 Ma in the Lingtai section (Sun et al., 2006c). Pollen

assemblages at Chaona indicate a substantially  warmer and more humid climate

from 4.61-4.07 Ma (Ma et al., 2005). These evidence indicate that during

4.60- 4.25 Ma the climate was warm and humid  in the central CLP.

Gleying has been implicated in reducing the value of magnetic susceptibility as a record of

precipitation during this period (Ding et al., 2001). When soil moisture regularly exceeds the

critical value, dissolution of ferrimagnetic minerals occurs and the susceptibility signal is

negatively correlated with pedogenesis (Liu et al., 2003). This by itself indicates that

precipitation was likely to have been very high during this interval.

In summary, a wet climate prevailed across the CLP in the early Pliocene. At the same

time, the hematite/goethite ratio from the SCS also indicates  enhanced precipitation

amount and the smectite/ kaolinite ratio indicates  increased

seasonality at  ~4.8 Ma (Fig. 7 i and j), and thus the  enhancement of

the palaeo-EASM (Clift et al., 2006, 2014). Th we regard the climatic change

evident in XSZ  section as the result of the expansion of the

Palaeo-EASM .

The remarkably increased amplitude of the 41-kyr filtered components from the XSZ

section and the deep sea $\delta^{18}O$ record at about 4.8 Ma indicates the expansion of the palaeo-EASM may have been related to changes in global temperature and ice volume. Furthermore, a decrease  in the input of  ice-rafted debris to the sediments of the  subarctic northwest Pacific was synchronous with the expansion of the palaeo-EASM during the early Pliocene (Fig. 6). In addition, from 4.8-4.7 Ma and 4.6-4.25 Ma, the high values of the three pedogenic indices at the XSZ section indicate that strong pedogenic intensity corresponded with high SSTs in the  Eastern  Equatorial Pacific (EEP). These coherence between the record of the XSZ section and marine records  implies  that phases of enhanced precipitation were  correlated with changes in SST and ice volume (or temperature) at northern high latitudes.

**5.4 Possible driver of  palaeo-EASM expansion during early Pliocene**

Ding (2001) proposed that the uplift of the TP to a critical elevation resulted in an enhanced summer monsoon system during 4.8-4.1 Ma.  TP uplift was shown to have had profound effects on the EASM in terms of its initiation and  strength as well as  changed distribution of the band of high  precipitation  in East Asia (Li et al., 1991, 2014; An et al., 2001). A  detailed modeling study demonstrated that the uplift of the northern TP mainly resulted in an intensified summer monsoon and increased precipitation in northeast Asia (Zhang et al., 2012). From 8.26-4.96 Ma, massive deltaic conglomerates were widely deposited and the sediment deposition rate increased, indicating the uplift of the Qilian Mountains (Song et al., 2001). At the same time, the Laji Mountains underwent  pronounced uplift by thrusting at  ~8 Ma, which resulted in the current basin-range pattern (Li et al., 1991; Fang et al.,

2005a; Zheng et al., 2000). However, geological and palaeontological records indicate that the uplift of the eastern and northern margins of the TP was very minor small from the late Miocene to the middle Pliocene (Li et al., 1991, 2015; Zheng et al., 2000; Fang et al., 2005a, 2005b). Therefore, we So we speculate that uplift of the TP was uplift may be not the major cause con of the tribution to the expansion of the palaeo-EASM at ~ occurred at 4.8 Ma.

[revised manuscript text omitted]

Lu, H., Zhang, F., Liu, X., & and Duce, R. A. (2004).: Periodicities of palaeoclimatic variations recorded by loess-paleosol sequences in chinaChina. Quaternary Science Reviews, 23(18–19), 1891–1900, 2004.

Lunt, D. J., Valdes, P. J., Haywood, A., &Rutt, I. C. (2008). Closure of the panama seaway during the pliocene: implications for climate and northern hemisphere glaciation. Climate Dynamics, 30(1), 1-18.

Ma, YuzhenY. Z., Wu, F. L.uli, FangANG, X. M.iaomin, Li, J. J.ijun, & An, Z. S.hisheng, and Wei, Wet

[revised manuscript text omitted]

---

## Author Response (AR2)

Dear, Editor Ran Feng,

We have carefully revised and edited the manuscript entitled "Late Miocene-Pliocene climate evolution recorded by the red clay covered on the Xiaoshuizi planation surface, NE Tibetan Plateau" based on the valuable comments and suggestions from four anonymous reviewers. We have canceled the subdivision of climate stages and modified the boundary of humid period from 4.8 Ma to 4.7 Ma. Below please find the detailed responses.

**Response to Report #1**

This is my second round of review of this manuscript. Compared with the last version, the revised manuscript is significantly improved. I have no major concerns but some minor questions and suggestions as listed below.

**Our response:** Many thanks for reviewing our manuscript again. We specially thank you for the valuable questions and suggestions. They have really helped improve our paper. Here are responses.

**P82-84: Red clay indicates aridification? And why this indicates uplift of TP? The logics here are not clear. In the next sentence, the authors state that red clay could reflect either arid or humid climate.**

**Our response:** Yes, the wide deposition of red clay during the late Miocene indicates an increased dust source area (related to the Asian aridification) and enhancement of the East Asian Winter Monsoon. It does not mean a limited local aridification but rather a generalized Asian aridification (Sun et al., 1998; Li et al., 2017). Asian aridification is not evidence for the uplift of the TP but it is related to the uplift. This expression is ambiguous. We remove "related to the uplift of the TP".

**P103: what are the climate trends in the northern TP? There is no description. Maybe "central Asia" or "NW China" is more accurate?**

Our response: Thanks for pointing it out. We would modify the "northern TP" to "central Asia".

**P112: "climatic characteristics"? what does this mean?**

**Our response:** The statement is ambiguous. We would modify the "The distinctive geomorphological and climatic characteristics of the XSZ red clay sequence differentiates it from the main CLP red clay, and " to " Due to its specific geographical location, the XSZ red clay ".

P124: better to show "Maxianshan" and "Longzhong Basin" in Fig. 1c.

Our response: We indicate the "Maxianshan" and "Longzhong Basin" in Fig.1c.

P147: From Fig. 1b, carbonate nodule horizons are more abundant in the upper part, e.g., < 4.8 Ma. So does appearance of carbonate horizons also reflect climate changes?

**Our response:** Yes, the Bk horizons with larger carbonate content consist of carbonate nodule horizons underlying leached zones in the field indicate a significant translocation of carbonate minerals from Bw horizons to deeper Bk horizon due to greater rainfall (He et al., 2013).

**P192: I can understand that larger CV indicates more variable climate, but how could you infer more humid from CV?**

Our response: Larger CV is not evidence for a more humid climate. A humid period is reflected in high

frequency occurrence of leaching (Bw) horizons with low carbonate content (< 8 %), a larger carbonate content contrast between Bw horizons and accumulation (Bk) horizons and intermittent enhancement of magnetic susceptibility. The Bk horizons with larger carbonate content consisting of carbonate nodule horizons underlying leached zones in the field indicate the significant translocation of carbonate minerals from Bw horizons to deeper Bk horizon due to greater rainfall.

P193: The depositional rate is not a good evidence here. As there are also other abrupt drops or increase. Why do you pick the 4.8 Ma out? I suggest to delete this evidence.

Our response: Thanks for your suggestion. We will take it into consideration.

L331: which part of Fig.5? There are many sub-parts in Fig.5, better to denote each clearly.

Our response: Many thanks. We modify "Fig.5" as "Fig.5 c".

L345-346: From Table 3, comparing the two intervals, the Rb/Sr difference is only 0.02 and the K2O/Na2O difference is 0.14, no more than 3.5 % of the exhibited value range (e.g., 0.3-0.9 for Rb/Sr; and 1-5 for K2O/Na2O). Does this small difference truly reflect climatic difference? What are the analytical errors of these parameters? Could this small difference be interpreted as analytical errors, or the small difference actually indicate no difference between interval I and interval II.

**Our response:** Firstly, the measurement error ranged from 0.1%-0.3%. Secondly, changes in chemical weathering indices show some similarity with the carbonate content that weathering intensity increased in Bw horizons and remarkably decreased in Bk horizons, which means that the differences between the two intervals are not analytical errors. Thirdly, the unusually low weathering indices values of interval II resulting from translocation of mobile elements from Bw horizons to deeper Bk horizons make the difference between the two intervals small.

**L348-350: Here, the authors indicate that the <2um/>40um ratio is characterized by low values. However, this contradicts with values presented in Table 3. This is apparently a false claim.**

**Our response:** Thanks for pointing it out. We would consider the limitations of the <2um/>40um ratio as a pedogenesis index. We add the statement: "Noticeably, during this interval of peak precipitation (4.6-4.25 Ma), the enhancement of  $<2 \mu m />40 \mu m$  ratio is not as strong as that of  $\chi$ pedo, which may indicate that the former is of limited value when pedogenic intensity is strong" after "carbonate accumulation" in line 441.

**L435: apparently, from Table 3, the <2um/>40um ratio is low during interval II**

**Our response:** We consider the limitation of the <2um/>40um ratio as a pedogenesis index. We add the statement "Noticeably, during this interval of peak precipitation (4.6-4.25 Ma), the enhancement of  $<2 \mu m />40 \mu m$  ratio is not as strong as that of  $\chi$ pedo, which may indicate that the former is of limited value when pedogenic intensity is strong" after "carbonate accumulation" in line 441.

**Response to Report #2**

The authors investigate a new red-clay serious with special regard to palaeoclimate reconstruction in a critical region of the Tibetan Plateau. The authors find an abrupt change in precipitation character (to wetter and seasonally more varied conditions) after 4.8Ma, which they interpret as an increased

westward reach of the East Asian Summer Monsoon (EASM). This work is important as it helps constrain this westward extension of EASM on the Tibetan Plateau during the late Miocene and Pliocene. Our response: Many thanks for reviewing our manuscript and giving approval to our work. Here, we provide some quick replies to your questions.

**General Comments:**

Overall, the manuscript is well structured and well written. However, important methodical details are omitted in the presentation of results. The language is good, although minor mistakes and occasional poor phrasing can be found throughout the manuscript. I recommend proofreading of a proficient English speaker. My major concern pertains to the lack of use or description of statistics and signal processing. The main example of this is the seemingly arbitrary subdivision of the record into Interval I and Interval II. I suggest either finding a more robust way for change point detection, or refraining from this subdivision altogether. The broad implications of this study remain the same even without the somewhat unnecessary subdivision. Furthermore, the term significant is used several times, but it is unclear how significance was determined. Furthermore, it is unclear how the statistical dependence between several variables was established. I urge the authors to take advantage of the existing tools in statistics to substantiate such claims. There needs to be a section (in "3. Materials and Methods") describing these methods.

**Our response:** Many thanks for the valuable suggestions which we take into consideration in the revised version. We have canceled the subdivision of climate stages. We add the statement: "After interpolation to a 3-kyr sampling interval we performed spectral analysis on detrended records of carbonate content and  $\chi_{pedo}$  using Redfit bases on a Lomb-Scargle Fourier transform combined with a Welch-Overlapped Segment Averaging procedure. We applied Gaussian band-pass filters at frequency of 0.09090-0.01111, 0.02174-0.02778 and 0.04167-0.05556 kyr-1 to extract oscillations associated with the 100-kyr, 41-kyr and 21-kyr periodicities. The significance of correlation is based on two-tailed test" after "4kyr or less" in line183.

Lastly, while the inclusion of discussion of possible climate change drivers is important, I urge the authors to highlight the uncertainties of these more, since only limited evidence is presented here.

**Our response:** Thanks for your valuable suggestions. We add statement: "However, there are several uncertainties associated with such an explanation. For example, the timing of the closure of the Panama Seaway is still debated (Bacon et al., 2015; O'Dea et al., 2016), and it is unclear how strongly these changes influenced the palaeo-EASM. Addressing these questions requires more geological evidence and precise model simulations of the early Pliocene climate. The value of our study lies in proposing the potential linkage of the evolution of palaeo-EASM and changes in temperatures of high northern latitudes and SSTs of the low latitude Pacific Ocean in the early Pliocene" at the end of line 546.

Note that I cannot comment on details about sample preparation and methods pertaining to proxy reconstructions as those lie outside my fields of expertise. I strongly recommend that other reviewers with

complementary expertise comments on these to compensate.

While I do believe this work to be valuable to several geoscientific communities, I can recommend publication only after the points above have been adequately addressed and a reviewer with a stronger geochemical background has commented on the manuscript.

**Additional Specific Comments:**

In the introduction, the authors correctly point out many problems of using the Pliocene as an analogue for future climate. An additional factor compromising the Pliocene as an analogue are the differences in palaeotopography.

L 50 "in" instead of "on" Our response: Many thanks. We modify "on" to "in".

L63 It should say "contrast" (not plural) to be consistent with the rest of the sentence, and maybe using "gradient" would be less confusing. What zonal gradients are you referring to? Do you simply mean zonal differences (differences along the same latitudes)? In this case, the term gradient may be a tad misleading, as it is used to describe the slope or one directional change as you have in case of meridional gradients.

**Our response:** Thanks for your suggestions. We modify the "contrasts" in line 64 to "contrast". Zonal gradient refers to the sea surface temperature contrast between the eastern and western equatorial Pacific. We modify "low east-west sea surface temperature gradient" in line 66 to "minor east-west surface temperature contrast".

**L74 "latter" instead of "later"**

Our response: Many thanks. We modify "later" to "latter".

L75-76 By "structural changes", I assume you are referring to spatial structure and mean that the regional expressions of global climate change were highly varied? Our response: Yes. We add "spatial" before "structural" in line 75.

L76 change to "... the regional climate is like ..."

Our response: We add "was" before "like".

L84 "was enhanced" instead of "enhanced"

Our response: We add "was" before "enhance".

L96 needs some rephrasing

Our response: We modify "waterlogging" to "dissolution of ferrimagnetic minerals" in line 95.

L112 What are these distinct geomorphological and climatic characteristics? Do you mean the above

**described geographical and climatic setting?**

**Our response:** Yes. The statement is ambiguous. We modify the "The distinctive geomorphological and climatic characteristics of the XSZ red clay sequence differentiates it from the main CLP red clay, and " to "Due to its specific geographical location, the XSZ red clay ".

L187 Does the coefficient of variation change significantly for ALL of the records? How was this established? By looking at it purely qualitatively, I would subdivide the section above the division up again into a higher variability lower part and lower variability upper part (that is similar to what is below the currently drawn line). In other words, the subdivision into interval I and interval II seems rather arbitrary. It looks to me like there is only a brief period of higher variability from 15-10m, interrupting the period of relatively low variability.

**Our response:** Many thanks for your questions. In fact, other reviewers have mentioned this problem before. We have been looking for evidence to show that two periods are different but we seem to be failed. The prime reason why we subdivide the records into two intervals is for the sake of discussion. We now remove the subdivision and highlight the humid and more variable climate period from ~16-5 m (4.7-3.9 Ma). We modify the statement of lines 186-195 to "Profiles of the various environmental proxies are illustrated in Figure 3. Notably, there is evidence for a relatively wet interval from ~16-5 m (4.7-3.9 Ma) which is reflected in the high-frequency occurrence of Bw horizons with a low carbonate content (< 8 %) and intermittent enhancement of magnetic susceptibility. There is a large contrast in carbonate content between Bw horizons and accumulation (Bk) horizons, which corresponds to variations in elemental contents. The Bk horizons, with a higher carbonate content, consist of carbonate nodule layers underlying leached zones in the field indicate the substantial translocation of carbonate minerals from Bw horizons to Bk horizons due to greater rainfall (He et al., 2013)."

**L264 What correlation analysis is this based on and how was significance determined?**

**Our response:** It is based on a two-tailed test. We now give the significance test table for correlation between  $CaO^*$ ,  $CaCO_3$  and Sr (Table 3).

**L342 How was the significance of this change determined (see above)?**

**Our response:** The "significant" used here is qualitative and subjective. We would remove the subdivision of climate stages and remove the sentence of lines 342-344.

**L393 Again, how was this relationship and significance determined?**

**Our response:** It is based on a t-test. The statement may be inappropriate. We remove "a significant" and "at 80% confidence interval".

**Fig. 1c: Please use standard units like km and hPa, not miles and mb**

Our response: We modify it.

**References**

- He, T., Chen, Y., Balsam, W., Qiang. X.K., Liu, L.W., Chen, J., Li, F.J.: Carbonate leaching processes in the Red Clay Formation, Chinese Loess Plateau: Fingerprinting East Asian summer monsoon variability during the late Miocene and Pliocene. Geophysical Research Letters, 40(1):194-198, 2013.
- Li, J. J., Ma, Z. H, Li, X. M., Peng, T. J., Guo, B. H., Zhang, J., Song, C. H., Liu, J. Hui, Z. C., Yu, H., Ye, X.Y., Liu, S. P., Wang, X. X.: Late Miocene-Pliocene geomorphological evolution of the Xiaoshuizi peneplain in the Maxian Mountains and its tectonic significance for the northeastern Tibetan

Plateau. Geomorphology. 295,393-405, 2017.

Sun, D.H., An, Z.S., Shaw, J., Bloemendal, J., Sun, Y.B.: Magnetostratigraphy and palaeoclimatic significance of Late Tertiary aeolian sequences in the Chinese Loess Plateau. Geophysical Journal International, 134(1):207-212, 1998.

**Other modifications**

**1. Modifications in abstract and introduction**

**1: In line 34, "6.7-4.8 Ma" is modified to "late Miocene".**

**2: In line 38, "occurred during 4.8-3.6 Ma" is modified to "occurred intermittently during 4.7-3.9 Ma".**

**3: In line 39, "4.8" is modified to "4.7".**

**4: In line 50, "on" is modified to "in".**

**5: In line 64, "contrasts" is modified to "contrast".**

**6: In line 67, "low" is modified to "minor" and "gradient" is modified to "contrast".**

**7: In line 74, "later" is modified to "latter".**

**8: In line 76, "spatial" is added before "change".**

**9: In line 86, "related to the uplift of the TP" is modified to "was".**

**10: In line 98, "waterlogging" is modified to "dissolution of ferrimagnetic minerals".**

**11: In line 106, "northern TP" is modified to "central Asia".**

**12: In line 114, "The distinctive geomorphological and climatic characteristics of the XSZ red clay sequence differentiates it from the main CLP red clay, and " is modified to " Due to its specific geographical location, the XSZ red clay ".**

**13: In line 158, "was" is modified to "were pre-treated with 10%  $H_2O_2$  to remove organic material, with 10% HCl to remove carbonates, and with 0.05 mol/L of (NaPO3)6 for dispersion. They were then".**

**2. Modifications in material and methods**

**1: In lines 150-151, "red layers" is modified to "red soil layers (Bw) characterized by loam and moderate medium angular blocky structure."**

**2: In line 166, "with an error of 0.1%-0.3%" is added after "PW2403".**

**3: In line 188, "After interpolation to a 3-kyr sampling interval, we performed spectral analysis on detrended records of carbonate content and  $\chi_{pedo}$  using Redfit, based on the Lomb-Scargle Fourier transform combined with a Welch-Overlapped Segment averaging procedure. We applied Gaussian band-pass filters at frequencies of 0.09090-0.01111, 0.02174-0.02778 and 0.04167-0.05556 kyr-1 to extract oscillations associated with the 100-kyr, 41-kyr and 21-kyr periodicities, respectively. The significance of the correlations is based on a two-tailed test" was added after "4 kyr or less".**

**3. Modifications in results**

**1: In lines 196-206, "and there is an obvious difference in the character of the fluctuations above and below the depth of 16.5 m (~ 4.8 Ma). Above 16.5 m, the carbonate content fluctuates at a lower level but with greater amplitude accompanied by the noted increase in nodule horizons underlaying leached zones in the field, and the magnetic susceptibility also fluctuates at greater amplitude" is modified to "Notably, there is evidence for a relatively wet interval from ~16-5 m (4.7-3.9 Ma) which is reflected in the high-frequency occurrence of leaching (Bw) horizons with a low carbonate content (< 8 %) and intermittent enhancement of magnetic susceptibility. There is a large contrast in carbonate content between Bw horizons and accumulation (Bk) horizons, which corresponds to variations in elemental contents. The Bk horizons, with a higher carbonate content, consist of carbonate nodule layers underlying leached zones in the field indicate the substantial**

translocation of carbonate minerals from Bw horizons to Bk horizons due to greater rainfall (He et al., 2013)." #2: In line 207, "above the boundary than below" is modified to "during this interval than in other intervals" #3: In line 208, "It suggests" is modified to "These various forms of evidence suggest".

**4: In line 209, "after 4.8 Ma" is modified to "during 4.7-3.9 Ma" and sentence of lines 209-211 was removed. #5: The statement of lines 215-228 is modified to "The carbonate content of the entire core fluctuates from 1.6-39.2% with an average of 15.9 %. From 42-16 m, the average carbonate content is high (17.1%) and the carbonate content decreases upwards. The contrast in the carbonate content between the Bw and Bk horizons is generally low; for the Bw horizons, the carbonate content is ~12% and values <8% are rare. Bk horizons, with a carbonate content of around or above 21%, are frequent (Fig. 3). From 16-5 m, there are fluctuations in carbonate content of large amplitude (1.6-39.1%) but the average value is low (13.3%). Leaching-accumulation horizons (Bw-Bk) are frequent; the Bw horizons have a carbonate content of <8%, while that of the Bk horizons is >21%. From 5-0 m, the average carbonate content increases to 15.5%; Bw horizons with a carbonate content <8% is absent, and the carbonate content contrast between the Bw and Bk horizons is low."**

**6: The statement of lines 230-240 is modified to " $K_2O$  ranges from 1.9-3.7% with an average of 2.8%; Na2O ranges from 0.14-1.54% with an average of 1.2%; Rb ranges from 74-134 ppm with an average of 106.2 ppm; and Sr ranges from 141-281 ppm with an average of 212.8 ppm. The variations in CaO exhibit the same trend as carbonate content with high values in Bk horizons and low values in Bw horizons. The changes of Sr show some similarity with magnetic susceptibility prior to 4.7 Ma but with CaO after 4.7 Ma."**

**7: In line 241, "Accordingly, table 2" is modified to "Reference to Table 2".**

**8: In lines 242-243, "From 16-5 m, CaO and Sr exhibit low values in Bk horizons and high values in Bw horizons, while the opposite is the case for  $K_2O$  and Rb" was added.**

**9: The statement of lines 244-247 is modified to "Finally, during 4.7-3.9 Ma, the amplitudes of the fluctuations in CaO,  $K_2O$ , Sr and Rb are greater than in the other intervals."**

**10: The statement of lines 249-269 is modified to "Variations of  $\chi_{hf}$ ,  $\chi_{lf}$  and  $\chi_{fd}$  are synchronous.  $\chi_{hf}$  ranges from 9.6-53.9×10-8 m3/kg with an average of 21.8×10-8 m3/kg;  $\chi_{lf}$  ranges from 11.4-59.0×10-8 m3/kg with an average of 23.1×10-8 m3/kg; and  $\chi_{fd}$  ranges from 0-4.7×10-8 m3/kg with an average of 1.2×10-8 m3/kg. From 42-16 m, three magnetic parameters show relatively flat and low values:  $\chi_{hf}$  ranges from 9.6-33.3×10-8 m3/kg; and  $\chi_{fd}$  ranges from 11.4-36.1×10-8 m3/kg with an average of 20.3×10-8 m3/kg; and  $\chi_{fd}$  ranges from 0-2.8×10-8 m3/kg with an average of 1.0×10-8 m3/kg. From 16-5 m, the values and amplitudes of three parameters are high:  $\chi_{hf}$  ranges from 13.8-53.9×10-8 m3/kg; and  $\chi_{fd}$  ranges from 0-4.7×10-8 m3/kg. From 16-5 m, the values and amplitudes of three parameters are high:  $\chi_{hf}$  ranges from 13.8-53.9×10-8 m3/kg; and  $\chi_{fd}$  ranges from 0-4.7×10-8 m3/kg with an average of 1.6×10-8 m3/kg. From 16-15 m, 13-11 m and 7-5 m, the values of the three parameters obviously increase. From 5-0 m, both the value and amplitudes of three parameters decrease:  $\chi_{hf}$  ranges from 12.8-32.9×10-8 m3/kg; with an average of 22.0×10-8 m3/kg;  $\chi_{lf}$  ranges from 13.6-34.6×10-8 m3/kg. The fluctuation of magnetic susceptibility is substantially different from that of carbonate content which indicates enhancement of magnetic susceptibility did not caused by leaching of the carbonate."**

**11: The sentence of lines 271-273 is modified to "The clay content ( $\leq 2 \mu m$ ) ranges from 3.8-13.5% with an average of 8.1%; and the >40 um content ranges from 0.7-13.5% with an average of 6%."**

**12: In line 274, "which correspond to peaks in magnetic susceptibility" is added after "6 m".**

**13: In line 276, "as well as to other proxies described above" is removed and "In addition, from 21-5 m the fluctuations in the >40 um fraction are roughly the inverse to those of magnetic susceptibility" is added.**

**14: In line 278, "From 6.7-4.8 Ma" is modified to "From 42-21m".**

**15: In line 279, "4.8 Ma" is modified to "21m".**

**4. Modifications in the discussion**

**1: In line 347, "Fig. 5" is modified to "Fig. 5 a-b".**

**2: In line 352, "4.8-3.9 Ma (Fig. 5 d)" is modified to "4.7-3.9 Ma (Fig. 5 d)".**

**3: In lines 374-375, all of the "4.8 Ma" are modified to "4.7 Ma" and "Fig. 5" is modified to "Fig. 5d".**

**4: In line 383, "6.7-4.8Ma" is modified to "the late Miocene".**

**5: In line 385-388, "As shown in Figure 6 and Table 3, there is a significant change in most of the proxies**

(carbonate, Rb/Sr, K2O/Na2O and xpedo) near 4.8-4.7 Ma, and therefore the climatic record can be divided into

two intervals. During interval I (6.7-4.8 Ma)," is modified to "During the late Miocene".

**6: In line 393, "<2 µm/>40 µm ratio" is removed.**

**7: In line 421, "interval of 6.7-4.8 Ma" is modified to "the late Miocene".**

**8: In line 347, "a significant" is removed.**

**9: In line 348, "at 80 % confidence interval (Fig. 4 f)" is modified to "(Fig. 4 e)".**

**10: In line 441, "from 6.7-4.8 Ma" is modified to "during the late Miocene".**

**11: In line 445, "from 6.7-4.8 Ma" is modified to "during the late Miocene".**

**12: In line 469, "Humid climate with pronounced fluctuations during 4.8-3.6 Ma" is modified to**

"Intermittently humid climate during the early Pliocene".

**13: In line 470, "interval II (4.8-3.6 Ma)" is modified to "the early Pliocene".**

**14: In line 471, "from ~4.7 Ma" is added after "humid".**

**15: In lines 473-474, all of "4.8 Ma" are modified to "4.7 Ma".**

**16: In lines 475, "7%" and "20%" are modified to "8%" and "21%" respectively.**

**17: The sentence of the lines 485-486 is modified to "Seasonal precipitation was intermittently enhanced from**

4.7-3.9 Ma, and so was weathering and pedogenic intensity".

**18: In lines 489-491, "Notably, during this interval of peak precipitation (4.6-4.25 Ma), the enhancement of the  $<2 \mu m />40 \mu m$  ratio is not as strong as that of  $\chi_{pedo}$ , which may indicate that the former is of limited value when pedogenic intensity is strong" is added.**

**19: In line 498, "4.8-3.6 Ma" is modified to "the early Pliocene".**

**20: In line 523, "4.8 Ma" is modified to "4.7 Ma".**

**21: In line 523, "the" is added before "early".**

**22: In line 547, "4.8 Ma" is modified to "4.7 Ma".**

**23: In line 549, "et al" is removed.**

**24: In lines 586-588, "A modelling experiment indicates that the precipitation of the CLP would increase when the tropical warm pool expended into subtropical region (Brierley et al., 2009)" is added after "extention". #25: In line 599, " may have" is added before "facilitated".**

**26: In line 600, the statement of "However, there are several uncertainties associated with such an explanation. For example, the timing of the closure of the Panama Seaway is still debated (Bacon et al., 2015; O'Dea et al., 2016), and it is unclear how strongly these changes influenced the palaeo-EASM. Addressing these questions requires more geological evidence and precise model simulations of the early Pliocene climate. The value of our study lies in proposing the potential linkage of the evolution of palaeo-EASM and changes in temperatures of high northern latitudes and SSTs of the low latitude Pacific Ocean in the early Pliocene" is added.**

**5. Modifications in coclusion**

**1: In line 613, "two interval of" is modified to "the".**

**2: In line 614, "the first interval (6.7-4.8 Ma)," is modified to "the late Miocene".**

**3: In line 617, "the second interval (4.8-3.6 Ma)," is modified to "the early Pliocene".**

**4: In line 619, "was large" is modified to "increased from 4.7-3.9 Ma".**

**6. Modifications in references**

References of "Bacon, C. D., Silvestro, D., Jaramillo, C., Smith, B. T., Chakrabarty, P., and Antonelli, A.: Biological evidence supports an early and complex emergence of the Isthmus of Panama. Proceedings of the National Academy of Sciences,112(19), 6110-6115, 2015" and "He, T., Chen, Y., Balsam, W., Qiang. X.K., Liu, L.W., Chen, J., Li, F.J.: Carbonate leaching processes in the Red Clay Formation, Chinese Loess Plateau: Fingerprinting East Asian summer monsoon variability during the late Miocene and Pliocene. Geophysical Research Letters, 40(1):194-198, 2013" are added.

**7. Modifications in Figures and tables**

**1: In Fig. 1 c, "Longzhong Basin" and "Maxianshan" are added.**

**2: Fig. 3 and Fig.4 are replaced by new Fig.3.and Fig.4 respectively.**

**3: In Fig. 5c, the dash line is removed.**

**4: In Fig. 6, the dash line is removed and labels are used in each graph.**

**5: In Fig. 7, the boundary line is removed.**

**6: Table1 Table2 and Table3 are all replaced with new Tables.**

Hopefully the revised version is now satisfactory for publication in Climate of the Past.

Best regards,

Jijun Li

[revised manuscript text omitted]

218 content is high (17.1%) and the carbonate content decreases upwards.; t The contrast in the

| 219                                                                                                                | carbonate content between the leach layers Bw and accumulation layers Bk horizons is                                                                                                                                                                                                                                                                                                                                                                                                                                                                                                                                                                                                                                                                    |
|--------------------------------------------------------------------------------------------------------------------|-----------------------------------------------------------------------------------------------------------------------------------------------------------------------------------------------------------------------------------------------------------------------------------------------------------------------------------------------------------------------------------------------------------------------------------------------------------------------------------------------------------------------------------------------------------------------------------------------------------------------------------------------------------------------------------------------------------------------------------------------------------------------|
| 220                                                                                                                | generally low; for the Bw horizons, the carbonate content is ~12% and values <8% are rare.                                                                                                                                                                                                                                                                                                                                                                                                                                                                                                                                                                                                                                                                     |
| 221                                                                                                                | Bk horizons, with a carbonate content of around or above 21%, are frequent and the                                                                                                                                                                                                                                                                                                                                                                                                                                                                                                                                                                                                                                                                                    |
| 222                                                                                                                | carbonate content decreases upwards (Fig. 3). From 29-16.5 m, the fluctuations are of greater                                                                                                                                                                                                                                                                                                                                                                                                                                                                                                                                                                                                                                                                         |
| 223                                                                                                                | amplitude than during 42-29 m. From During Interval H 16-5 m, there are fluctuations in                                                                                                                                                                                                                                                                                                                                                                                                                                                                                                                                                                                                                                                                        |
| 224                                                                                                                | carbonate content of a-large amplitude (1.6-39.1%) but the average value is low                                                                                                                                                                                                                                                                                                                                                                                                                                                                                                                                                                                                                                                                                       |
| 225                                                                                                                | (13.83%)From 16.5-4.5 m there are several Bw-Bk horizons are frequent; the Bw horizons                                                                                                                                                                                                                                                                                                                                                                                                                                                                                                                                                                                                                                                                         |
| 226                                                                                                                | have a carbonate content of $<78\%$ , while that of the leached layersBk_horizons is $>2021\%$ .                                                                                                                                                                                                                                                                                                                                                                                                                                                                                                                                                                                                                                                               |
| 227                                                                                                                | From the accumulation layers 5-0 m, the average carbonate content increases to 15.5%; Bw                                                                                                                                                                                                                                                                                                                                                                                                                                                                                                                                                                                                                                                                              |
| 228                                                                                                                | horizons with a carbonate content <8% is absent, and the carbonate content contrast between                                                                                                                                                                                                                                                                                                                                                                                                                                                                                                                                                                                                                                                                           |
| 229                                                                                                                | the Bw and Bk horizons is low.                                                                                                                                                                                                                                                                                                                                                                                                                                                                                                                                                                                                                                                                                                                                        |
| 230                                                                                                                | Element geochemistry                                                                                                                                                                                                                                                                                                                                                                                                                                                                                                                                                                                                                                                                                                                                                  |
| 231                                                                                                                | The XSZ red clay consists mainly of SiO 2 , Al 2 O 3 , CaO and Fe 2 O 3 with low                                                                                                                                                                                                                                                                                                                                                                                                                                                                                                                                                                                                                               |
| 232                                                                                                                | concentrations (<5%) of MgO, K 2 O, Na 2 O, Sr, Rb and Ba (Table 1). During Interval [], K 2 O                                                                                                                                                                                                                                                                                                                                                                                                                                                                                                                                                                                                                                       |
|                                                                                                                    |                                                                                                                                                                                                                                                                                                                                                                                                                                                                                                                                                                                                                                                                                                                                                                       |
| 233                                                                                                                | ranges from 1.9-3. $\frac{37}{9}$ % with an average of 2. $\frac{68}{9}$ %; Na 2 O ranges from 0.14-1. $\frac{5254}{9}$ % with an                                                                                                                                                                                                                                                                                                                                                                                                                                                                                                                                                                                                                          |
| 233
234                                                                                                         | ranges from 1.9-3.37% with an average of 2.68%; Na 2 O ranges from 0.14-1.5254% with an average of 1.2%; Rb ranges from $\frac{8074}{125}$ ppm with an average of 103.96.2 ppm; and                                                                                                                                                                                                                                                                                                                                                                                                                                                                                                                                                                        |
| 233
234
235                                                                                                  | ranges from 1.9-3.37% with an average of 2.68%; Na 2 O ranges from 0.14-1.5254% with an average of 1.2%; Rb ranges from $\frac{8074}{125}$ ppm with an average of $103.96.2$ ppm; and Sr ranges from $\frac{150}{252141}$ ppm with an average of $\frac{211.7212.8}{212.8}$ ppm. During Interval II,                                                                                                                                                                                                                                                                                                                                                                                                                                                       |
| 233
234
235
236                                                                                           | ranges from 1.9-3. 37 % with an average of 2. 68 %; Na 2 O ranges from 0.14-1. 5254 % with an average of 1.2%; Rb ranges from 8074 - 125-134 ppm with an average of 10 3.96.2 ppm; and Sr ranges from 150-252141-281 ppm with an average of 211.7212.8 ppm. During Interval II , K2O ranges from 2-3.7% with an average content of 3%. Na2O ranges from 0.94-1.54 % with                                                                                                                                                                                                                                                                                |
|  <li>233</li> <li>234</li> <li>235</li> <li>236</li> <li>237</li>                                         | ranges from 1.9-3. 37 % with an average of 2. 68 %; Na 2 O ranges from 0.14-1. 5254 % with an average of 1.2%; Rb ranges from 8074-125-134 ppm with an average of 10 3.96.2 ppm; and Sr ranges from 150-252141-281 ppm with an average of 211.7212.8 ppm. During Interval II , K 2 O ranges from 2 3.7% with an average content of 3%. Na 2 O ranges from 0.94-1.54 % with an average content of 1.23%. Rb ranges from 74-134 ppm with an average content of 109.9                                                                                                                                                                                                    |
|  <li>233</li> <li>234</li> <li>235</li> <li>236</li> <li>237</li> <li>238</li>                            | ranges from 1.9-3. 37 % with an average of 2. 68 %; Na 2 O ranges from 0.14-1. 5254 % with an average of 1.2%; Rb ranges from 8074-125-134 ppm with an average of 10 3.96.2 ppm; and Sr ranges from 150-252141-281 ppm with an average of 211.7212.8 ppm. During Interval II , K 2 O ranges from 2 3.7% with an average content of 3%. Na 2 O ranges from 0.94-1.54 % with an average content of 1.23%. Rb ranges from 74-134 ppm with an average content of 109.9 ppm. Sr ranges from 141-281 ppm with an average content of 214.6 ppm. The variations in                                                                                                            |
|  <li>233</li> <li>234</li> <li>235</li> <li>236</li> <li>237</li> <li>238</li> <li>239</li>               | ranges from 1.9-3.37% with an average of 2.68%; Na 2 O ranges from 0.14-1.5254% with an average of 1.2%; Rb ranges from 8074-125-134 ppm with an average of 103.96.2 ppm; and Sr ranges from 150-252141-281 ppm with an average of 211.7212.8 ppm. During Interval II , K 2 O ranges from 2 3.7% with an average content of 3%. Na 2 O ranges from 0.94-1.54% with an average content of 1.23%. Rb ranges from 74-134 ppm with an average content of 109.9 ppm. Sr ranges from 141-281 ppm with an average content of 214.6 ppm. The variations in CaO exhibit the same trend as carbonate content_with high values in Bk horizons and low                                                                                    |
|  <li>233</li> <li>234</li> <li>235</li> <li>236</li> <li>237</li> <li>238</li> <li>239</li> <li>240</li>  | ranges from 1.9-3.37% with an average of 2.68%; Na 2 O 
[revised manuscript text omitted]

---

## Author Response (AR3)

Dear, Editor Ran Feng,

We have carefully revised and edited the manuscript entitled "Late Miocene-Pliocene climate evolution recorded by the red clay covered on the Xiaoshuizi planation surface, NE Tibetan Plateau" based on your valuable comments and suggestions. This time, we mainly revised the discussion of possible mechanisms for palaeo-EASM expansion during early Pliocene. Many thanks for spending so much time to review our manuscript. Wish you and yours have a happy Spring Festival in advance.

**Judging by previous reviews, I lean towards trusting your geochemical and sedimentary analyses. However, I do have concerns about your discussion. I am worried that the discussion of mechanisms accounting for EASM expansion during early Pliocene was built upon misunderstanding of early Pliocene tropical SST records. The records cited in your discussion did not cover late Miocene, and hence should not be used to argue for expansion of monsoon starting from early Pliocene. The long records from western tropical Pacific actually suggest similar and warmer than present-day SSTs for both early Pliocene and late Miocene.**

**I recommend rewrite and shorten the discussion. Plesae stick with interpretations from sedimentary and geochemical features to monsoonal climate. The discussion of mechanisms reads very weak, and as shown above, erroneous at various places.**

**Our response:** Many thanks for your valuable suggestions. We remove the statement of lines 488-498 and lines 546-570.

**See details in the below:**

**Line 566 to 572, Brierley et al., 2009; Fedorov et al., 2013 did not suggest that equatorial SSTs remain stable or cooled. Instead, Brierley et al. suggests that warm pool was expanded during the early to mid-Pliocene. Zhang et al., (2014, science) actually suggests that western tropical Pacific was significantly warmer during the late Miocene and early Pliocene. The argument from line 566 and 575 is based on misunderstanding of published literatures.**

**Our response:** We would remove this part.

**Line 572 to line 575, the cited studies provide no theoretical basis on how the northern high latitude warming would affect ITCZ position. This argument again has no support.**

**Our response:** We modify the statement as "On the other hand, the unusually warm Arctic and the West Antarctic ice-sheet expansion by 6–5 Ma (Zachos et al., 2001, 2008) steepened interhemispheric thermal gradient and could further cause the thermal equator to move northward (Chiang and Friedman, 2012; Broecker and Putnam, 2013). This facilitated the northwestward expansion of the palaeo-EASM, which is also proposed as the driving mechanism for northwestward migration of the monsoon rain belt for the warm Holocene (Yang et al., 2015)."

**Line 582 to 584, again, misinterpretation of Brierley et al and Fedorov et al. Notice that the tropical Pacific records discussed in these papers DO NOT cover late Miocene for the most part. You CANNOT cite them to highlight the uniqueness of early Pliocene. In fact, based on**

**Zhang et al., 2012, the western tropical Pacific during late Miocene is not that different from early Pliocene.**

**Our response:** We remove the statement.

**Thanks for taking time to revise the manuscript! I am looking forward to your submission!**

**Other modifications**

**1: In lines 40-42, the statement of "the poleward expansion of the tropical warm pool into subtropical regions, and the freshening of the subtropical Pacific" is removed.**

**2: The statement of lines 488-498 is removed.**

**3: Lines 533-536, the statement of "The warming of the northern high latitude region led to increases in summer temperature in the mid-latitudes of Eurasia. However, equatorial SSTs remained stable or cooled slightly (Brierley et al., 2009; Fedorov et al., 2013), and thus the land-ocean thermal contrast was intensified. Furthermore," is modified to "Furthermore, a decrease in the input of ice-rafted debris to the sediments of the subarctic northwest Pacific was synchronous with the expansion of the palaeo-EASM during the early Pliocene (Fig. 6). The warming of the Northern Hemisphere and".**

**4: In line 537, "also" is removed.**

**5: In line 546-548 "and small meridional heat gradient in the Northern Hemisphere pushed the Intertropical Convergence Zone northwards (Chang et al., 2013; Sun et al., 2015), which weakened the westerly circulation and thus facilitated the northwestward expansion of the palaeo-EASM" is modified to "and West Antarctic ice-sheet expansion by 6–5 Ma (Zachos et al., 2001, 2008) steepened interhemispheric thermal gradient and could further cause the thermal equator to move northward (Chiang and Friedman, 2012; Broecker and Putnam, 2013). This facilitated the northwestward expansion of the palaeo-EASM which is also proposed as the driving mechanism for northwestward migration of the monsoon rain belt for the warm Holocene (Yang et al., 2015)."**

**6: The statement of lines 549-570 is removed.**

**7: In line 571, "In conclusion" is modified to "Therefore".**

**8: In line 572-574, "accompanied by the vast poleward expansion of the tropical warm pool into subtropical regions and the freshening of the subtropical Pacific," is modified to "in response to the closure of the Panamanian Seaway".**

**9: In line 581, "and SSTs of the low latitude Pacific Ocean" is removed.**

**10: In lines 595-597, "the vast poleward expansion of the tropical warm pool into subtropical regions, and the freshening of the subtropical Pacific," is removed.**

**11: References of line635, line 698 and line702 are removed and "Broecker, W.S., Putnam A.E.: Hydrologic impacts of past shifts of Earth's thermal equator offer insight into those to be produced by fossil fuel CO2. Proc Natl Acad Sci USA 110(42):16710–16715, 2013. Chiang J. C. H., Friedman A. R.: Tropical cooling, interhemispheric thermal gradients, and tropical climate change. Annu Rev Earth Planet Sci 40(1):383–412, 2012. Yang, S.L., Ding, Z.L., Li, Y.Y., Wang, X., Jiang, W.Y., Huang, X.F.: Warming-induced northwestward migration of the East Asian monsoon rain**

belt from the Last Glacial Maximum to the mid-Holocene. Proc. Natl. Acad. Sci. USA 112, 13178–13183, 2015. Zachos, J.C., Dickens, G.R., Zeebe, R.E., 2008. An early Cenozoic perspective on green-house warming and carbon-cycle dynamics. Nature.451, 279–283" are added.

**12: Fig 6 h and i are removed.**

Hopefully the revised version is now satisfactory for publication in Climate of the Past.

Best regards,

Jijun Li

[revised manuscript text omitted]